# Resolving ecological feedbacks on the ocean carbon sink in Earth system models

David I. Armstrong McKay*[1,2], Sarah E. Cornell[1,2], Katherine Richardson[3], Johan Rockström[1,4]

[1]Stockholm Resilience Centre, Stockholm University, Stockholm, 106 91, Sweden
[2]Bolin Centre for Climate Research, Stockholm University, Stockholm, 106 91, Sweden
[3]Globe Institute, Center for Macroecology, Evolution and Climate, University of Copenhagen, Copenhagen, 2100, Denmark
[4]Potsdam Institute for Climate Impact Research, Potsdam, 14473, Germany

*Correspondence to*: David I. Armstrong McKay (david.armstrongmckay@su.se)

**Abstract.** The Earth's oceans are one of the largest sinks in the Earth system for anthropogenic $CO_2$ emissions, acting as a negative feedback on climate change. Earth system models predict that climate change will lead to a weakening ocean carbon uptake rate as warm water holds less dissolved $CO_2$ and biological productivity declines. However, most Earth system models do not incorporate the impact of warming on bacterial remineralisation and rely on simplified representations of plankton ecology that do not resolve the potential impact of climate change on ecosystem structure or elemental stoichiometry. Here we use a recently-developed extension of the cGEnIE Earth system model (ecoGEnIE) featuring a trait-based scheme for plankton ecology (ECOGEM), and also incorporate cGEnIE's temperature-dependent remineralisation (TDR) scheme. This enables evaluation of the impact of both ecological dynamics and temperature-dependent remineralisation on particulate organic carbon (POC) export in response to climate change. We find that including TDR strengthens POC export relative to default runs due to increased nutrient recycling (+~1.3%), while ECOGEM weakens POC export by enabling a shift to smaller plankton classes (-~0.9%). However, interactions with carbonate chemistry cause opposite sign responses for the carbon sink in both cases: TDR leads to a smaller sink relative to default runs (-~1.0%) whereas ECOGEM leads to a larger sink (+~0.2%). Combining TDR and ECOGEM results in a net strengthening of POC export (+~0.1%) and a net reduction in carbon sink (-~0.7%) relative to default. These results illustrate the degree to which ecological dynamics and biodiversity modulate the strength of the biological pump, and demonstrate that Earth system models need to incorporate ecological complexity in order to resolve nonlinear climate-biosphere feedbacks.

## 1. Introduction

Oceans absorb about a quarter of anthropogenic carbon dioxide emissions, drawing down around 2-3 PgCy[-1] in recent decades (Ciais et al., 2013; Friedlingstein et al., 2019; Gruber et al., 2019). The mechanisms of carbon sink processes are well understood: solubility (dissolution) and biological (soft tissue and hard carbonate) pumps transfer carbon to the deep ocean where it remains on timescales of several centuries to millennia (Broecker and Peng, 1982). However, increasing ocean temperature as a result of global warming could potentially lead to a weakening of this ocean carbon sink (Arora et al., 2013;

Ciais et al., 2013). The global carbon sink uptake rate was observed to decline by $\sim 0.91\% \, y^{-1}$ between 1959 and 2012, of which approximately 40% is estimated to be due to feedback responses of sink processes (nonlinear carbon-cycle responses to $CO_2$ and carbon–climate coupling) with the oceans playing a large role (Raupach et al., 2014). The combined effect of future feedbacks on both land and ocean carbon sinks reduce the RCP4.5-compatible anthropogenic carbon budget by $\sim 157 \pm 76$ PgC
(Ciais et al., 2013).

This sink weakening might therefore act as a positive feedback on anthropogenic warming (Steffen et al., 2018). However, many of the Earth system models (ESMs) used to make these carbon sink projections do not incorporate sufficient ecological complexity to fully resolve these feedbacks, including for the ocean the impact of both warming and acidification on metabolic
dynamics, ecosystem structure, and nutrient stoichiometry (Ciais et al., 2013). Of the ten ESMs from the Coupled Model Intercomparison Project Phase 5 (CMIP5) used for carbon sink projections in the fifth assessment report of the Intergovernmental Panel on Climate Change (IPCC AR5), only one resolves the impact of warming on organic carbon remineralisation, three resolve different plankton sizes, and three resolve changing nutrient usage ratios (discussed in Background below), all of which critically influence the biological pump in a warming ocean. While there have been some
improvements in the next generation CMIP6 ESMs, most still use a fixed remineralisation parameterisation for exported organic carbon and feature broad size classes rather than a full spectrum of plankton size classes.

In this study we investigate changes in the biological pump in response to climate change and ocean acidification using ecoGEnIE, an ESM of intermediate complexity (EMIC) with more complex biogeochemistry and ecosystem dynamics than
present in most CMIP ESMs. The ecoGEnIE model allows temperature-dependent remineralisation, greater biodiversity via size trait-based plankton ecology, and flexible elemental stoichiometry. This combination allows the impact of metabolic and ecological dynamics on the biological pump and the ocean carbon sink in response to climate change to emerge, while the choice of an EMIC makes such additional complexity computationally tractable. We simulate a suite of historical and future climate change scenarios and assess the impact on the ocean carbon sink of replacing the default remineralisation
parameterisation with the temperature-dependent scheme and/or the parameterised biogeochemistry module with ecoGEnIE's new explicit trait-based plankton ecology scheme.

This manuscript is structured as follows. In section 2 we give detailed background on the role of the biological pump, how it may be affected by climate change and ocean acidification, and to what extent current Earth system models resolve these
effects. In section 3 we describe the ecoGEnIE model and our experimental setup. In section 4 we describe the results of our experiments, focusing on the contrasting results for biological pump strength and the ocean carbon sink across the different model configurations. And finally in section 5 we discuss the implications and limitations of our results.

## 2.    Background

The primary driver of a weakening ocean carbon sink in response to anthropogenic climate change is the reduced $CO_2$ dissolution capacity of warmer water (i.e. a weaker solubility pump), but changes in the biological pump modulate this physicochemical process by affecting the vertical partitioning of carbon within the ocean. In general Earth system models project reduced export production (i.e. a weakening of the biological pump) as a result of ocean stratification reducing nutrient availability (Bopp et al., 2013), but a reduced efficacy of the biological pump due to increased marine bacterial respiration has also been suggested as an important factor in past warm episodes (Boscolo-Galazzo et al., 2018; John et al., 2014a; Olivarez Lyle and Lyle, 2006).

[Figure 1]

The biological pump describes the fixation and export of carbon and nutrients from the surface to the poorly ventilated deep ocean by biological activity. The vast majority of this organic matter is remineralised as it sinks and is later gradually returned in dissolved form to surface waters by ocean upwelling (Figure 1). The formation and export of calcium carbonate shells (Particulate Inorganic Carbon; PIC) also forms part of the biological pump, but hereafter we focus on the soft-tissue biological pump as it is the dominant driver of surface carbon export (Dunne et al., 2007).

After organic carbon is fixed in the surface euphotic layer by phytoplankton and some is consumed by zooplankton, Particulate Organic Matter (POM) begins to be remineralised by detritivorous bacteria as it falls through the water column as POM rain. Most POM is remineralised to Dissolved Organic Matter (DOM) within the epipelagic mixed layer (~0-200m) where the nutrients released are rapidly recycled into 'regenerated' production (Dugdale and Goering, 1967) and in the mesopelagic zone (~200-1000m) below, but up to 4-12 $PgCy^{-1}$ of Particulate Organic Carbon (POC) leaves the surface ocean (Ciais et al., 2013; Dunne et al., 2007; Henson et al., 2011, 2012; Mouw et al., 2016a). This remineralisation profile follows a power law-like distribution, with a rapid geometric decline in export flux from the base of the mixed layer to a small asymptotic flux by ~1000m. Once in the poorly ventilated deep ocean, the surviving POM (most of which is subsequently remineralised to DOM) remains on centennial-to-millennial timescales before being eventually returned to the surface by upwelling, while a tiny fraction of mostly recalcitrant POM is buried in sediment and so sequestered on geological timescales.

The simplified representation of plankton ecology and the biological pump shown in Figure 1 forms the basis of many marine biogeochemical models, such as the one-size fixed-trait phyto- and zooplankton classes in the common NPZD (Nutrient-Phytoplankton-Zooplankton-Detritus) scheme (Friedrichs et al., 2007; Kwiatkowski et al., 2014). This approach misses many important biogeochemical processes though, prompting the development of 'dynamic green ocean models' which introduce multiple Plankton Functional Types (PFTs) with differentiated biogeochemical roles (Aumont et al., 2003; Quere et al., 2005).

However, this class of plankton model is still limited by a profusion of poorly constrained parameters based on limited observations, taxonomic overspecificity, and still-limited representation of biodiversity (Anderson, 2005; Boscolo-Galazzo et al., 2018; Friedrichs et al., 2007; Shimoda and Arhonditsis, 2016; Ward et al., 2018).

Although significant progress has been made since IPCC AR5 in optimising biogeochemical and ecological parameterisations in both NPZD and dynamic green ocean models using novel data assimilation and statistical techniques (Chien et al., 2020; Frants et al., 2016; Kriest, 2017; Kriest et al., 2020; Niemeyer et al., 2019; Sauerland et al., 2019; Schartau et al., 2017; Yao et al., 2019), neither approach fully accounts for allometric effects in biogeochemistry. Cell size distribution and elemental stoichiometry are dominant traits controlling plankton ecosystem function and total production (Finkel et al., 2010; Guidi et

al., 2009) and projections indicate that the fraction of large phytoplankton will increase with nutrient availability and decrease with warming (Mousing et al., 2014). Plankton size also has a substantial effect on POC export efficiency, with observations and models suggesting that although smaller plankton favour a greater proportion of POC being exported from the surface layer this POC dominated by small, slow sinking particles degrades more rapidly in the mesopelagic zone (Leung et al., 2021; Mouw et al., 2016b; Omand et al., 2020; Weber et al., 2016). Trait-based plankton models have been proposed to cover this

allometric gap, based on simulating generic ecosystem rules using key functional traits such as size rather than specific taxonomic identity, allowing ecosystem structure, biodiversity, and biogeography to emerge without being parameterised (Bruggeman and Kooijman, 2007; Follows et al., 2007; Harfoot et al., 2014; McGill et al., 2006). These ecosystem models still do not enable better understanding of Earth system feedbacks though because they have not been systematically incorporated into ESMs and so do not capture wider biogeochemical and large-scale physical dynamics.


Most biogeochemical models feature fixed phytoplankton stoichiometry, often following the canonical Redfield ratio for C:N:P of 106:16:1 or similar (Martiny et al., 2014; Redfield, 1934). However, real organisms can deviate substantially from this ratio, depending on cell size, functional group. and environmental conditions, with the Redfield ratio only emerging on a wider scale (Finkel et al., 2010). Climate change and ocean acidification are expected to substantially change ecosystem

composition and nutrient availability, while increasing temperatures and $CO_2$ concentrations have a direct impact on nutrient assimilation (Martiny et al., 2016; Moreno and Martiny, 2018; Riebesell et al., 2009). The C:P ratio has also been observed to increase with decreasing P availability as phytoplankton increased their P usage efficiency, which could help maintain production and therefore export despite expansion of low-nutrient oligotrophic zones ('oligotrophication') (Galbraith and Martiny, 2015). It is therefore likely that stoichiometry of POC may change in response to ocean warming and acidification,

with potential knock-on effects for the efficacy of the biological pump as a whole (Moreno et al., 2018). Despite this, flexible stoichiometry – with nutrient uptake by phytoplankton depending on current availability and their current cell quota – is rarely incorporated in ocean biogeochemistry models (Ward et al., 2018).

Metabolic processes are also temperature-dependent, and so ocean temperature partly determines many marine biogeochemical
patterns (Hoppe et al., 2002; Laws et al., 2000; Regaudie-De-Gioux and Duarte, 2012). For every 10°C increase in temperature, photosynthesis in any location is expected to increase by up to 100% (represented by a Q10 factor of 1-2), while average community respiration is expected to increase by between 100 and 200% (Q10 = 2-3) (Bendtsen et al., 2015; Boscolo-Galazzo et al., 2018; Eppley, 1972; Pomeroy and Wiebe, 2001; Regaudie-de-Gioux and Duarte, 2012; Sarmento et al., 2010). If warming-induced increases in respiration rates rise faster than production rates, organic matter will be remineralised more
quickly, shoaling the remineralisation depth (the e-folding point at which ~63% of POC is remineralised) (Boscolo-Galazzo et al., 2018; John et al., 2014a; Kwon et al., 2009) and may also reduce transfer efficiency within the mesopelagic zone (Fakhraee et al., 2020; Weber et al., 2016). One might expect this to reduce carbon export overall as less carbon makes it out of the surface ocean, but increased remineralisation also allows more nutrients to be recycled back into the surface, potentially resulting in more regenerated production (Leung et al., 2021; Segschneider and Bendtsen, 2013; Taucher and Oschlies, 2011).
Even a small shift in the remineralisation depth could have a significant potential impact on atmospheric $CO_2$, potentially acting as a positive climate feedback mechanism. For example, a global deepening of 24m of the e-folding depth (for example as a result of cooling) reduced $CO_2$ by 10-27 ppm in one model (Kwon et al., 2009). Although the biological pump itself does not act as a carbon sink when in long-term equilibrium (as exported carbon is returned to the surface by upwelling on millennial timescales), a change in biological pump strength would create a transient carbon sink if it enables accumulation of carbon in
the deep ocean before a new equilibrium state being reached (Goodwin et al., 2008).

Other processes that affect the biological pump and remineralisation will also be impacted by climate change. Ocean stratification is projected to increase, as surface warming increases the temperature gradient (Ciais et al., 2013; Riebesell et al., 2009). This reduces the nutrient flux from deep to surface waters, potentially leading to oligotrophication in lower latitude
surface waters (Bopp et al., 2005; Sarmiento et al., 2004). Oligotrophication leads to lower overall productivity in productive regions, but warming will not substantially affect productivity in existing oligotrophic regions where production is already limited (Richardson and Bendtsen, 2017). Oligotrophic regions may also be more productive than expected due to continued sub-surface production in deep chlorophyll maxima, but most ESMs do not resolve this phenomenon (Richardson and Bendtsen, 2019). The reduction in nutrient supply may also favour smaller plankton that can better cope with warmer and
oligotrophic conditions, resulting in a shift in ecosystem dynamics and function (Beaugrand et al., 2010; Bopp et al., 2005; Finkel et al., 2010). Reduced mixing rates along with surface warming also results in ocean interior deoxygenation, leading to an expansion of oxygen minimum zones, reduced nitrogen availability due to increasing denitrification, and increased phosphate release from affected sediments (Ciais et al., 2013; Keeling et al., 2010; Stramma et al., 2008).

The organic biological pump may also be affected by ocean acidification through shifting ecosystem composition, altered nutrient availability, and stoichiometric effects (Ciais et al., 2013; Nagelkerken and Connell, 2015; Riebesell et al., 2009; Tagliabue et al., 2011). Acidification may increase the C:N uptake ratio and decrease the N:P uptake ratio, potentially making

production more efficient (Riebesell et al., 2007; Tagliabue et al., 2011). Acidification could also lead to reduced particle ballasting – the hypothesised process by which denser falling PIC protects associated POC and so increases POC export – by reducing the supply of PIC and therefore reducing the efficiency of POC export (Armstrong et al., 2001; Klaas and Archer, 2002). However, the overall effect of ocean acidification feedbacks remains uncertain (Doney et al., 2020), and many of these processes are not resolved by ESMs. Furthermore, the human-driven loss of organisms higher up the food chain as a result of overharvesting and habitat degradation has a considerable yet poorly quantified effect on the biological pump (Pershing et al., 2010). Many of these factors influence and/or are influenced by both the magnitude of primary production and the remineralisation depth.

**[Table 1]**

Despite these known influences on the biological pump, many of the ESMs used for the IPCC AR5's ocean carbon sink projections incorporated few if any of these biogeochemical processes (Ciais et al., 2013; Schwinger et al., 2014). One study (Segschneider and Bendtsen, 2013) quantified the impact of including TDR, modifying the CMIP5 model MPI-ESM and its marine biogeochemistry model HAMOCC5.2, and projected an ~18 PgC reduction in ocean carbon uptake by 2100 under high emission scenario RCP8.5. However, only one out of ten CMIP5 ESMs featured non-fixed POC remineralisation profiles by enabling TDR (CanESM2) (Table 1), with most instead prescribing a fixed attenuating remineralisation profile with vertical POC flux following modern ocean observations (sometimes called the 'Martin Curve' (Bendtsen et al., 2015; Dunne et al., 2007; Martin et al., 1987)). Additionally, NPZD-type models cannot fully resolve the potential impact of climate change or ocean acidification on ecosystem structure, biodiversity, and plankton size shifts as they do not resolve allometric or stoichiometric effects. Only four of the ten CMIP5 ESMs featured multiple PFTs with different ecosystem functions beyond a simple NPZD scheme. Of these, only three account for plankton size in some way, and only three featured at least partially flexible stoichiometry (e.g. nutrient quotas and optimal allocation) that allow potential changes in nutrient utilisation in response to changing environmental conditions to be resolved (Kwiatkowski et al., 2018; Moreno and Martiny, 2018).

The next generation of CMIP6 ESMs for IPCC AR6 are currently in the process of completion, so insufficient results are available for use as comparison in this study. These models show some improvements in these regards, with five models reporting an increase in the number of explicit or implicit PFT or bacteria classes, three models introducing more variable stoichiometry (although one model has instead reduced flexible stoichiometry), and two models introducing more than one sinking POC classes (Séférian et al., 2020). Despite these improvements, the CMIP6 models still only feature broad size classes rather than a full spectrum of plankton size classes, only three have fully flexible stoichiometry, and most still use a fixed remineralisation profile for exported POC. Investigating changes in the biological pump in response to the physical and chemical perturbations of climate change and ocean acidification therefore requires an ESM with more complex biogeochemistry and ecosystem dynamics than present in these ESMs.

## 3. Methods

### 3.1. The cGEnIE model

ecoGEnIE is an extension of cGEnIE – the carbon-centric Grid Enabled Integrated Earth system model, an EMIC based on a modular framework efficiently resolving ocean circulation, biogeochemistry, and optional deep-sea sediment that has been simplified to focus on long-term carbon cycle (Ridgwell et al., 2007; Ridgwell and Schmidt, 2010). cGEnIE has been used in many previous studies of climate-carbon cycle interactions in both modern (Tagliabue et al., 2016) and palaeo applications (Gibbs et al., 2016; John et al., 2014a; Meyer et al., 2016; Monteiro et al., 2012; Norris et al., 2013; Ridgwell and Schmidt, 2010). EMICs such as cGEnIE have lower spatiotemporal resolution than more comprehensive ESMs based on atmosphere-ocean general circulation models and so are limited in their physical realism, but they are also less computationally expensive and thus well-suited for investigating more complex biogeochemical dynamics and performing efficient simulations of longer timescales or multiple scenarios (Claussen et al., 2002; Ward et al., 2018).

cGEnIE's climate model (C-GOLDSTEIN) features 3D reduced physics (frictional geostrophic, non-eddy resolving) ocean circulation model coupled to a 2D energy–moisture balance model of the atmosphere and a dynamic–thermodynamic sea-ice model (Edwards and Marsh, 2005; Marsh et al., 2011). C-GOLDSTEIN is configured on a 36 x 36 equal area horizontal grid (each cell being 10° in longitude and varying from ~3.2° to 19.2° in latitude), has 16 logarithmically-spaced vertical layers, and 96 time steps per year. The horizontal and vertical transport of heat, salinity, and biogeochemical tracers is calculated via a combined parameterisation for isoneutral diffusion and eddy-induced advection, and features a surface mixed layer scheme (based on the seasonal thermocline model of Kraus and Turner (1967)). cGEnIE also features a comprehensive ocean biogeochemistry module (BIOGEM) with phosphorus (in the form of phosphate, $PO_4$) and iron as the co-limiting nutrients (Ridgwell et al., 2007; Ridgwell and Schmidt, 2010; Ward et al., 2018). Organic matter production and export is parameterised in BIOGEM as a function of nutrient availability and following a fixed dissolved to particulate organic matter (DOM:POM) ratio, while $CaCO_3$ production and export is parameterised by a saturation state-dependent particulate inorganic to organic carbon (PIC:POC) rain ratio. BIOGEM by default uses fixed remineralisation profiles similar to the Martin curve for the sinking labile fractions of both POC and PIC (Martin et al., 1987; Ridgwell et al., 2007), but includes an optional temperature-dependent remineralisation scheme which has previously been used to explore the biological pump in warm palaeo oceans (John et al., 2014b). An updated calibration of this scheme which also couples TDR with temperature-dependent export production was also recently developed (Crichton et al., 2021) and is the version (*cGEnIE.muffin* v0.9.13) used in this paper.

## 3.2. The ecoGEnIE extension

The current cGEnIE version (*cGEnIE.muffin*) has recently been extended to ecoGEnIE (v.1.0) by incorporating a new scheme for plankton ecology (ECOGEM), replacing cGEnIE's implicit, flux-based parameterisation biogeochemistry module BIOGEM with an explicitly resolved and temperature-sensitive trait-based ecosystem module (Ward et al., 2018). In contrast to BIOGEM, biomass is now explicitly resolved, with each plankton population subject to ecophysiological processes including nutrient uptake (subject to quota saturation), photosynthesis and oxygen production (subject to light limitation, photoacclimation, and seasonal light attenuation within a variable mixed layer depth), predation (subject to prey-switching, prey refugia, and prey assimilation), and mortality. Many of these processes are temperature-sensitive (nutrient uptake, photosynthesis, and predation) or size-dependent (maximum photosynthetic and nutrient uptake rates, nutrient affinities, cell carbon quotas, maximum prey ingestion rates, and DOM fraction). In this configuration of ecoGEnIE (v.1.0) there are two plankton functional types (PFTs) available: phytoplankton (with nutrient uptake and photosynthetic traits enabled) and zooplankton (with predation traits enabled), with further classes such as calcifiers and silicifiers to be made available in future. As calcifiers are not explicitly represented, $CaCO_3$ production and export is still controlled by a saturation state-dependent ratio to POC export. Explicitly resolving biomass also allows introduces a lag between environmental forcing and ecosystem response, allowing seasonal cycles and transient behaviour in POC production and export to emerge (Galbraith et al., 2015).

As size is the dominant trait controlling plankton biogeochemical function and response to warming (Finkel et al., 2010; Mousing et al., 2014) each PFT is further split into 8 size classes ranging from 0.6μm to 1900μm. Zooplankton graze on all potential prey subject to availability with an optimum predator:prey length ratio of 10. This allows a better resolution of biodiversity within the model relative to models without size classes, with the ecosystem capable of shifting to a different structure in response to environmental forcing. ECOGEM also includes flexible stoichiometry rather than being fixed to the canonical Redfield ratio, allowing dynamic usage of nutrients in response to warming, ocean acidification, and nutrient availability to also be resolved (Boscolo-Galazzo et al., 2018; Martiny et al., 2016; Moreno and Martiny, 2018). DOM production is explicit in ECOGEM and so allows a variable and plankton size-dependent POM:DOM ratio, variations in which may have a significant impact on primary production in oligotrophic regions (Richardson and Bendtsen, 2017) and would result in reduced POM export with a shift to smaller plankton classes.

Although using an EMIC such as cGEnIE/ecoGEnIE allows for greater ecological resolution, it introduces different limitations. cGEnIE/ecoGEnIE has coarse spatial (36 x 36 equal area horizontal grid and 16 ocean layers) and temporal resolution (every ~4 days for C-GOLDSTEIN, every ~8 days for BIOGEM, and every ~0.4 days for ECOGEM), and so is not able to fully resolve spatial circulation and ecological patterns, vertical POC distribution, or the dynamics that potentially link stratification and deep chlorophyll maxima in oligotrophic regions (Richardson and Bendtsen, 2017, 2019). Subtle differences in spatial resolution and physical framework representations can have a substantial impact on circulation patterns, which could affect

plankton community structure and the residence time of exported nutrients and carbon (Pasquier and Holzer, 2016; Sinha et al., 2010). Currently only two PFTs are available in ecoGEnIE (phytoplankton and zooplankton, with PIC export set as

saturation state-dependent ratio of POC), limiting the extent to which hard pump dynamics involving calcifiers and silicifiers can emerge in our results. ecoGEnIE has not yet been fully recalibrated to the modern ocean and does not perform quite as well against observational data for key biogeochemical tracers (DIC, ALK; $PO_4$, $O_2$) as cGEnIE (Ward et al., 2018), but the results are still broadly similar (reproducing approximately 90% of the global variability in DIC, more than 70% for $PO_4$, $O_2$, and ALK, and more than 50% for surface chlorophyll, and broadly captures vertical distributions of these tracers). In this study

we focus primarily on the global biological pump response rather than its spatial patterns, and are also particularly concerned with surface DIC and its relation to ocean carbon sink dynamics, and so this configuration is sufficient for this global analysis.

### 3.3.    Experimental setup

We assess the differing impacts of replacing cGEnIE's Fixed Profile Remineralisation (FPR) parameterisation with its Temperature-Dependent Remineralisation (TDR) scheme (John et al., 2014b) and replacing cGEnIE's original parameterised

biogeochemistry BIOGEM module (BIO) with ecoGEnIE's trait-based ECOGEM module (ECO) (Ward et al., 2018). We test each new element both separately and in combination, analysing four cGEnIE/ecoGEnIE configurations:

- **BIO+FPR** is cGEnIE with the default BIOGEM module (BIO) and the default Fixed Profile Remineralisation scheme (FPR)

- **BIO+TDR** is cGEnIE with the default BIOGEM module (BIO) and the alternative Temperature-Dependent Remineralisation scheme (TDR)

- **ECO+FPR** is ecoGEnIE, incorporating the trait-based ECOGEM module with flexible stoichiometry (ECO), and the default Fixed Profile Remineralisation scheme (FPR)

- **ECO+TDR** is ecoGEnIE (ECO) and the alternative Temperature-Dependent Remineralisation scheme (TDR)

We use the global POC export flux (PgCy$^{-1}$) from the surface layer (fixed in cGEnIE/ecoGEnIE as the top 80.8m of the ocean, compared with ~100m in some studies (Martin et al., 1987)) as our measure of biological pump strength and compare cumulative changes up to the year 2100 CE, and also quantify cumulative changes in the ocean carbon sink for each configuration through the air-to-sea $CO_2$ flux. We calculate cumulative changes in biological pump and ocean carbon sink

capacity for the policy-relevant timescale of the 21$^{st}$ century CE (Table 2), but results are also shown up to 2500 CE (Figure 2).

Each configuration is run under its default published calibration (or a combination of published parameters for ECO+TDR) as well as configurations recalibrated to result in the same preindustrial global biological pump strength (POC export of ~7.5 PgCy$^{-1}$ and PIC export of ~1 PgCy$^{-1}$) and similar global mean total Dissolved Inorganic Carbon (DIC), Alkalinity (ALK), and surface DIC speciation relative to the BIO+FPR and observational data (see Supplementary Table S1 & Figures S1-S45). The configurations were recalibrated to have as similar a carbon cycle as possible in order to make the results easily comparable across the configurations, while POC export was chosen as the primary calibration constraint as the main variable being analysed. However, some differences remain between the recalibrated configurations as well as with the observational data. The main difference is a higher POC and PIC sedimentation rates in the recalibrated ECO configurations as a result of specifying a higher recalcitrant POC fraction (from ~5% to ~32-35%) and higher PIC:POC ratio (from 4.85% to ~5%). This recalcitrant POC fraction and the resulting rain rate is unrealistically high compared to observations, but was necessary in order to respectively counter much higher POC export and lower PIC export in ecoGEnIE. In cGEnIE recalcitrant POC remains inert until sedimentation and so does not directly interact with the rest of the carbon cycle, and in sediment module-disabled configurations of cGEnIE POC and PIC rain is returned as deep ocean DIC, ALK, and nutrients upon reaching the sea floor meaning total ocean DIC is still conserved. Although biological pump perturbations on sub-overturning timescales (<500-1000y) will not significantly affect surface DIC via upwelled deep water within that time, a higher recalcitrant fraction would increase the average lifetime of regenerated DIC and nutrients in the ocean and gradually reduce the nutrient supply to the surface from intermediate waters. Optimising for equivalent POC export also leads to surface carbonate concentration ($[CO_3]$) being reduced in the BIO+TDR recalibration compared with the default calibration, leading to a reduced carbonate buffer for the ocean carbon sink in these runs. In this paper we focus on the results of default calibrations, but in order to explore the mechanisms driving differences between the configuration responses and to constrain the impact of differing biological pump baselines we also present the recalibration results in the Supplementary Material and discuss the differences in our Results.

Each model configuration is spun-up for 10,000 years and restarted at 0 CE (10000 Holocene Era, HE), and then forced in emissions mode from 1765 CE with combined historical and future CMIP5 RCP total $CO_2$ emission scenarios (3PD, 4.5, 6.0, and 8.5, corresponding to low, moderate, high, and severe emission scenarios respectively; 3PD used instead of RCP2.6 to allow for long-term simulation beyond 2100 CE) extended through to 2500 CE in order to assess multi-centennial dynamics (Meinshausen et al., 2011).

## 4.  Results

### 4.1.  Physical Climate Response

In its default configuration (BIO+FPR) cGEnIE projects surface air temperature warming of 1.8°C, 2.6°C, 3.2°C, and 4.2°C by 2100 relative to 1850-1900 in RCPs 3PD, 4.5, 6.0, and 8.5 respectively, which compares favourably with CMIP5 projections for these scenarios (1.6°C±0.4°C, 2.4°C±0.5°C, 2.8°C±0.5°C, and 4.3°C±0.7°C) (Collins et al., 2013). In the ocean, cGEnIE-BIO+FPR projects sea surface warming of 1.2°C, 1,8°C, 2.1°C and 2.8°C by 2100 in RCPs 3PD, 4.5, 6.0, and 8.5 respectively (see Supplementary Figure S46 for spatial patterns of warming in RCP4.5). This can be compared to 0.6°C and 2.0°C warming in the top 100m in CMIP5 for RCPs 2.6 and 8.5, with the apparent bias towards greater warming in cGEnIE reflecting the narrower surface layer (80.8m) versus the CMIP5 assessment. For baselines, cGEnIE's (BIO+FPR) preindustrial surface air temperature (SAT) global baseline (1850-1900) is ~12.5°C and the preindustrial sea surface temperature (SST) global baseline is ~18.8°C, both of which also lie within the CMIP5 range (SAT towards the lower end, SST in the higher end) (see Supplementary Figure S47 for spatial patterns of preindustrial warmth in RCP4.5). The model's circulation response are almost identical across the experiments, with only marginal differences in warming (<~0.01°C differences in ocean temperature for RCP4.5) between the scenarios across the configurations.

### 4.2.  Biological Pump Strength

[Table 2]

Our results show that the biological pump weakens by 2100 CE under most scenarios and configurations, but adding TDR and trait-based plankton ecology with flexible stoichiometry has strong and opposite impacts on relative biological pump strength.

[Figure 2]

Under the default cGEnIE configuration (BIO+FPR) anthropogenic climate change results in an overall weakening of the biological pump, with global POC flux falling below preindustrial by 2100 CE by ~6.1%  under RCP4.5 and ~9.8% under RCP8.5 (Figure 2; Table 2; Supplementary Figure S48). This is in line with past projections of a 7.2% decline in surface POC export under SRES A2 (warming levels between RCPs 6.0 and 8.5) during the 21[st] century in a EMIC with an NPZD biogeochemistry module (Taucher and Oschlies, 2011), and a selection of CMIP5 ESMs declining by between ~9 and ~20% under RCP8.5 during the 21[st] century with greater declines in models resolving dynamic plankton size classes and in the lower latitudes (Bopp et al., 2013; Cabré et al., 2015; Fu et al., 2016). In cGEnIE this is primarily driven by stratification resulting

in reduced surface nutrient concentrations and decreased primary production in high-productivity low and mid-latitude waters

(Figure 3a) in line with previous model results (Bopp et al., 2005; Ciais et al., 2013; Crichton et al., 2021; Riebesell et al., 2009; Sarmiento et al., 2004), along with reduced nutrient supply to the North Atlantic and Eastern Arctic due to overturning circulation slowdown. In contrast, there is an increase in production in high-latitude waters, where the mixed layer is already so much deeper than cGEnIE's surface layer (mostly >>100m, versus cGEnIE's ~81m surface layer; Supplementary Figure S49) that stratification actually increases productivity by more effectively confining nutrients within cGEnIE's surface layer.

This partially matches theoretical expectations in which stratification drives increased polar productivity by confining phytoplankton within the euphotic zone (Riebesell et al., 2009), but the mechanism driving this effect in cGEnIE is different as plankton are confined to the surface layer.

**[Figure 3]**


Adding TDR (BIO+TDR) leads to a substantially different result than the default cGEnIE configuration with a far smaller biological pump weakening that eventually reverses by the late 21$^{st}$ or early 22$^{nd}$ century, resulting in an overall ~0.6% increase in POC export under RCP4.5 and a ~0.8% decline under RCP8.5 by 2100 CE (Figure 2). This might be expected to be because TDR results in an initial decrease in biological pump strength with warming as more POC is remineralised within the surface

layer, which also leads to a shallower remineralisation depth and an increase in nutrient recycling and regenerated production in the surface layer. However, in this model POC remineralisation only occurs below the surface layer, and so regenerated production is not directly represented. Furthermore, observations show that while an increase in nutrient recycling within the surface layer can lead to an increase in production by reducing nutrient loss, it does not directly lead to an increase in export as well as it is the reduction in export driving the increase in production. Only a new allochthonous source of nutrients to the

surface layer would allow sustained increases in both production and export (Dugdale and Goering, 1967; Laws, 1991; Laws et al., 2000).

In our results the shoaling of the remineralisation depth increases PO$_4$ concentrations in the layers below the productive surface (cGEnIE layers 2-3, ~81-283m) from remineralisation that would otherwise have occurred deeper in intermediate waters

(Supplementary Figure S50). This in turn leads to increased allochthonous PO$_4$ input to the surface layer through mixing, which is sufficient to lead to an elevated baseline in new production and POC export in warmer lower-latitude waters (and conversely lower baseline POC export in cooler mid latitudes; Figure 4a) and with warming counters the decline in POC export in the low and mid-latitudes observed in BIO+FPR (Figure 3b). This result is consistent with previous modelling, which has shown that shoaling of the remineralisation depth in a common biogeochemical model leads to increased POC export (Kwon

et al., 2009), and that including TDR in an EMIC resulted in increased Net Primary Production and a marginally smaller decrease in POC export under RCP8.5 (Taucher and Oschlies, 2011). A recent update to cGEnIE's TDR scheme (Crichton et al., 2021) also found a similar result, with historical warming resulting in a ~0.3% decline in POC export with TDR activated

versus ~2.9% without. In contrast, in higher latitudes including TDR leads to a lower baseline POC export than with FPR (Figure 4a), as colder waters result in a deep remineralisation depth and less $PO_4$ returned to the surface layer.


**[Figure 4]**

Activating ecoGEnIE (ECO+FPR) instead of TDR results in a greater weakening of the biological pump than in BIO+FPR, with global POC flux falling by ~7.5% by 2100 CE under RCP4.5 and ~11.8% under RCP8.5 (Figure 2). Adding ECOGEM
allows an overall decrease in average plankton size in response to climate change (Supplementary Figure S51), as warming and stratification leads to oligotrophication in lower latitude waters which favours smaller plankton size classes, and is in line with previous observational and modelling studies (Finkel et al., 2010; Riebesell et al., 2009). Smaller taxa produce more DOM than POM (Finkel et al., 2010) and so the shift to smaller plankton classes in warmer regions decreases overall baseline POC and PIC export (Figure 4b), which accentuates the reduction in export due to stratification-induced nutrient and biomass
decline in the low and mid-latitudes (Figure 3c and Supplementary Figure S52). This decline is sufficient to counteract the negative feedback of the shift to smaller particles increasing surface nutrient recycling due to shallower remineralisation (Leung et al., 2021). Activating ECOGEM also enables flexible stoichiometry, but the effect of this is difficult to disentangle from that of multiple size classes as well. However, some patterns and trends can be seen. The preindustrial POM export C:P ratio lies above the standard Redfield ratio of 106:1 across most of the ocean outside the Southern Ocean, reaching ~200:1 in
equatorial upwelling regions and the global mean closely matching recent observations of 163:1 (Supplementary Figure S53) (Martiny et al., 2014). By 2100 CE this ratio increases across almost the entire ocean, especially along the Antarctic Polar Front and in the Arctic Ocean (Supplementary Figure S54). This indicates that the amount of carbon exported for every unit of phosphorus increases with warming in response to stratification, partly ameliorating the decline in carbon export.

Without stratification and nutrient restriction, higher equilibrium temperatures in a previous ecoGEnIE study were associated with higher export production and mean cell size despite lower overall biomass (Wilson et al., 2018). Although increased phytoplankton nutrient usage (which is temperature-dependent in ecoGEnIE) boosted small phytoplankton production in their study, this increase was assimilated by zooplankton grazing (which is also temperature-dependent). This allowed larger phytoplankton to compete against small phytoplankton with higher nutrient affinities, and resulted in increased particulate
export from larger phytoplankton and inefficient zooplankton feeding despite lower overall ecosystem biomass (Ward et al., 2014). When baseline nutrient fluxes were instead elevated without higher temperatures, increases in small phytoplankton biomass were again limited by zooplankton grazing and allowed larger phytoplankton more competitive, but unlike warming alone higher nutrient fluxes facilitated both elevated total ecosystem production and export. In our ecoGEnIE results though transient warming is accompanied by both stratification and reduced nutrient flux in lower latitudes, resulting in an overall
shift to smaller phytoplankton dominance despite warming allowing greater phytoplankton nutrient usage and grazing. This reduces total ecosystem biomass and therefore POC export in the lower latitudes, which the consequent reduction in large

phytoplankton abundance and their grazing further accentuates. Together this leads to a greater decline in POC export in ECO+FPR than in BIO+FPR.

In contrast to low-latitudes, in most high-latitude waters biomass increases while mean cell size and export decline (Figure 3c and Supplementary Figures S51, S52, & S55), and along the Antarctic Polar Front biomass decreases, mean cell size is stable or increases, and POC export increases. The latter is because warming in nutrient-rich upwelling regions allows for increased zooplankton and larger phytoplankton abundance (Supplementary Figures S56-61) and therefore leads to restrained total biomass due to grazing coupled with increased export. In non-upwelling polar regions such as the western Arctic where

nutrients are limited but unlike in low-latitudes warming-induced stratification does not restrict nutrient flux further, warming preferentially boosts smaller phytoplankton (6µm vs. 19µm) which along with a commensurate decline in dependent zooplankton (19µm) and top-down grazing pressure leads to increased overall biomass but lower export. In the eastern Arctic this process is not as apparent due to the interference of overturning circulation slowdown resulting in a moderate reduction in formerly elevated nutrient availability. This leads to a reduction in medium relative to small phytoplankton classes (19µm vs.

1.9 & 6µm) and a commensurate shift to smaller zooplankton classes (6 & 19µm vs. 60µm), and therefore relatively stable biomass and mean cell size coupled with reduced export.

Adding both trait-based plankton ecology and TDR (ECO+TDR) produces a complex result, with the weakening effect of adding ECO on the biological pump partly counteracting the strengthening effect of adding TDR. The overall effect is a

moderate reduction in POC flux by ~2.7% by 2100 CE under RCP4.5 and ~5.4% under RCP8.5 (Figure 2), as decreasing plankton size and POC export in lower latitude waters due to adding ECO reduces the capacity for nutrient recycling to increase as a result of adding TDR (Figure 4c). The combined effect of ECO+TDR relative to BIO+FPR in this model is therefore an additional ~0.1% reduction in the weakening of the biological pump by 2100 CE relative to preindustrial across the RCPs (Figure 3d), resulting in ~5 PgC more POC being exported by the biological pump in this model by 2100 CE. In all

configurations and scenarios the changes in the biological pump continue past 2100 CE, and in many cases only begin to stabilise after several hundred years (Figure 2).

Using recalibrated instead of the published default calibrations for each configuration results in the same overall pattern of TDR ameliorating and ECO amplifying the biological pump weakening with warming, but with a greater weakening for

ECO+FPR and ECO+TDR, reduced long-term strengthening for BIO+TDR, and a greater rather than smaller net weakening in ECO+TDR relative to BIO+FPR (-~1.4% vs. +~0.1%; Supplementary Figure S62). These recalibrations correct for the substantially different baseline biological pumps in the default calibrations, with baseline POC export of ~8.1 PgCy$^{-1}$ in BIO+TDR, ~11.3 PgCy$^{-1}$ in ECO+FPR, and ~11 PgCy$^{-1}$ in ECO+TDR (Figure 4, left). High production and export leads to differing initial ecosystem structure and therefore amplified effects on remineralisation when POC export changes with

warming, which acts as a confounding factor when comparing their responses. However, in order to make baseline POC export

equivalent in the ECO recalibrations it was necessary to substantially increase the recalcitrant fraction of POC export. This leads to reduced remineralisation in intermediate waters and gradually limits surface nutrient supply, and therefore is a factor in the amplified decline in POC export observed in our ECO recalibration results. The general pattern of our results though – activating TDR or ECO leading to a relative strengthening or weakening of the biological pump respectively, and activating
both leading to an overall weakening – is consistent across the different calibrations, indicating that these trends are robust.

### 4.3.      Ocean Carbon Sink Capacity

In previous discussions of empirical and model results it has been understood that a decrease in biological pump strength directly leads to a corresponding decrease in the ocean carbon sink capacity, as less POC is exported from the surface to deep ocean and so more $CO_2$ remains in surface waters and therefore the atmosphere (Boscolo-Galazzo et al., 2018; John et al.,
2014a; Olivarez Lyle and Lyle, 2006; Steffen et al., 2018). This is the case when comparing long-term equilibrium states, with for example warm palaeoclimate states with a stronger biological pump storing more carbon in the ocean. However, in transient scenarios such as today reduced POC export affects many other processes, which results in a nonlinear relation between biological pump strength and the ocean carbon sink capacity that can lead to counter-intuitive outcomes (Gnanadesikan and Marinov, 2008; Kwon et al., 2009).


[Figure 5]

In our simulations, the relative strengthening of the biological pump when TDR is included actually leads to a ~1.0% net decrease in the ocean carbon sink capacity by 2100 CE (Table 2, Figure 5). Conversely, the relative weakening of the biological
pump with ECOGEM activated instead (ECO+FPR) is associated with a ~0.2% net increase in the ocean carbon sink capacity. Combining both ECOGEM and TDR (ECO+TDR) results in a smaller overall relative weakening of the biological pump compared to default, and a ~0.7% net decrease in the ocean carbon sink capacity (~4.9 PgC under RCP4.5, ~6.2 PgC under RCP8.5) by 2100 CE (Table 2). The physical climate and circulation response is effectively identical across these different configurations, indicating that the differences are biogeochemically driven. Including trait-based ecology using size classes
therefore largely but not entirely offsets the impact on the ocean carbon sink of also including TDR in this model during the 21$^{st}$ century. The model thus suggests that ecological dynamics increases the resilience of plankton ecosystem functioning against the pressures of climate change.

A decrease in particulate export does not automatically result in a decrease in the ocean carbon sink capacity in this model as
a result of changing remineralisation depths and interactions with carbonate chemistry and ocean acidification. Adding TDR results in greater production of both POC and PIC relative to BIO+FPR in non-polar regions in response to warming shoaling

the remineralisation depth, as described in the Section 4.2. This has two effects in our results. Firstly, shallower remineralisation and increased POC export from remineralised nutrients results in an increase in respired $CO_2$ in surface waters relative to BIO+FPR. Secondly, increased $CaCO_3$ formation and surface nutrient remineralisation also results in overall lower

ALK in surface waters relative to BIO+FPR, which through DIC speciation leads to an overall decrease in the concentration of dissolved carbonate ($[CO_3]$) and decreased surface pH and carbonate saturation state ($\Omega$) (Supplementary Figure S63) (as theoretically described by Zeebe and Wolf-Gladrow (2001), and similar to the mechanisms described by Kwon et al. (2009)). Together with shallower remineralisation this increases the partial pressure of $CO_2$ in surface waters ($pCO_2$), therefore reducing the capacity for additional $CO_2$ to dissolve from the atmosphere into the ocean. This effect on the air-to-sea $CO_2$ flux gradually

limits the total DIC content for the whole ocean and therefore the ocean carbon sink as a whole (see explanatory schematic in Supplementary Figure S64). Ocean acidification also concurrently increases surface $pCO_2$ and decreases $\Omega$ and PIC production (Supplementary Figure S65), and so adding TDR tends to amplify ocean acidification by further increasing surface $pCO_2$ in response to warming. Conversely, as shown in Section 4.2 adding ECOGEM instead reduces total ecosystem biomass and POC export with warming relative to BIO+FPR as a result of the shift to smaller plankton taxa. This shift increases the

DOM:POM production ratio, which results in a greater reduction in POC export and subsurface remineralisation. $CaCO_3$ formation and PIC export is also reduced as in this version of ecoGEnIE $CaCO_3$ production is fixed as a saturation-state dependent ratio to POC export. Together this leads to lower DIC, higher ALK, increased $[CO_3]$ and $\Omega$ relative to BIO+FPR (Supplementary Figure S66), and therefore decreased $pCO_2$ in low and mid-latitude surface waters and increased air-to-sea $CO_2$ flux and total ocean DIC in the long-term. Introducing ECOGEM and the resultant oligotrophication-induced plankton

size shift therefore slightly counters the ocean acidification trend.

The same experiments were also repeated using recalibrated configurations, as the default calibrations have different baseline biological pumps and carbonate chemistry which act as confounding factors in their response. Compared to the recalibrations baseline POC production and export is higher and more resilient in the default TDR and ECO calibrations (Supplementary

Table S1), and therefore changes in POC export have a reduced impact on surface carbonate chemistry than if the export baseline was equivalent. In the ECO configurations $[CO_3]$ is also much lower (~70-80 vs, ~106 $\mu$mol kg$^{-1}$ in BIO+FPR) and $[CO_2]$ much higher (~40-50 vs. 24 $\mu$mol kg$^{-1}$ in BIO+FPR) than in the recalibrations, resulting in substantially weaker carbonate buffering in the default configurations. Using the recalibrations instead (Supplementary Figure S67) increases the long-term sink strengthening effect by ECO and reduces sink weakening by TDR relative to the default configurations, which

after the 21$^{st}$ century results in a net sink strengthening with ECO+TDR rather than a net weakening. However, the small differences in surface carbonate chemistry between the recalibrated configurations still have some confounding effects on our carbon sink results, as does the adjustment of POC export parameters. As discussed in Section 4.2, a higher recalcitrant POC fraction in the ECO recalibrations reduces remineralisation in intermediate waters and gradually reduces surface nutrient supply and productivity, which amplifies the carbon sink capacity reduction described above. Furthermore, the BIO+TDR

recalibration has a ~7.5% lower baseline $[CO_3]$ than BIO+FPR (Supplementary Table S1), which as discussed in Section 3.3

somewhat reduces carbonate buffering and so could explain a proportion of the simulated carbon sink weakening through a reduced solubility pump. Both higher and lower [$CO_3$] in the default TDR and ECO calibrations respectively are associated with reduced carbon sink capacity relative to the recalibrations though, while in the recalibrated configurations ECO+FPR shows an increase in carbon sink capacity despite lower [$CO_3$] than ECO+TDR. Together this indicates that [$CO_3$] has a
relatively minor impact on the sign and magnitude of our carbon sink results, and although the POC recalcitrant fraction recalibration does affect our results more strongly the broad trends of relative sink weakening with TDR and relative sink strengthening with ECO are robust.

## 5.      Discussion

These results clearly illustrate the importance of incorporating multiple dimensions of ecological complexity within Earth
system models in order to capture the impact of nonlinear climate-biosphere feedbacks, biodiversity, and ecological resilience on the future dynamics of carbon sinks. However, although the introduction of either TDR or ECO leads to opposing trends in POC export in response to warming, the overall impact on the ocean carbon sink is relatively modest. Our cGEnIE experiments simulate a decline in the ocean carbon sink capacity of around ~6.5 PgC (~0.06 PgCy$^{-1}$) during the 21st century under an RCP8.5 scenario when accounting for TDR. This can be compared to a previous estimate of a ~18 PgC (~0.18 PgCy$^{-1}$) decline
in ocean carbon sink capacity by 2100 CE in response to RCP8.5 made using a simpler NPZD-based ecosystem representation that differentiated silicifying plankton (Segschneider and Bendtsen, 2013), and to the 2018 ocean carbon sink uptake rate of 2.6±0.6 PgCy$^{-1}$ (Friedlingstein et al., 2019). This decline is partially countered when greater ecological complexity and flexible stoichiometry is introduced as well, with a shift to smaller plankton classes in response to oligotrophication leading to a 21$^{st}$ century ocean carbon sink reduction of ~5.9 PgC. Other processes that are not resolved in this configuration of ecoGEnIE
could also substantially affect the biological pump, such as ballasting, calcifier-silicifier trade-offs, nitrogen cycle and stoichiometry-acidification feedbacks (Buchanan et al., 2019; Dutkiewicz et al., 2015; Landolfi et al., 2017; Riebesell et al., 2007; Somes et al., 2016; Tagliabue et al., 2011), deep chlorophyll maxima, and on longer timescales redox-dependent feedbacks (Niemeyer et al., 2017; Watson, 2016). Limited physical resolution can have significant impacts on biogeochemistry (Sinha et al., 2010), and so also limits our results. Further work is required to assess the impact of these features on our
estimates.

Few of the ESMs used in CMIP5 sufficiently resolve marine ecology, instead relying on simple plankton ecosystems that are often highly parameterised with minimal or non-existent ecological and metabolic dynamics (Table 1). This reduces computational expense and so allows higher resolution of important physical processes, but comes at the price of poorly
resolving known biogeochemical and ecological feedbacks that can substantially affect carbon partitioning (Anderson, 2005; Ward et al., 2018). To date, gains in computational power have largely been allocated to improved resolution and physical process representation, while despite recent progress biogeochemical parameters have remained too poorly constrained to

allow greater biogeochemical complexity in high resolution ESMs. However, the development of trait-based ecological models could enable ESMs to include more complex marine biogeochemical modules without compromising the high resolution representation of physical processes. An approach that focuses on functional traits and generic ecosystem rules potentially reduces the need for taxonomic-specific parameterisations and also allows better representation of allometric effects. Development of biogeochemical models with higher physical resolution would also allow more accurate representation of fine-scale biogeochemical processes such as the interaction of stratification, the nutricline, and deep chlorophyll maxima in oligotrophic regions (Richardson and Bendtsen, 2017, 2019), issues raised in this study that have not been possible to explore. EMICs with lower physical resolution can more readily incorporate ecological complexity though, and remain a crucial tool for further exploring these feedbacks in the interim (Chien et al., 2020; Frants et al., 2016; Kriest, 2017; Kriest et al., 2020; Niemeyer et al., 2019; Sauerland et al., 2019; Schartau et al., 2017; Ward et al., 2018; Wilson et al., 2018; Yao et al., 2019).

In this study we focus on the dominant soft-tissue biological pump, but the variable response of plankton classes with different shell types to climate change and ocean acidification also has an impact on the biological pump. For instance, silicifiers with opal-based shells such as diatoms thrive in nutrient-rich waters. Segschneider and Bendtsen (2013) found that the increased nutrient recycling when TDR was introduced in their model initially drives an increase in diatom production and opal export in response to climate change. In their model, this soon leads to silicate-depleted surface waters and suppressed diatom production, allowing a subsequent increase in calcifying plankton and PIC export instead. This has the effect of reducing surface alkalinity and increasing surface $pCO_2$, which drives a substantial proportion of the large ocean carbon sink reduction in their analysis. Despite the likely importance of this 'hard-shell' mechanism, ecoGEnIE does not currently allow independent representation of calcifiers and does not represent silicifiers at all, and so the potential impact of this mechanism is not resolved by our results. However, the model of Segschneider and Bendtsen (2013) does not feature size classes or flexible stoichiometry, which we have shown is important for determining the soft-tissue biological pump response. In order to fully compare our results it will be necessary to repeat these simulations with the silicifier-enabled ECOGEM currently under development. Together, resolving plankton size classes, TDR, flexible stoichiometry, and separate silicifier and calcifier functional types will allow the response of the marine biological pump to climate change to be more fully diagnosed.

Further development will also allow the potential impact of ballasting to be assessed. Using a different EMIC, Kvale et al. (2015; 2019) found that adding ballasting alongside calcifier functional types mitigated the biological pump response to ocean warming by facilitating increased calcifier production and therefore increasing nutrient export from the surface. In contrast, activating ballasting in ecoGEnIE without separating out a competitive calcifier functional type would likely result in greater surface waters remineralisation in scenarios with reduced PIC production. However, empirical observations have suggested that the ballasting effect on the ocean carbon sink is weaker than has been hypothesised (Wilson et al., 2012).

## 6.    Conclusions

The response of the biological pump to climate change is important for projecting climate feedbacks and the future behaviour of the ocean carbon sink, but many of the most influential Earth system models fail to incorporate sufficient metabolic or ecological complexity for this to be fully resolved. In this study, we have investigated the impact of integrating temperature-dependent remineralisation, size-based biodiversity, and flexible nutrient usage on the biological pump and ocean carbon sink in response to climate change. As expected, we found that adding temperature-dependent remineralisation to an Earth system model of intermediate complexity (ecoGEnIE) results in a greater weakening of the ocean carbon sink as a result of climate change. However, this actually results from a relative strengthening of the biological pump itself as a result of shallower nutrient remineralisation, contrary to the common expectation that the direct effect of warming further amplifies a weakening of the biological pump. Conversely, adding trait-based ecosystem dynamics instead results in an even weaker biological pump as a result of oligotrophication favouring smaller plankton, and in turn a larger ocean carbon sink. Finally, combining both of these features results in a small relative strengthening of the biological pump and a modest reduction in the ocean carbon sink capacity relative to default simulations.

Together, this implies that the biological pump positive feedback on climate change may be larger than CMIP5 models project, but is potentially less than some more recent model projections (Segschneider and Bendtsen, 2013; Steffen et al., 2018). This study has primarily focused on the allometric aspects of dominant soft-tissue components of the biological pump, and the results clearly illustrate the substantial degree to which ecological dynamics and biodiversity can modulate the strength of climate-biosphere feedbacks. These complex relations require further analyses and validation, but at present comparison of model studies is a challenge because today's ESMs take such different approaches and simplifications. Trait-based ecological modules that go beyond simple biogeochemical traits could in future enable ESMs to include more ecological complexity without compromising the high resolution representation of physical processes, and allow feedbacks such as the marine biological pump to be more fully resolved.

## Author Contributions

DIAM, SEC, & KR conceived of the study; DIAM designed the study, configured and ran the model, and performed the analyses; DIAM wrote the paper with input from SEC, KR, & JR.

## Data availability

*cGEnIE.muffin* is available for download from https://github.com/derpycode/cgenie.muffin, and a manual detailing code installation, model configuration, and extensive tutorials is available from https://github.com/derpycode/muffindoc. Modern

observational data for model-data comparison are available from http://www.seao2.info/mymuffin.html. $CO_2$ emission

scenarios for forcing the model are available from http://www.pik-potsdam.de/~mmalte/rcps/ (Meinshausen et al., 2011).

**Conflicts of Interest**

The authors declare that they have no conflict of interest

**Acknowledgements**

This work was supported by the European Research Council Advanced Investigator project "Earth Resilience in the

Anthropocene" ERA (ERC-2016-ADG-743080) and a core grant to Stockholm Resilience Centre by Mistra. We thank Andy Ridgwell and Ben Ward for discussions on using ecoGEnIE, and Toby Tyrrell for discussions on variable stoichiometry.

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

 **Figures and Tables**

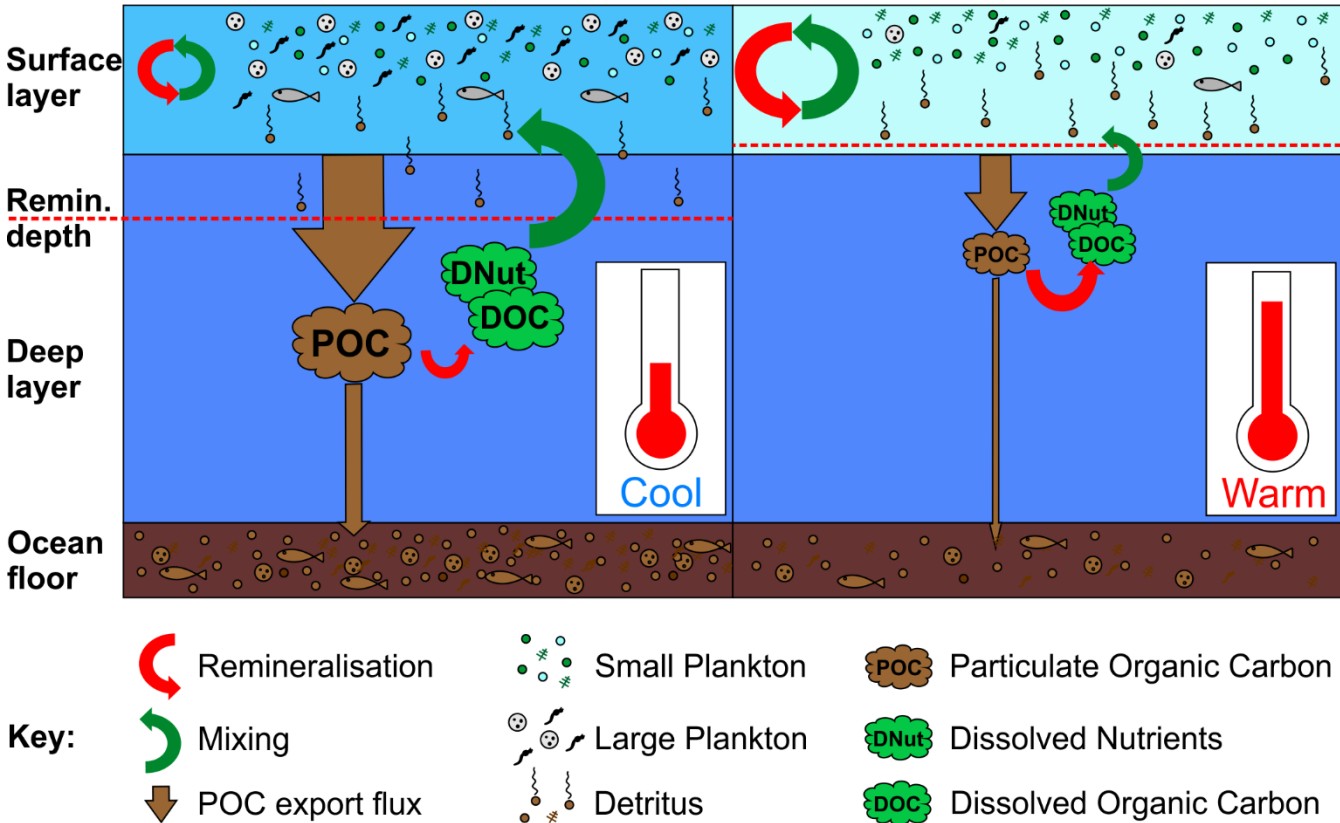

**Key:**
| | | | |
|---|---|---|---|
| ↻ Remineralisation | Small Plankton | POC Particulate Organic Carbon |
| Mixing | Large Plankton | DNut Dissolved Nutrients |
| POC export flux | Detritus | DOC Dissolved Organic Carbon |

**Figure 1: Schematic illustrating the impact of warming on the soft tissue biological pump.** On the left-side, under cooler preindustrial conditions cGEnIE's surface layer remains fairly well mixed with the deep ocean (green arrow), returning dissolved nutrients and carbon (DNut & DOC) from the remineralisation of exported POC (red arrow), while some POC is remineralised partly within the surface layer
(surface red arrows). On the right-side, warming leads to a shift to dominance by smaller plankton as well as stratification leading to less mixing between the shallow and deep ocean, while shoaling of the remineralisation depth leads to greater recycling of nutrients and carbon close to the surface layer, combining to result in an overall reduction in POC export and sedimentation and an overall increase in the residence time of nutrients and carbon in the ocean.

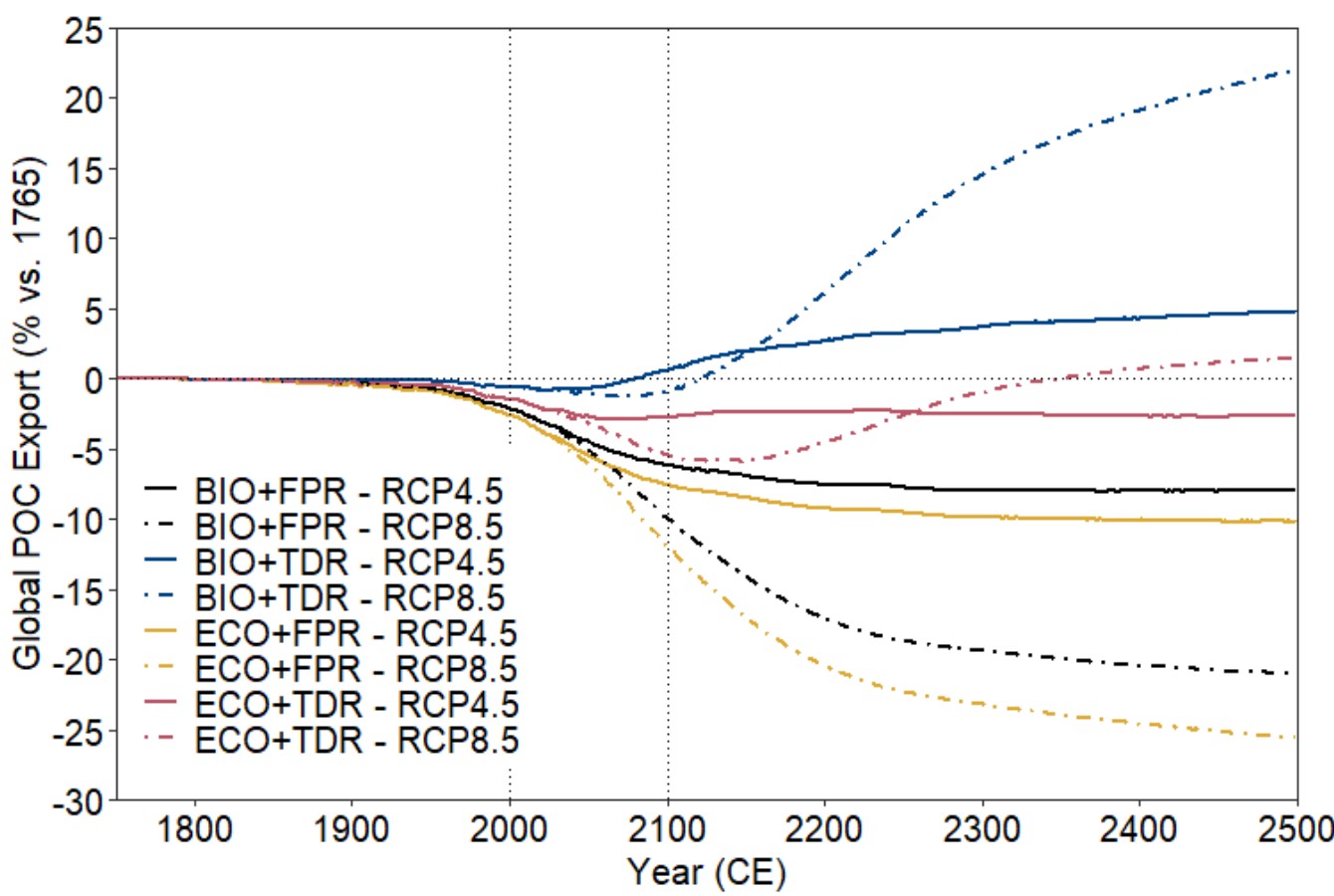

**Figure 2: cGEnIE/ecoGEnIE simulation results for global POC export flux under different configurations and forcing scenarios.**
Results for RCP4.5 (solid lines) and RCP8.5 (dot-dashed lines) are shown for each of the configurations (BIO+FPR – black; BIO+TDR –
blue; ECO+FPR – yellow; ECO+TDR – red), and the baseline POC export and the 21st century marked by the horizontal and vertical dotted
lines respectively. Results for all emission scenarios are shown in Supplementary Figure S48.

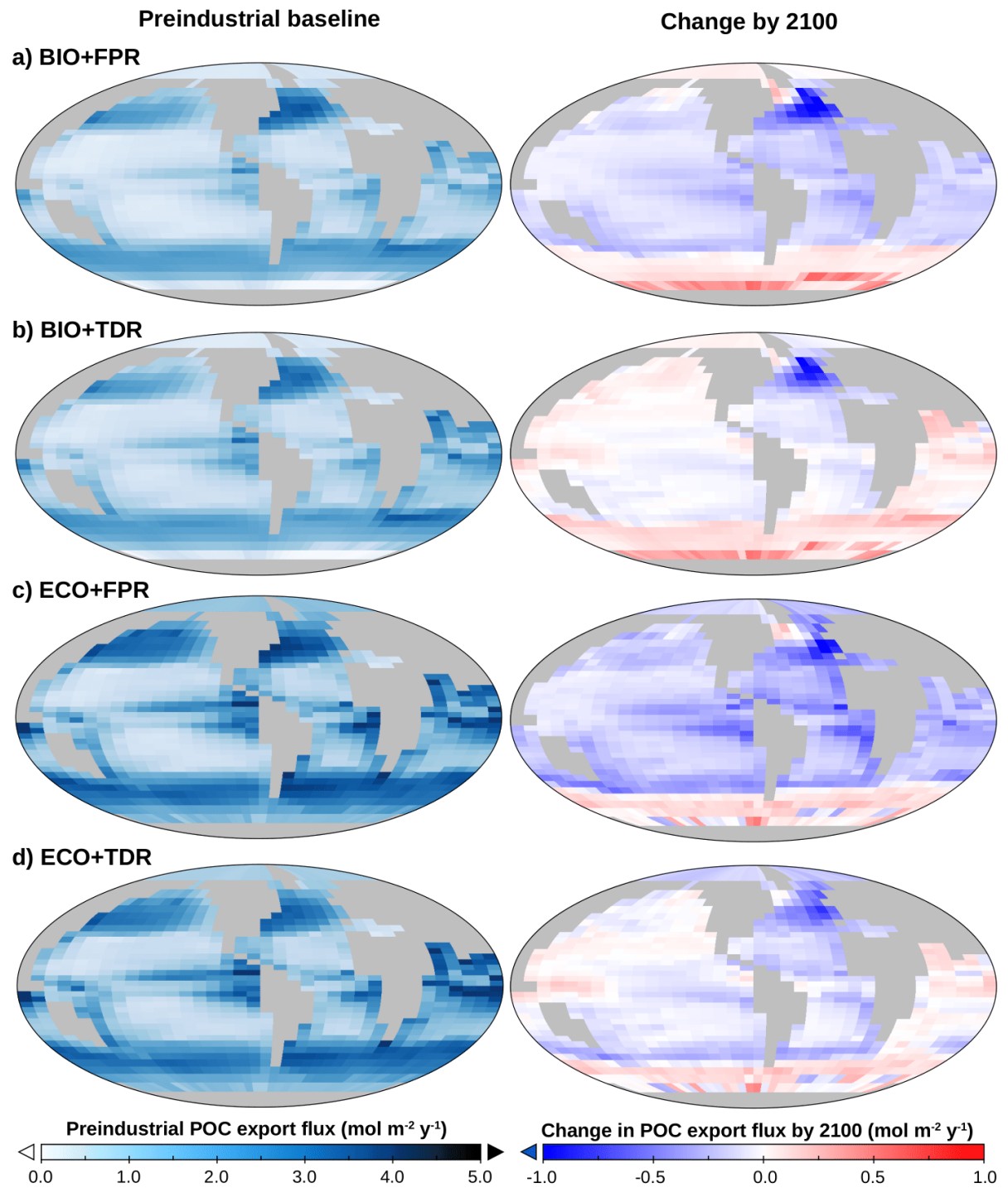

**Figure 3:a) cGEnIE/ecoGEnIE POC export maps for default calibrations of BIO+FPR (a), BIO+TDR (b), ECO+FPR (c), and ECO+TDR (d), showing baseline export patterns (left) and the change in POC export by 2100 relative to the 1765 preindustrial baseline as a result of RCP4.5 (right).** Plot created with Panoply, available from NASA Goddard Space Flight Center.

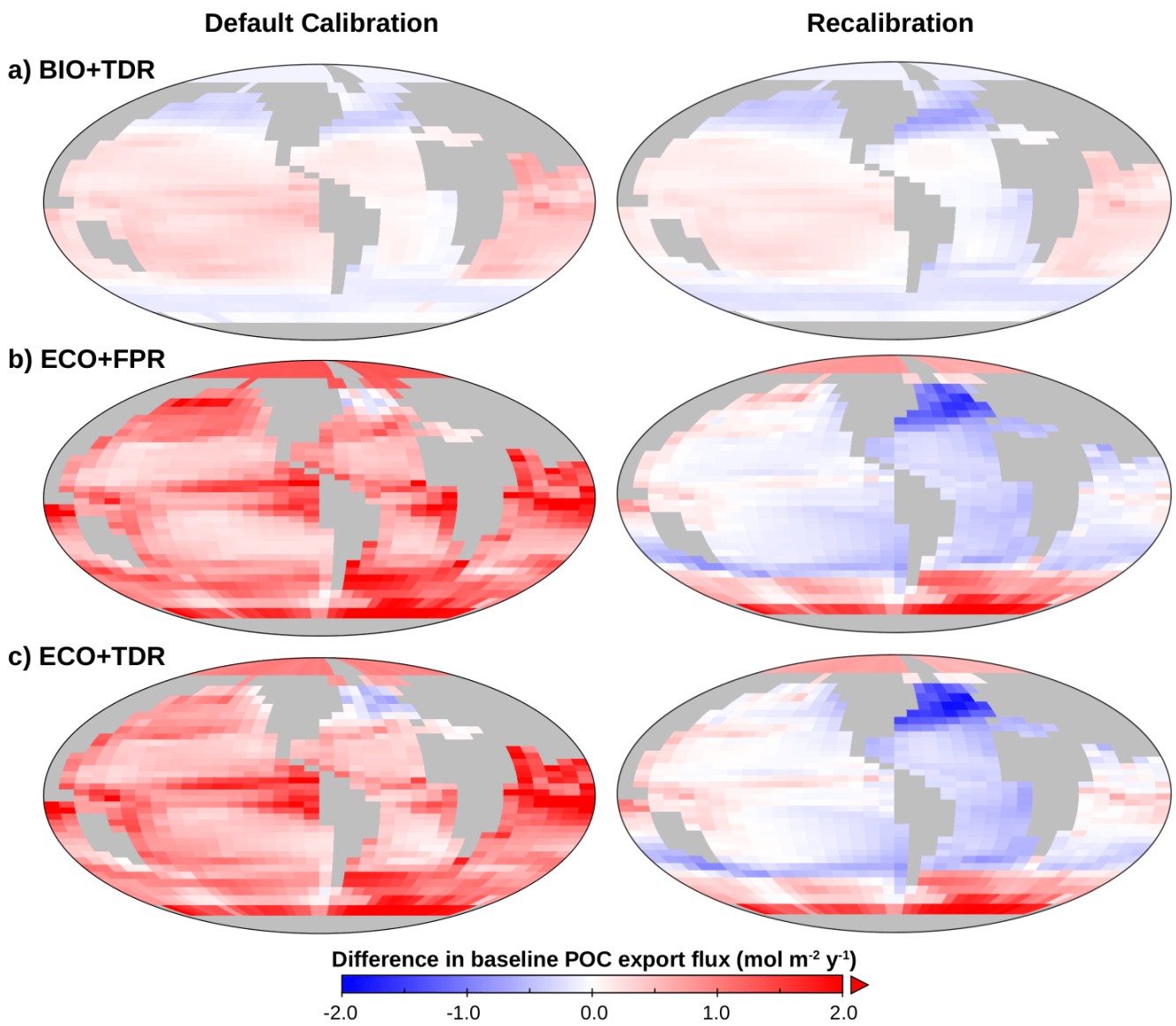

**Figure 4: cGEnIE/ecoGEnIE POC export maps, showing changes in baseline for BIO+TDR (a), ECO+FPR (b), and ECO+TDR (c) in the default calibration (left) and recalibrations (right) relative to the default BIO+FPR configuration (Figure 3a, left).** Plot created with Panoply, available from NASA Goddard Space Flight Center.

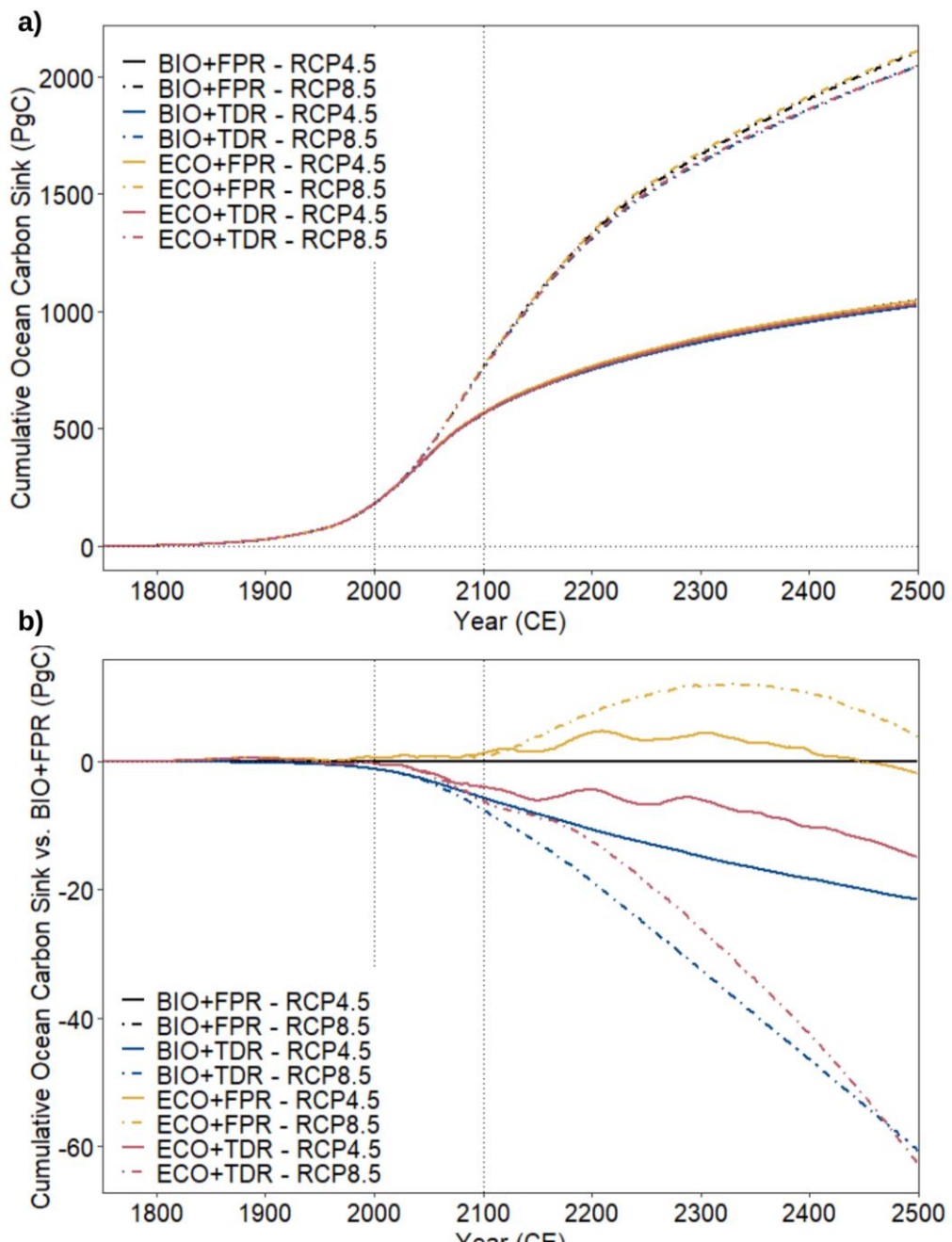

**Figure 5: cGEnIE/ecoGEnIE simulation results for a) the absolute cumulative ocean carbon sink and b) the cumulative ocean carbon sink relative to BIO+FPR under different configurations and forcing scenarios.** Results for RCP4.5 (solid lines) and RCP8.5 (dot-dashed lines) are shown for each of the default calibration configurations (BIO+FPR – black; BIO+TDR – blue; ECO+FPR – yellow; ECO+TDR – red), and the 21st century marked by the vertical dotted lines.

**Table 1:** **Features critical for resolving biological pump dynamics of CMIP5 ESMs used to simulate ocean carbon sink projections in IPCC AR5.** Details based on IPCC AR5 WG1 Table 6.11, Table 9.A.1, and cited literature. Note that there are some mismatches between number of functional groups reported in the literature and the IPCC description. Highlighted cells indicate the models with the most (\*\*) or moderately (\*) comprehensive – but not necessarily sufficient – representation of the relevant model feature.

| ES Model (variant) | BGCM module | Key references | Type ※ | Functional Groups † | Remineral-isation ‡ | Element Cycles § | Elemental Stoichiometry | | Limiting Nutrient | Shell Si/Ca ◇ | Ballasting option ⸎ |
|---|---|---|---|---|---|---|---|---|---|---|---|
| BCC-CSM1.1 | (Inc.) | Based on OCMIP2 & MOM4; (Najjar et al., 2007; Wu et al., 2014) | Nutrient-restoring | 0T: N/A [0] | Fixed rate | C, P, O | Fixed | | P | N/A | N/A |
| CESM1 (BGC) | BEC | (Long et al., 2013; Moore et al., 2013) | **PFTs | **4.5T: 3.5P (diatom, diazo., small + cocco. fraction) 1Z [4] | Fixed rate (soft / ballast) | **C, N, P, Fe, Si, O | *Fixed quasi-Redfield (except diaz) Fe quota | | **N, P, Fe, Si | **Diatoms (Si) & Coccos (Ca) separated by classes | *Yes, part of fixed remin. rate |
| CanESM2 | CMOC | (Arora et al., 2009, 2011; Christian et al., 2010) | NPZD | 2T: 1P 1Z [1X] | **TDR | C, N | Fixed - Redfield | | N, (Fe-param'd) | Temp-depend rain ratio for Ca, no Si | No mention |
| GFDL-ESM (2G & 2M) | TOPAZ2 | (Dunne et al., 2012, 2013; Henson et al., 2009) | **PFTs | **3.5T: 3P (small, large, diazo.) 1Z implicit [6X] | Fixed rate (an/oxic) | **C, N, P, Fe, Si, O | **Variable N:P, optimal allocation, PFe quota | | **N + Am, P, Fe | *No classes; opal (Si), calc., & arag.(Ca) "diagnosed" | **Yes |
| HADGEM2 (CC & ES) | Diat-HadOCC | (Collins et al., 2011; Halloran, 2012; Jones et al., 2011; Palmer and Totterdell, 2001) | **PFTs | *3T: 2P (non/diatoms) 1Z [3] | Fixed rate | C, N, Fe, Si | Fixed - Redfield | | *N, Fe, Si (& DMS) | **Si/Ca separated as diatom/non classes | No mention |
| INM-CM4 | (Inc.) | (Volodin, 2007) | Parameterised POC flux | 0T: N/A [0] | N/A | C | N/A | | N/A | N/A | N/A |
| IPSL-CM5 (A-LR, A-MR, & B-LR) | PISCES | (Aumont et al., 2003; Aumont and Bopp, 2006; Dufresne et al., 2013) | **PFTs | **4T: 2P (diatom, nano) 2Z (micro, meso) [2X] | Fixed rate (an/oxic) | **C, N, P, Fe, Si | *C,N,P Redfield; Fe,Si quota | | **N + Am, P, Fe, Si | **Diatom Si class; ad hoc parametrised Rain Ratio for Ca | No |
| MIROC.ESM (- & CHEM) | (NPZD-type) | (Kawamiya et al., 2000; Schmittner et al., 2005; Watanabe et al., 2011) | NPZD | 2T: 1P 1Z [2] | Fixed rate | C, N | Fixed - Redfield | | N | N/A | No mention |
| MPI-ESM (LR) | HAMOC C5 | (Ilyina et al., 2013) | | | | | | | | *Si/Ca fractionated by Si availability | No mention |
| NorESM1 (ME) | HAMOC C5 | (Tjiputra et al., 2013) | NPZD | 2T: 1P 1Z [2] | Fixed rate (an/oxic) | **C, N, P, Fe, Si | Fixed - Redfield | | **N, P, Fe | | |

※ NPZD = Nutrient Phytoplankton Zooplankton Detritus pools; PFTs = Plankton Functional Types (diatoms, coccolithopores, etc.)

† #T:=No. Total; #P=No. Phytoplankton types; #Z= No. Zooplankton types; [#]= IPCC AR5.1 table 6.11 No. plankton types; [#X]= mismatch between cited literature and IPCC AR5.1 Table 6.11

‡ Fixed rate = prescribed remineralisation profile for sinking POC (sometimes split by class); TDR= temperature-dependent remineralisation

§ Major & minor nutrient cycles present

| Fixed= set ratio of C:N:P etc. (e.g. Redfield Ratio) in OM; Variable/quota= OM can take up / store differing ratios of nutrient relative to C

¶ One major limiting nutrient (P or N), co-limitation by both, and/or micronutrients (e.g. Fe) as well. Am=ammonium

◊ Silicifiers & calcifiers differentiated (& by parameterisation or by functional classes)

⁇ Ballasting (OM sticks to sinking heavy PIC) available as an option. N/A means not applicable (model type does not allow ballasting to be parameterised); No mention means ballasting is not mentioned in the key references (implying ballasting is not available).

Table 2: Simulated changes in POC export and air-to-sea CO₂ flux by 2100 CE under different climate scenarios (CMIP5 RCPs 3PD, 4.5, 6.0, and 8.5), illustrating the relative changes in biological pump strength and ocean carbon sink capacity respectively.

| Climate scenario (CMIP5 RCP) | Model Configur-ation | Biological Pump Strength | | | Ocean Carbon Sink Capacity | |
|---|---|---|---|---|---|---|
| | | POC export in 2100 (PgC) *(% change vs. preindustrial)* | Cumulative ΔPOC export by 2100 relative to preindustrial rates (PgC) *(%)* | Cumulative ΔPOC export by 2100 relative to BIO+FPR (PgC) *(%)* | Cumulative Air-to-Sea CO₂ transfer by 2100 (PgC) *(% in ocean vs. total)* | Cumulative Air-to-Sea CO₂ transfer by 2100 relative to BIO+FPR (PgC) *(%)* |
| **3PD** (low warming, 1.8°C by 2100) | **BIO+FPR** | 7.20 *(-4.2%)* | -35.59 *(-1.4%)* | 0 *(0%)* | 450.1 *(52.4%)* | 0 *(0%)* |
| | **BIO+TDR** | 8.15 *(+1.2%)* | -2.86 *(-0.1%)* | +32.72 *(+1.2%)* | 445.6 *(51.9%)* | -4.52 *(-1.0%)* |
| | **ECO+FPR** | 10.74 *(-5.3%)* | -66.60 *(-1.7%)* | -31.02 *(-0.8%)* | 451.8 *(52.6%)* | +1.75 *(+0.4%)* |
| | **ECO+TDR** | 10.84 *(-1.5%)* | -29.86 *(-0.8%)* | -5.72 *(+0.2%)* | 447.6 *(52.1%)* | -2.51 *(-0.6%)* |
| **4.5** (moderate warming, 2.6°C by 2100) | **BIO+FPR** | 7.05 *(-6.1%)* | -40.57 *(-1.6%)* | 0 *(0%)* | 568.4 *(44.3%)* | 0 *(0%)* |
| | **BIO+TDR** | 8.10 *(+0.6%)* | -5.56 *(-0.2%)* | +35.01 *(+1.3%)* | 562.7 *(43.9%)* | -5.70 *(-1.0%)* |
| | **ECO+FPR** | 10.49 *(-7.5%)* | -74.89 *(-2.0%)* | -34.32 *(-0.9%)* | 569.8 *(44.4%)* | +1.37 *(+0.2%)* |
| | **ECO+TDR** | 10.70 *(-2.7%)* | -35.34 *(-1.0%)* | -5.23 *(+0.1%)* | 564.5 *(44.0%)* | -3.92 *(-0.7%)* |
| **6.0** (high warming, 3.2°C by 2100) | **BIO+FPR** | 6.94 *(-7.5%)* | -42.16 *(-1.7%)* | 0 *(0%)* | 641.6 *(38.0%)* | 0 *(0%)* |
| | **BIO+TDR** | 8.03 *(-0.2%)* | -7.02 *(-0.3%)* | +35.13 *(+1.3%)* | 635.3 *(37.6%)* | -6.30 *(-1.0%)* |
| | **ECO+FPR** | 10.31 *(-9.1%)* | -77.37 *(-2.0%)* | -35.21 *(-0.9%)* | 642.5 *(38.0%)* | +0.89 *(+0.1%)* |
| | **ECO+TDR** | 10.57 *(-3.9%)* | -37.59 *(-1.0%)* | -4.57 *(+0.1%)* | 636.7 *(37.7%)* | -4.93 *(-0.8%)* |
| **8.5** (severe warming, 4.2°C by 2100) | **BIO+FPR** | 6.77 *(-9.8%)* | -48.63 *(-1.9%)* | 0 *(0%)* | 766.3 *(31.5%)* | 0 *(0%)* |
| | **BIO+TDR** | 7.98 *(-0.8%)* | -10.27 *(-0.4%)* | +38.36 *(+1.4%)* | 758.7 *(31.2%)* | -7.64 *(-1.0%)* |
| | **ECO+FPR** | 10.00 *(-11.8%)* | -88.45 *(-2.3%)* | -39.82 *(-1.0%)* | 767.3 *(31.6%)* | +0.99 *(+0.1%)* |
| | **ECO+TDR** | 10.40 *(-5.4%)* | -44.69 *(-1.2%)* | -3.94 *(+0.1%)* | 760.2 *(31.3%)* | -6.15 *(-0.8%)* |