# Peer review of "Resolving ecological feedbacks on the ocean carbon sink in Earth system models"

_Earth System Dynamics, 2020_

## Referee Comment (RC1) · Anonymous Referee #1 · 10 Aug 2020

Summary

The manuscript uses two biogeochemical models (one simple, one complex) and two remineralisation schemes (one invariant in temperature, one temperature-dependent) in a factorial experiment to investigate how export production and carbon storage of the ocean change over the 21st century (and beyond). They find that: 1. Export production declines more in the complex model because of structural foodweb changes; 2. Export production declines less when temperature dependent remineralisation is used, regardless of ecosystem complexity; 3. There is a complex relationship with ocean uptake of carbon, mediated in part by model calcification. The authors argue

these results underscore the need for model complexity (realistic dynamic ecology and temperature-dependent remineralisation) to properly represent the ocean's carbon cycle feedbacks.

Major comments and assessment

I have a number of minor points across the manuscript and supplementary (see below), as well as several major comments about the study:

1. Considering that this is a study focused on export production, the authors skip over how exactly this is defined in their model, and how this relates to what other researchers conventionally understand by the term. More generally, the paper's analysis largely avoids what the different models and parameterisations mean for the patterns (geographical, temporal) and magnitudes of export flux, and even how this varies with depth. The major statistics chosen for analysis deal simply with export at some undefined horizon and with how this affects air-sea exchange, and doesn't really look at spatiotemporal patterns of ocean DIC content (e.g. depth distributions).

2. The authors do not adequately describe either the physical or biogeochemical models that they are using. Given this study is focused on the vertical transfer of organic (and inorganic) material, I would expect some description of the physical domain (e.g. vertical grid), and reassurance that important vertical processes are represented (e.g. mixing). Similarly, key results appear to depend on model parameterisations of calcification, but details of this are entirely omitted. These omissions make it difficult to determine the validity of some of the conclusions reached. (Even the precise nature of the scenario simulations, e.g. emissions- vs. concentration-driven is not made clear.)

3. On a related point, the manuscript's evaluation of model performance (physical and biogeochemical) is insufficient. There is nothing on how physically realistic the preindustrial and future scenarios are; for instance, how aspects such as the magnitude and pattern of temperature and mixing change compare to other more physically-realistic models. These are of key importance to the biogeochemistry models used here, and

specifically to the temperature and nutrient responses investigated.

4. While deficiencies with models are always inevitable, it is important that these are addressed in manuscripts. The noted lack of model description and evaluation are joined by a lack of appropriate discussion and caveats. Combined with the authors strong (over-confident?) conclusions, these absences make it difficult for readers to accurately assess the significance of this manuscript.

5. When presenting its key findings of change in export production between the various combinations of model and parameterisations, the manuscript does not any provide contextualising information. How, for instance, do the modelled changes compare to those of the CMIP models mentioned elsewhere? Are the changes found here within or outside the ranges found by these other (less sophisticated) models? This omission, in particular, makes it difficult to understand this work in context. Table 1, for example, could be expanded to include such metrics from CMIP5.

All that said, the manuscript is otherwise relatively straightforward to understand and follow. And I am fairly confident that their findings are accurate in identifying certain processes as important, and therefore relevant to both observational and modelling scientists. However, it remains the case that the manuscript's omissions (both in background and analysis) undermine its credibility, and I recommend major revisions before it can be accepted for publication.

Minor comments

Ln. 28: saying "gradually" in the same sentence as "centuries to millennia" might convey the wrong impression; I might be inclined to just delete "gradually"

Ln. 32: what do you mean by "sink feedbacks"?; this could be enhanced stratification decreasing mixing and / or nutrient supply; be clearer please

Ln. 40: what about AR6?; quite a lot of model output is available, together with a lot of supported information about model formulation; you might also wish to consult
doi:10.1007/s40641-020-00160-0 or doi:10.5194/bg-17-3439-2020 on AR6

Ln. 52: a "this manuscript is structured as follows" (or similar) might help here

Ln. 55: erm, what about a straight decline fed by a decreased amount of production?

Ln. 60-61: I seriously doubt this; there is a large body of work focused exclusively on the soft pump whereas there is far less focus on the hard and silicic acid pumps; in part because, unlike the soft pump, the activity of these latter pumps is less ecologically-driven and more physiochemically-driven (cf. the importance of the CCD)

Ln. 72: note that Kwiatkowski et al. (2014) describes 6 models, of which only 1 has a single size class; the rest have at least 2 size classes

Ln. 74-76: as opposed to some other "class of plankton model" which is not limited ...

Ln. 84: can you give an idea why such evidently superior (as described here) models still fail to capture BGC and "large-scale" dynamics?; do you mean physical resolution problems?

Ln. 94: "shoaling" rather than "raising"?

Ln. 94: "the point at which most POC is remineralised" - is this a formal definition you're using in this study?; it might be good to give it a name (e.g. R_50, or something) if so

Ln. 95: at some point it might be nice to demonstrate that faster remin. leads to greater export despite the remin.; maybe that's yet to come

Ln. 98-99: cf. my previous remark, you'll need to demonstrate that faster remin. doesn't act as a *positive* feedback mechanism (i.e. warming -> faster remin. -> less C export -> more atmospheric CO2 -> more warming -> ...)

Ln. 107: "will not substantially affect productivity in existing oligotrophic regions" - is this actually important?; the important aspect of oligotrophication is that formerly productive regions of the ocean become less productive; that unproductive regions remain equally

unproductive is not obviously important

Ln. 107: "the depth rather than the intensity" - my understanding is that most research already focuses on the depth and not the intensity of stratification; usually studies focus on change in mixed layer depth - key for resetting surface nutrient conditions during periods of deeper seasonal mixing (e.g. winter)

Ln. 111: "result in ocean interior deoxygenation" - as does warmer surface temperatures that control the oxygen concentration of water ventilated into the interior

Ln. 111: "reduce nitrogen availability" - are you referring to denitrification here?; you should be specific if you are; otherwise one might assume you mean reduced nitrogen (and other nutrients) from reduced mixing

Ln. 114-117: it might be helpful to note the significant uncertainties in OA feedbacks on marine BGC; something like doi:10.1146/annurev-environ-012320-083019 might help here

Ln. 116: "by which POC sticks to denser falling PIC" - this kind-of reads as POC is somehow "attracted" to CaCO3; instead, the ballast hypothesis is more about POC *already associated* with CaCO3 somehow being "protected" by it to reach further into the ocean interior

Ln. 124: "IPCC AR5" - as already mentioned, the archive of simulations used in AR6 is already quite well-stocked with models

Ln. 130: "(Bendtsen et al., 2015; Dunne et al., 2007; Martin et al., 1987)" - this list of citations is ambiguous; are these submodels of remineralisation used in specific models in Table 1 or what?

Ln. 130: Given that Dunne et al. invokes a ballast model that dynamically alters the remineralisation profile, this characterisation seems inaccurate

Ln. 131: this statement is characteristic of many in this manuscript; i.e. bombastic

on how deficient existing models are; given this manuscript is using a reduced-physics ocean model with extremely poor vertical resolution (<= 16 levels; it's unclear, see below), a more measured toned might seem appropriate

Ln. 141-154: this description must be augmented by a description of the underlying physical model so that readers understand that it imposes its own limitations on this study; while pointing to other uses of the model is fine, a minimal description that notes model resolution (horizontal, vertical, temporal) and the reduced physics nature of this model is key; noting what the model does around processes relevant to export production (e.g. mixing, convection) would help readers unfamiliar with this particular EMIC

Ln. 151: "have lower spatiotemporal resolution" - cf. my previous remark, expand on what's meant by "lower"

Ln. 151-154: ". . . well-suited for investigating more complex biogeochemical dynamics . . . " - only if the physics they represent is up to the job, of course; and that may depend on the specific application; a reader should be interested in how reduced vertical resolution impacts a model of vertical POC transfer

Ln. 155: the history of the models can probably be skipped to focus on what's being used here; unless it's important for this particular study (which it might be)

Ln. 161: what's the relationship between the size classes?; does each predator graze everything smaller than it is, or is there another scheme?

Ln. 161: it would be helpful to have some idea (e.g. a sentence or two) on how the different size classes differ from one another; e.g. photosynthesis parameters, nutrient uptake, growth rates, grazing rates, mortality rates

Ln. 168: "16 layers" - ah, finally!

Ln. 169: does the model resolve seasonal mixing of different ecological regimes?

Ln. 170: is there information about what "not quite as well" means?; e.g. which properties were examined to determine this?; ones directly relevant to this study, or more indirect?

Ln. 176: ah-ha; we do need to know about BIOGEM; good!

Ln. 186: "POC export" - how is this defined?; it is the flux at a particular depth, or just out of the model's uppermost level?; in cGENIE this can be quite deep ($\sim$80m?), but it may not conform to field norms; e.g. Martin et al. (1987) used 100m

Ln. 186: "Gt C" - The SI unit is Pg C rather than Gt C

Ln. 191: "shorter sub-overturning timescales" - this is opaque; I presume you mean that, as this model doesn't include sediments, DIC is conserved within the ocean regardless of export production (which might not be true if it were "lost" to sediments); but I don't understand the reference to timescales; not least because a 10,000 y simulation should be enough for several complete overturnings of the ocean

Ln. 193: "total CO2 emission scenarios" - is the model being run in emissions mode (i.e. a time-varying amount of CO2 is being added to the model atmosphere and then redistributed, including into the ocean) or in concentrations mode (i.e. a time-varying atmospheric concentration is specified but cannot be affected by different ocean uptake responses); make this clear

Ln. 193: this list of scenarios seems to omit the low scenario RCP 2.6; any reason?; this unlikely, low emissions scenario (as SSP126) is still used in CMIP6

Ln. 193: "3PD, 4p5, 6p0, and 8p5" - these scenarios are more typically referred to as e.g. RCP 8.5, where the period is a decimal point

Ln. 199: do you evaluate the performance of your modelled warming relative to other models (e.g. CMIP5, CMIP6)?; as the marine BGC models you're using are sensitive to warming, it would be useful to know how realistic this is in the model; it should be straightforward to compare model output to, say, corresponding CMIP5 / CMIP6 output;

e.g. change in magnitude / pattern of ocean temperature, mixing, etc.

Ln. 200: cf. my last remark, maybe say something here about what happens physically in your scenario simulations

Ln. 200: you should also perhaps begin by discussing how the different models represent the pre-industrial situation; some of this is covered (I think) in the supplementary material, but I have comments there too

Ln. 217: "0.3%" - I've mentioned over at the table itself, but you might like to add such relative stats there

Ln. 221: how is the mixed layer handled in cGENIE?; older versions of the model don't really have a mixed layer

Ln. 223: do you really mean "new production" here?; the distinction may be confused by changes in mixed layer depth itself across the scenario period

Ln. 268: this part of the manuscript is confusing without some clarity on how the hard pump works in this model; both in terms of CaCO3 production (and controls on this), and how it dissolves down the water column

Ln. 277: it's difficult to tell, but this just sounds like the two models differ in the strength of their hard pumps (probably relative to the soft pump), with the result that they have different hard pump changes relative to soft pump ones into the future; and because the models are not well-described on this point, it's hard to decide what's going on

Ln. 282: why "initial"?

Ln. 282: "the importance of incorporating multiple dimensions of ecological complexity" - the paper doesn't really present anything concerning the modelled ecological complexity; passing comment is made on shifts in size structure, but nothing is shown, not even as supplementary information

Ln. 284-287: the modelled change in export production is presented without any contextualised reference to other work on this; there is a large body of published work on how export may change, ranging from studies using individual models through to meta-analyses using, for instance, CMIP output; to make clear the significance of the distinction the authors are highlighting, the existing span of estimates needs to be clear

Ln. 286: "a much simpler NPZD-based ecosystem" - much simpler, perhaps, but possibly different in an important way for the hard pump

Ln. 307-309: I wouldn't necessarily expect TDR to increase the production of diatoms because opal is dissolved and not remineralised; if anything, one might expect diatoms to do less well as time passed because - although N and P might be becoming more available (because they're getting remineralised faster) Si would not be; there's probably some subtlety I'm missing here, however; expand to make clearer

Ln. 311: "a subsequent increase in calcifying plankton and PIC export" - this result may be quite dependent on how calcifiers are modelled; this already isn't clear in this study, so I'd suggest drawing the parallels out more fully

Ln. 314: "which we have shown is critical" - I don't think this has been clearly shown here; the model is too incompletely described, and the physical model probably leaves something to be desired

Ln. 320: there are no caveats in this paper about the quality of the model, physical or biogeochemical; the only caveats seem to relate to making the biogeochemical model even more complex without any consideration of whether the physical framework is adequate; work has emphasised the potential importance of physical frameworks for BGC models, e.g. doi:10.1016/j.pocean.2009.10.003

Ln. 320: things left undiscussed include: 1. how realistic is this model's response under climate change (e.g. pattern and magnitude of temperature change; compare with CMIP5 / CMIP6); 2. can this model realistically represent mixing; 3. can this model realistically represent vertical gradients of properties given grid cell thicknesses

(even close to the surface); 4. how dependent are results on (undescribed) hard pump submodel

Ln. 320: some of the above points cannot easily be addressed here; but they should be properly acknowledged and discussed, and they should temper the conclusions drawn here; it may well be that these are accurate, but the physical and BGC models used here should give some pause for thought

Ln. 320: per my comment on Table 2, the different scenarios get pretty short shrift here; they're just stand-ins for different amounts of warming / emissions; it's not clear that they couldn't be thinned to small, medium and large warming

Ln. 320: equally, it's not made very clear what the differences between the atmospheric $CO_2$ concentrations across the scenarios mean for the ocean uptake numbers here; we should expect larger numbers for higher RCPs, but does efficiency of uptake of $CO_2$ change (or is that too far for this paper?)

Ln. 322: "critical"? - while there may well be feedbacks such as those described here, I think the authors are arguably exaggerating their importance, especially given the magnitude of the numbers they find; I'd suggest "may be important" is more suitable wording

Ln. 328: "as expected" -> "than expected"?

Ln. 326-329: this sentence is too long to be parsed well; it's important so make it clear

Ln. 333: "post-CMIP5 projections"? - you might need to explain what you mean by this

Fig. 1: "surface layer" = cGENIE's top box?

Fig. 2: perhaps use different colours rather than linestyles to separate the four different models?; then save solid and dashed styles for the two scenarios

Ln. 651: it might be helpful to note the distinction you're drawing between "N/A", "No", and "No mention" here; also, assuming "no mention" means you couldn't find any reference to this in the model descriptions, have you considered contacting the model authors to ask?

Table 2: while this sort of summary is of key importance, it might also be useful to see how these numbers change in time (beyond the single supplementary figure)

Table 2: it occurs that the manuscript does not clearly address the different scenarios; it might be better to reorganise so that the results are organised by model first and then by scenario; that way the span of results between scenarios (i.e. the effect on the properties for different degrees of warming) are clearer to see

Table 2: this kind-of omits what happens for the period 1850-2000; it might not be important, mind

Table 2: might it be useful to note what the changes in this table represent in relative terms as well (e.g. what's this delta as a percent of the total flux over this period?)

Table 2: columns 4 and 6 - "default cGENIE" = "BIO+FPR", so perhaps just say that?

Table 2: thinking about the ocean uptake column, what about the efficiency of ocean uptake and how this varies with scenario and time?; this may require information about emissions (see my previous remark about emissions vs. concentration simulations)

Supplementary material

Ln. 17: perhaps show the observational field as well so that the relative size of these errors is clear

Fig. S1: which time point is being compared here?; presumably near-present day given the choice of observational product

Fig. S1, caption: when you say "surface" are you comparing the concentration in the uppermost layer of GLODAP with the uppermost layer of your model, or are you depth-averaging so that the intercomparison is fairer?; if not, you will need to explain why the intercomparison you're doing is the right one; this applies to alkalinity too (and nutrients

if you plot them)

Fig. S1, caption: why not write this as 2 mmol / kg?; ditto for the graphs; "E-0X" notation is a little annoying when we've got scientific prefixes available to us

Figure S3: this pattern looks interesting; is it salinity-related?; i.e. does it reflect a bias in model salinity?

Fig. S3: alkalinity is usually given in equivalents rather than mols

Fig. S3, colour scale: "1.00E-04" - see previous remark about units

Fig. S5: I'm assuming annual mean chlorophyll here; although I note that the model's Arctic is negatively biased to almost 1 mg chl / m3 - that implies quite a high annual mean observational chlorophyll; has it been time-averaged correctly?

Fig. S5: chlorophyll is not usually a brilliant metric to compare models to; I'd suggest using nutrients

Fig. S5: also, you could compare to productivity; that possibly is even more relevant to the problem at hand

Fig. S5, colour scale: I see the "milli" prefix is getting used here! ;-)

Fig. S6: I presume these colour scales are being used for parity with the previous delta plots?; I understand that, but it might be more informative to use a more relevant colour scale to help readers delineate where models differ geographically

Figure S19, colour scale: "5.8E-07" - as well as "milli", there is also "micro"

---

## Referee Comment (RC2) · Jamie Wilson (Referee) · 10 Sep 2020

Armstrong McKay et al., present a set of Earth system model experiments to demonstrate the importance of resolving plankton ecosystems for the ocean carbon sink when considering biogeochemical feedbacks. In particular they focus on the temperature dependent remineralisation of particulate organic carbon (POC) as a key feedback. They achieve this by simulating future climate scenarios in an Earth system model with both a parameterised export production scheme and trait-based ecosystem model. They find that the strength of the biological pump, i.e., export production of POC, generally increases when adding temperature dependence of POC. In contrast, export production

decreases significantly when using the plankton ecosystem model. The net change in the carbon sink associated with these changes is further modulated by concurrent changes in the inorganic carbon pump.

In general, the concept of the manuscript is interesting given that few studies have approached the interactions between ecological and biogeochemical complexity due to computational limitations. The use of EcoGEnIE here facilitates this novel idea in a straightforward and logical way. The key results seem generally sound but there are significant parts of the results that are not backed up with figures and many explanations about the role of different processes that are not quantified. As such, the manuscript makes a good case for resolving ecological complexity but not necessarily which components of this complexity are important and why. The manuscript would benefit from major revisions including new figures and additional experiments to quantitatively show how the various components of the ecological complexity lead to the main results.

General Comments

1) Calibration of global biological pump strength

The spin-ups are all calibrated to have the same global POC export ($\sim$7.5 Gt C year-1) but there are no details about how this has been achieved in the model, i.e., what parameters were modified. However the calibration has been achieved it needs to be described as the BIO+TDR and ECO+FPR set-ups now differ from their published versions (John et al., 2014. P3; Ward et al., 2018. GMD).

The supplementary plots comparing each set-up after calibration also need to be more comprehensive. It is not totally surprising that surface fields are similar across set-ups as POC export has been calibrated to the same global value, particularly for those fields that are strongly influenced by export such as PO4. These fields may then differ more in the ocean interior. The authors should add difference plots showing depth slices for the various fields and/or a Taylor diagram to show how the calibration affects

global fit statistics like correlation and standard deviation.

The main concern I have is that the authors may have achieved the same POC export by altering parameters associated with POC remineralisation - because the BIO+FPR and ECO+FPR output in Table S1 should have the same POC sedimentation:export ratio if the fixed remineralisation profile is the same. Apologies if this is not the case, but if it is it may have implications for the results in the manuscript. Firstly, the differences in POC export in Figure 4 would be a combination of adding the various TDR and ECO components but also the calibration adjustments that presumably vary across set-ups to achieve the same POC export. (This is significant for any calibration). Secondly, if the ECO experiments have deeper remineralisation to offset the higher POC export in the Ward et al., (2018) set-up, this could potentially bias the results if the deep ocean takes longer to experience changes in temperature, i.e., the transient ECO response may be slower due to the calibration. The relative change in carbon/nutrient feedbacks may also differ because the residence time of carbon/nutrients in the ocean interior is different and because carbon/nutrients may be redistributed spatially via different circulation pathways (e.g., Pasquier and Holzer 2016, JGR Oceans). While I don't think this changes the general findings of the manuscript, it does make me question the relative magnitude of changes between each set-up.

I do appreciate that the baseline states will aways differ in some way because of the use of different parameterisations! The authors need to acknowledge the reason for choosing to constrain POC export across runs and what issues this may introduce, e.g., are there differences in spatial export patterns; are you compensating for any errors in the circulation and biogeochemical model? Alternatively, the original BIO+FPR, BIO+TDR and ECO+FPR set-ups have all been (somewhat) calibrated to achieve similar global distributions of dissolved tracers compared to observations. The authors could repeat their experiments with these published set-ups and recalibrate just the ECO+TDR set-up to achieve similar global tracer distributions. This can be defined using various fit statistics like root mean square error. This alternative set of results

would help demonstrate that the POC export calibration is not biasing the results.

2) Background and Model Description

The description of processes in the Background section and the model description is too brief to support the main results. The biological pump is described mainly in terms of export production but has little description of the role of POC remineralisation and circulation. A few statements describing that the POC flux rapidly decreases with depth to a small asymptotic flux by ~1000m and that the ventilation age of the ocean increases with depth would really help clarify a lot of later statements in the results. Similarly, there is no basic description of the allometric relationships for plankton and how they relate to metrics like primary production.

The model description is also very sparse in specific details that would aid the reader in understanding the results in more detail. For example, important details such as the saturation-state dependent PIC:POC rainratio (Ridgwell et al., 2007: Biogeosciences) and the nature of allometric trends like size-dependent DOC:POC export production (Ward et al., 2018: GMD) are not described. Whilst these are described fully else-where, it would help to describe these briefly as they are directly relevant to the results and discussion.

3) Results from the plankton ecosystem modelling

The description of how plankton ecosystem structure impacts the biological pump is difficult to follow (mainly lines 231 - 246 and other related sentences throughout). There are a lot of discussion of changes in plankton size but this is never visualised despite being a standard output of the model.

I am not totally convinced by the explanation of why the ecosystem model leads to a greater decrease in export production. Size structure, variable stoichiometry and DOC:POC export ratio are all alluded to throughout the manuscript but there are additional components that haven't been considered. In steady-state warm-climate ex-

periments using the same model there is a net decrease in plankton biomass due to increased grazing pressure because grazing rates are temperature dependent in Eco-GEnIE (Wilson et al., 2018, Paleoceanography and Paleoclimatology). This grazing effect also co-varies with nutrient availability leading to distinct latitudinal trends in size, biomass and export. This needs to be factored into the explanation here. This is a novel application application of a model of this type so it would be really helpful and informative to know what aspects of the ecological complexity are crucial to this result!

4) Results from the Ocean Carbon Sink capacity

I found it hard to follow the logic in this section because the factors involved are not quantified and/or illustrated in figures. A figure illustrating the changes described would really help to clarify the text in this section.

The increase in export production but decrease in carbon sequestration has been noted before (Kwon et al. 2009, Nature Geoscience; Gnanadesikan & Marinov 2008, Marine Ecology Progress Series). The impact on carbon sequestration is in part due to a change in organic carbon cycling and in inorganic carbon cycling but It is not clear in the manuscript what the relative impact of these processes are. This could be separated by running additional experiments with a uniform PIC:POC rain-ratio to remove the impact of any spatial differences in POC export between se-ups and a prescribed spatially variable ratio from the associated spinup to isolate the impact of changing saturation state.

Specific Comments

Lines 18 - 20: the manuscript does not actually show plankton size or deal significantly with ocean acidification

Introduction/Background: generally I found the structure of these sections difficult to follow. Particularly there are a number of concepts and abbreviations in the Introduction, such as Fixed Profile Remineralisation, that are not described sufficiently until the

[Figure]

Background section.

Line 51: cGEnIE does not have a NPZD model. It parameterises the export of production by plankton as a function of nutrient availability using Michalies-Menten kinectics and other limiting factors. This needs to be made clearer throughout the manuscript.

Line 54: "a weakening carbon sink" - w.r.t. anthropogenic climate change?

Line 56: the biological pump is described too briefly here and focused very much on the export of organic matter from the surface. It would help readers to expand here on the additional role of depth variation in remineralisation rates and ocean ventilation ages, particularly as this is a key concept needed to understand the model results.

Lines 60 - 61: this statement surprised me! There have been significant model developments that try to resolve the ecological drivers of the soft-tissue pump such as cell-size and aggregation (e.g., Jokulsdottir & Archer 2016. GMD; Omand et al., 2020. Scientific Reports). I am not sure we are at a stage where we fully understand the interactions yet or are able to couple these models into global biogeochemical models though.

Lines 90 - 91: Strictly speaking it is the metabolic rates that increase between 100% and 200% whereas gross primary production and community remineralisation are additionally limited by other factors.

Line 94: "raising the remineralisation depth . . . higher up in the water column" - this is repetition. Either the remineralisation depth moves higher up or it is raised.

Line 94: "(the point at which most POC is remineralised)" - an e-folding depth is often used to define this as the depth at which 63% of the exported flux has been remineralised.

Lines 106 - 108: please briefly outline why this happens

Line 161: "better representation of biodiversity" - relative to what? If relative to cGE-

nIE then this is really just resolving diversity (and biomass!) compared to the export production parameterisation.

Line 175 & 179: NPZD here is misleading as the export production scheme in cGENIE does not resolve plankton biomass, phytoplankton or zooplankton.

Line 204: though the biological pump strength does increase for the BIO+TDR experiments by 2100

Line 207: How does the 6.1% decrease (and generally across all experiments) compare with CMIP model simulations?

Lines 210 - 212: this is not an explanation of what is happening in cGEnIE as it does not resolve plankton and productivity is restricted to a single surface layer.

Line 218: "more POC is being remineralised with warming" - I struggled to follow the logic of this. Does this mean more POC is remineralised in the surface ocean so lowering export production? If so, this should be checked that POC remineralisation occurs in the surface grid-boxes in cGEnIE and is not exported from the base of the surface layer.

Lines 220 - 221: "warming-induced shoaling of the remineralisation depth has been modelled to reduce POC export (Kwon et al., 2009)" - this may be a typo or the wrong reference? The Kwon paper perturbs the remineralisation depth directly for a fixed climate, and it shows POC export increasing, not decreasing, with increasingly shallower remineralisation depths (e.g., Fig. 2a in Kwon for values of b>0.9).

Line 237: "rapidity of carbon cycling within the surface ocean" - what does this refer to? A shift from POC to DOC export production? If so, I would expect the increase in the rate of nutrient cycling associated with more semi-labile DOC production to rather increase production and biomass because it will be remineralised near the surface due its short lifetime.

Lines 231 - 246: plankton size outputs are available in EcoGENIE but are not plotted

to support any of these statements. This would be an interesting thing to see!

Line 277: "adding ECOGEM reduces total ecosystem POC/PIC production" - i cannot see this in a figure and it is not described or demonstrated why this happens as a result of having ECOGEM

Lines 294 - 304: I wonder if resolving plankton biomass also plays an important role as part of this? Galbraith et al., (2015) in JAMES showed nicely that seasonal/transient behaviour varies between a model with parameterised export of POC and one that explicitly resolves plankton biomass. A parameterised model, like cGEnIE, responds much more rapidly to environmental changes because growth rates are not buffered by a biomass pool. There are a few entries in Tables 1 that this parameterised export model. Following on from this, it would be interesting to speculate what the representation of ecological complexity needs to be to reliably simulate the biological pumps response to environmental change.

Lines 313 - 314: "does not feature trait-based size classes or flexible stoichiometry, which we have shown is critical for determining the soft-tissue biological pump response" - the role of flexible stoichiometry has not been explored here.

Lines 325: "flexible nutrient usage" - the influence/impact of this has not really been quantified or discussed in the manuscript.

Figure 5: it would help to combine these panels with Figure 3 so they can be compared side by side.

Figure 4: I spent most of the time thinking these differences were 2100 vs. baseline because that is the format of the other figures. Expanding the labelling might help clarify this.

Table 1: this is very valuable, thank you!

[Figure]

---

## Short Comment (SC1) · 20 Sep 2020

Conceptual issues:

Page 3, Line 69:
*…up to 4-12 GtCy-1 of POC reaches the deep ocean where it becomes part of the long-term carbon sink on centennial-to-millennial timescales*

We suggest rephrasing as this is not the marine carbon sink of anthropogenic CO2 mentioned earlier in the text. In fact, the biological pump is, in steady state, neither a carbon sink nor source as it fluxes as much (organic) carbon to depth as is transported back in inorganic form to the surface ocean. The timescale describes how long it take one carbon atom to take a full loop, but it does not say anything about a timescale of a possible sink or source of carbon.

Page 9, Lines 259-266:
*In our simulations, the relative strengthening of the biological pump when TDR is included actually leads to a net decrease in the ocean carbon sink capacity during the 21st century (Table 2). Conversely, the relative weakening of the biological pump with ECOGEM activated instead (ECO+FPR) is associated with a net increase in the ocean carbon sink capacity. Combining both ECOGEM and TDR (ECO+TDR) results in a smaller overall relative weakening of the biological pump compared to default, and a marginal net decrease in the ocean carbon sink capacity of ~0.4 GtC (~2.4 GtC under RCP8p5) over the 21st Century. Including trait-based ecology using size classes largely but not entirely offsets the impact on the ocean carbon sink of also including TDR in this model. The model thus suggests that ecological dynamics increases the resilience of plankton ecosystem functioning against the pressures of climate change.*

We believe this interpretation needs to be revisited. The overwhelming contribution to the ocean's uptake of anthropogenic CO2 is from the solubility pump. Though your

models share the same physics, surface ocean DIC and ALK are only 'calibrated' to be similar (lines 188-190). Table S1 indicates that at least BIO+TDR has a considerably smaller surface carbonate ion concentration compared to BIO+FPR, indicating that the carbonate buffer is reduced compared to BIO, suggesting that in that model run the solubility pump should be weaker. It is actually this pair of model experiments, which shows by far the strongest difference in marine CO2 uptake (Table 2). BIO+TDR has lower CO2 uptake, consistent with the lower initial buffer, but inconsistent with the higher POC-export response.

In the follow-up paragraph (lines 268ff) this is being discussed somehow, but still analyzed as an effect of differences of biological pump representations. However, the different marine CO2 uptake seems in fact to result from different surface DIC and ALK concentrations at the end of the respective model spin-ups. If the models were calibrated to have identical surface DIC and ALK values, buffer factors and the responses of the solubility pump to increasing atmospheric CO2 would be similar, allowing a more straightforward identification of biological effects on the oceanic CO2 uptake.

Page 10, Line 289:
*Other processes that are not resolved in this configuration of ecoGEnIE could also substantially affect the biological pump though, such as ballasting, calcifier-silicifier trade-offs, and deep chlorophyll maxima (discussed more fully below), and further work is required to assess their impact on our estimates*

Recent work has also demonstrated the importance of representing interacting N-cycle processes (such as N2 fixation and water column and benthic denitrification) to capture important feedback processes that affect biological export production (Somes et al., 2016; Landolfi et al., 2017) and potentially air-sea CO2 exchange (Buchanan et al., 2019) and ecosystem restructuring (Dutkiewicz et al., 2013). Also, redox-dependent feedbacks in nutrient cycles are not included in most current models, but may be relevant even on centennial timescales and will require an adequate representation of marine oxygen distributions (Watson, 2016, Niemeyer et al., 2017).

Unjustified statements:

Page 1, Line 22 (Abstract) and, similarly, p 10, Line 282:
*These results clearly illustrate the substantial degree to which ecological dynamics and biodiversity modulate the strength of climate-biosphere feedbacks, and demonstrate that Earth system models need to incorporate more ecological complexity in order to resolve carbon sink weakening.*

The last column of Table 2 shows that the addition of ecological complexity results in a maximum weakening of the ocean carbon sink capacity of 2.39 GtC over a 100 year period (a 0.4% weakening). If ecological complexity is removed and only temperature dependent remineralization is considered, the maximum weakening is still only 0.9%. These differences in marine carbon uptake, even if buffer factors were similar and effects were caused predominantly by differences in the representation of biological processes, do not strongly demonstrate the need for

additional model complexity. Instead, if the same metric is applied, one might even argue that uncertainties can probably be reduced by a far larger amount by investigating better representations of, for example, physical processes.

Page 10, Line 297:
*To date, gains in computational power have largely been allocated to improved resolution and physical process representation. This study suggests that it is timely for the research community to debate again where future gains should be focused …*

While we agree that such a debate is important and should always be encouraged, this statement makes the misleading impression that this paper is the first to propose potential gains from shifting some resources from even higher ocean resolution to more complex bgc models. However, what has been hindering the application of more complex biogeochemical models is not necessarily computation-power issues related to performing simulations, but the uncertainty in biogeochemical model parameters and the associated computational costs in properly calibrating these parameters. Fortunately, since a few years, biogeochemical and ecological parameter optimization has emerged as a very active field of research that exploits recent gains in computational power. Please see Frants et al. (2016), Chien et al. (2020), Schartau et al. (2017), Kriest et al. (2020), Kriest 2017, Niemeyer et al. (2019), Sauerland et al. (2019), Yao et al. (2019), to name just a few. So far most of this work has not been carried out with 'proper' ESMs, but with EMICS. To pose all discussion of ocean carbon cycle dynamics research in the framework of CMIP is misleading. Actually, including poorly calibrated more complex bgc models in high resolution ESMs is likely not in favor of more reliable marine CO2 uptake predictions. Since such calibration can be done on the EMIC level, it is rather more important to propose studies with well calibrated models to better understand the relationship between bgc complexity and marine CO2 uptake.

Page 11, Line 318:
*Empirical observations have suggested that the ballasting effect is weaker than hypothesized (Wilson et al., 2012), making ballasting unlikely to substantially alter our findings, but it would likely result in greater surface layer remineralization in scenarios with reduced PIC production*

Please see Kvale et al. (2015, 2019) for 2 explorations of how ballasting can affect export in a temperature-dependent remineralization model (EMIC) over long-term simulations. Depending on how you choose to represent calcification, the ballasting effect can alter the pathway to the long-term response of your model via modification of suboxic volumes, which regulate denitrification, nitrate availability, and hence primary and export production.

In summary, this manuscript could be improved on two fronts. The first is stylistic, in which the introduction and discussion of the state-of-the art should include references to the recent ecology and biological pump work happening with both EMICs and ESMs. This is important for giving proper context to the present study. The second is technical, in which the major conclusions must be shown to be independent of different states of the carbon chemistry at the end of the spin up of

the respective models. This can be demonstrated, for example, by carbon separation techniques (e.g., Koeve et al., 2014) or a better model calibration that adequately controls for buffer chemistry. On technical aspects we are always happy to offer advice, and invite the manuscript authors to contact us for further discussion.

References

Buchanan, P.J., Chase, Z., Matear, R.J. et al. Marine nitrogen fixers mediate a low latitude pathway for atmospheric CO2 drawdown. Nat Commun 10, 4611 (2019). https://doi.org/10.1038/s41467-019-12549-z

Chien, C.-T., Pahlow, M., Schartau, M., and Oschlies, A.: Optimality-Based Non-Redfield Plankton-Ecosystem Model (OPEMv1.0) in the UVic-ESCM 2.9. Part II: Sensitivity Analysis and Model Calibration, Geosci. Model Dev. Discuss., https://doi.org/10.5194/gmd-2019-324, accepted, 2020.

Dutkiewicz, S., B. A. Ward, J. R. Scott, and M. J. Follows (2013), Winners and losers: Phytoplankton habitat and productivity shifts in a warmer ocean, Global Biogeochem. Cycles, 26, GB1012, doi:10.1002/gbc.20042

Frants, Marina; Holzer, Mark; DeVries, Timothy; Matear, Richard. Constraints on the global marine iron cycle from a simple inverse model. Journal of Geophysical Research: Biogeosciences. 2016; 121:28-51.

Koeve, W., Duteil, O., Oschlies, A., Kähler, P., and Segschneider, J.: Methods to evaluate CaCO3 cycle modules in coupled global biogeochemical ocean models, Geosci. Model Dev., 7, 2393–2408, https://doi.org/10.5194/gmd-7-2393-2014, 2014.

Kriest, I.: Calibration of a simple and a complex model of global marine biogeochemistry, Biogeosciences, 14, 4965–4984, https://doi.org/10.5194/bg-14-4965-2017, 2017.

Kriest, I., Kähler, P., Koeve, W., Kvale, K., Sauerland, V., and Oschlies, A.: One size fits all? Calibrating an ocean biogeochemistry model for different circulations, Biogeosciences, 17, 3057–3082, https://doi.org/10.5194/bg-17-3057-2020, 2020.

Kvale, K. F., Meissner, K. J., and Keller, D. P.: Potential increasing dominance of heterotrophy in the global ocean, Environmental Research Letters, 10, https://doi.org/10.1088/1748-9326/10/7/074009, 2015a.

Kvale, K. F., Turner, K., Landolfi, A., and Meissner, K. J.: Phytoplankton calcifiers control nitrate cycling and the pace of transition in warming icehouse and cooling greenhouse climates, Biogeosciences, 16, 1019–1034, https://doi.org/10.5194/bg-16-1019-2019, https://www.biogeosciences.net/16/1019/2019/, 2019.

Landolfi A, Somes CJ, Koeve W, Zamora LM, Oschlies A. Oceanic nitrogen cycling and N2O flux perturbations in the Anthropocene. Glob Biogeochem Cycles. 2017;31:1236–55. https://doi.org/10.1002/2017GB005633.

Niemeyer, D. T. P. Kemena, K. J. Meissner, and A. Oschlies. A model study of warming-induced phosphorus–oxygen feedbacks in open-ocean oxygen minimum zones on millennial timescales. Earth System Dynamics, 8(2):357–367, 2017.

Niemeyer, D., Kriest, I., and Oschlies, A.: The effect of marine aggregate parameterisations on nutrients and oxygen minimum zones in a global biogeochemical model, Biogeosciences, 16, 3095–3111, https://doi.org/10.5194/bg-16-3095-2019, 2019.

Sauerland, V., Kriest, I., Oschlies, A., & Srivastav, A. (2019). Multiobjective calibration of a global biogeochemical ocean model against nutrients, oxygen, and oxygen minimum zones. Journal of Advances in Modeling Earth
Systems, 11, 1285– 1308. https://doi.org/10.1029/2018MS001510

Schartau, M., Wallhead, P., Hemmings, J., Löptien, U., Kriest, I., Krishna, S., Ward, B. A., Slawig, T., and Oschlies, A.: Reviews and syntheses: parameter identification in marine planktonic ecosystem modelling, Biogeosciences, 14, 1647–1701, https://doi.org/10.5194/bg-14-1647-2017, 2017.

Somes, C. J., A. Landolfi, W. Koeve, and A. Oschlies (2016), Limited impact of atmospheric nitrogen deposition on marine productivity due to biogeochemical feedbacks in a global ocean model, Geophys. Res. Lett., 43, 4500– 4509, doi:10.1002/2016GL068335

Watson, A. J. Oceans on the edge of anoxia. Science, 354(6319):1529, 12 2016.

Wanxuan Yao et al 2019 Environ. Res. Lett. 14 114009

---

## Author Comment (AC1) · 8 Oct 2020

Thank you for a thorough and clear review of our paper. Here we will respond in brief to your comments and describe how we will subsequently revise the paper, prior to a full response to referees after the editor's decision.

We recognise that we need to provide a fuller description of the model in the revised manuscript, which Reviewer #2 also picked up on. Although the model is described in detail elsewhere (and for simpler studies citing those may be sufficient), given our intended audience is a wider selection of Earth system model users and those more generally interested in climate feedbacks we agree that it would be useful to provide more

model details. We will focus particularly on physical aspects, hard pump processes, represented ecological interactions between the different plankton size classes, and the limitations these factors introduce. We will also emphasise more the limitations that the low-res physical representation imposes in the Discussion and Conclusions.

You are right to say that the physical climate change response of c/ecoGEnIE is important context to interpreting the response of the biological pump and ocean carbon sink in comparison to other models. We will include details of this at the start of the Results section in the revised manuscript. We will also make clear in the Methods that our experiments are emissions-driven rather than concentration-driven (so as to allow carbon cycle feedbacks to fully emerge). We can also confirm that in this study we quantify the biological pump as the POC export from cGEnIE's surface box (with its base at ∼81m depth), which we indicate in lines 195-197 but can further emphasise (and discuss the limitations of) in our revisions.

Both yourself and Reviewer #2 asked for more context in the way of comparing our results with existing ESM/EMIC projections, which we will include in our revisions. We will also add graphs of the long-term carbon sink results beyond the 21st century cumulative to the supplementary material (along with graphs for the biological pump for RCPs 3PD/2.6 and 6.0 in addition to 4.5 and 8.0 in the main manuscript), which we left out of the initial submission for reasons of brevity. We will also present results directly showing shifts in plankton size distribution during the ECO configuration runs, which Reviewer #2 requested as well.

We recognise that our wording on the importance of the biological pump for climate feedbacks and the implications of our results (as "critical") gave the wrong impression. We of course accept that the solubility pump is the dominant factor in the ocean carbon sink (as stated on line 54) with biological processes are of second order importance, and that our carbon sinks results show relatively minor changes (<∼1%) in comparison to the biological pump (<∼10%). We believe our results help illustrate that representing ecological dynamics relating to metabolism and size classes is important (even if

not dominant) for biological pump projections but less so for the overall ocean carbon sink, and we will clarify this in the revised manuscript. However, we do not agree that the example given of a typically bombastic statement ("Additionally, NPZD-type models cannot fully resolve the potential impact of climate change or ocean acidification on ecosystem structure, biodiversity, and plankton size shifts.") does overstate the situation – given that allometric effects are shown to be important for climate impacts on plankton community structure and function, it seems relatively uncontroversial to state that models that don't represent allometric effects will be limited in resolving climate impacts on ecosystem structure and function. It was definitely not our intention though to imply that the model we use here is not limited in any way, or that other models are fatally limited in comparison. We will carefully review the manuscript for unclear or missing statements on respective model limitations, and will clarify or rephrase these instances as necessary.

Regarding the minor comments, we will provide clarifications in the revised manuscript where relevant, including exploring rearranging table 2 to improve its clarity, rephrasing the statement on past hard pump model development, and revising the flagged supplementary figures and captions.

---

## Author Comment (AC2) · 8 Oct 2020

Thank you Jamie for a thoughtful and clear review of our paper. Here we will respond in brief to your comments and describe how we will subsequently revise the paper, prior to a full response to referees after the editor's decision.

We recognise that we need to provide a fuller description of the model in the revised manuscript, which Reviewer #1 also picked up on. Although the model is described in detail elsewhere (and for simpler studies citing those may be sufficient), given our intended audience is a wider selection of Earth system model users and those more generally interested in climate feedbacks we agree that it would be useful to provide

more model details. We will include saturation state-dependent PIC:POC ratio and size-dependent DOC:POC production. We will also provide additional background discussion of the role of POC remineralisation depth and ventilation processes in the operation of the biological pump. Thank you for pointing out that BIOGEM is a parametrised rather than an explicit NPZD-type biogeochemistry module – this was an inadvertent mistake and will be corrected in the revised manuscript, and the implications of resolving biomass explicitly discussed.

You are correct that in order to achieve equivalent POC export in the four configurations, some POC remineralisation parameters were altered, although we did not change the remineralisation depths (and so the potential depth timescale bias described doesn't directly apply). Instead we changed the proportion of recalcitrant POC (increasing it in the ECO configurations) and to a lesser extent the PIC:POC ratio, with the aim of equivalent baseline POC & PIC export across all four configurations and as similar carbonate chemistry as possible. Recalibrating the setups to have as similar a carbon cycle as possible was pursued in order to make the results easily comparable across the configurations, while POC export was chosen as the primary calibration constraint as the main variable being analysed. However, we recognise that we weren't clear as to how and why the model was recalibrated or how the calibration choices may limit the results, and this along with revised supplementary plots of the model-data fit for each calibration will be made clearer in the revised manuscript. Short Comment #1 also brings up a similar issue on whether difference in [CO3] across the setups are affecting our ocean carbon sink results, which will also investigate in our revisions. We can present the results of existing uncalibrated/published configurations for BIO+FPR, BIO+TDR, and ECO+FPR along with a recalibrated ECO+TDR in order to illustrate the impact of the POC export calibration relative to the changed ecological dynamics (which we believe will be relatively small).

We also recognise that as critical elements to the paper our explanations of the mechanisms proposed to drive both the biological pump and ocean carbon sink responses

could be clearer. In the revised manuscript we will expand and clarify these explanations utilising extra figures and simulations where necessary, as well as investigate and explain the potential role of additional mechanisms such as zooplankton grazing. As a brief specific response, on line 218 we do indeed mean that we believe more POC is remineralised in the surface layer and so initially lowers export production, the impacts of which we will clarify and further justify in the revised manuscript. While we do not directly discuss the impacts of representing stoichiometry or ocean acidification on our results they are implicit (for example, the latter in our discussion of the mechanisms behind ocean carbon sink changes, line 268-280). However, in our revisions we will explicitly discuss their role in our results. Both yourself and Reviewer #1 also asked for more in the way of comparing our results with existing ESM/EMIC projections in order to provide additional context along with results directly showing shifts in plankton size distribution, which we will include in our revisions as well.

Regarding the minor comments, we will provide clarifications in the revised manuscript where relevant, including clarifying terminology in the Introduction, rephrasing the statement on past hard pump model development, and exploring a reorganisation of the Figures.

---

## Author Comment (AC3) · 8 Oct 2020

Thank you for your comment on our paper. Here we will respond in brief to your comments and describe how we will subsequently revise the paper in response.

You are correct to note that DIC and ALK are not identical across our four configuration baselines, which is the result of prioritising calibrating equivalent POC & PIC export (the logic of which we explain in response to Reviewer #2, and will make clearer in the revisions) and the difficulty of exactly calibrating multiple variables. The uncalibrated ECO configuration has significantly different surface carbonate chemistry than BIO+FPR (with much lower [CO3] of $\sim$70-80 $\mu$mol/kg and higher DIC of $\sim$2280

$\mu$mol/kg), which we endeavoured to calibrate as closely as to the BIO+FPR configuration possible to minimise its impact (while prioritising POC & PIC export, hence the remaining difference). We also note that despite these differences ECO+TDR has a greater carbon sink reduction than ECO+FPR relative to BIO+FPR in most scenarios despite the larger mismatch in surface [CO3] of ECO+FPR, and similarly that ocean carbon sink capacity under ECO+FPR increases relative to BIO+FPR in three scenarios despite its lower buffer capacity, which indicates that surface [CO3] is not the primary factor in the relative ocean carbon sink differences for at least the ECO configurations. However we recognise that the remaining differences still introduce a confounding factor to our ocean carbon sink results. In our revisions, this needs to be acknowledged and constrained. Reviewer #2 brought up a different issue with the calibration process (relating to POC remineralisation parameters) which in response to we will revisit uncalibrated or recalibrated configurations to assess the impact of different calibration choices. For example, the initial BIO+TDR configuration has surface [CO3] very close to BIO+FPR ($\sim$105.55 vs 105.53 $\mu$mol/kg) despite higher POC export ($\sim$8 PgC), and so we can use simulations with this uncalibrated configuration to constrain the impact of surface [CO3] on our BIO+TDR results.

We recognise that our wording on the importance of the biological pump for climate feedbacks and the implications of our results (as "critical") gave the wrong impression. We of course accept that the solubility pump is the dominant factor in the ocean carbon sink (as stated on line 54) with biological processes are of second order importance, and that our carbon sinks results show relatively minor changes ($<\sim$1%) in comparison to the biological pump ($<\sim$10%). We also do not believe this study to be the first to propose more complex biogeochemistry in ESMs or to assess the inclusion of temperature-dependent remineralisation or plankton size traits in an ESM or EMIC (although we do believe we provide novel insights on the interaction of these factors). Our focus on CMIP relates to their dominance in ocean carbon sink and climate feedbacks discourse, but in our revisions we will expand the Background to include more recent biological pump research in EMICs and ESMs as well to put our study in a wider

context.

An important aspect of this study is the separation and the nonlinear relation of the biological impact and the ocean carbon sink, but we recognise this could be made clearer in the Background. In the revisions we will further clarify that the biological pump is not itself a carbon sink, but that a transient shift in the remineralisation depth can change the partitioning of carbon within the ocean and so indirectly affect the ocean carbon sink (as per e.g. Kwon et al, 2009: Nat. GS). We will also clarify in our revisions that the nitrogen cycle, redox-dependent feedbacks, and ballasting may also have significant impacts on the biological pump, and their exclusion in this study is therefore an additional limitation on our results.

---

## Author Response (AR1)

**Authors' response to reviewers' comments on "*Resolving ecological feedbacks on the ocean carbon sink in Earth system models*" (Manuscript reference: esd-2020-41)**

Dear Prof. Crucifix and Reviewers,

Thank you for your reviews of our manuscript "*Resolving ecological feedbacks on the ocean carbon sink in Earth system models*". Please find below our detailed responses to the reviews. We include the original comments and our response under each point in **bold** text and with line numbers referencing the revised manuscript with changes marked.

We hope we have sufficiently answered your queries in our response.

Yours sincerely,

David I. Armstrong M^cKay & co-authors

**Response to Reviewer 1 Anonymous (R1):**

Major comments and assessment

I have a number of minor points across the manuscript and supplementary (see below), as well as several major comments about the study:

1. Considering that this is a study focused on export production, the authors skip over how exactly this is defined in their model, and how this relates to what other researchers conventionally understand by the term. More generally, the paper's analysis largely avoids what the different models and parameterisations mean for the patterns (geographical, temporal) and magnitudes of export flux, and even how this varies with depth. The major statistics chosen for analysis deal simply with export at some undefined horizon and with how this affects air-sea exchange, and doesn't really look at spatiotemporal patterns of ocean DIC content (e.g. depth distributions).

**POC export was defined (on old ln.195-197) as the POC flux at the bottom of cGEniE's surface layer at ~81m depth, but we have made this clearer in our revisions (pg.10 ln.334). We also describe how due to ecoGEnIE's provisional calibration against observational data we focus on the global rather than spatial response (old ln. 170-172), which we have also clarified (pg.9 ln.316).**

2. The authors do not adequately describe either the physical or biogeochemical models that they are using. Given this study is focused on the vertical transfer of organic (and inorganic) material, I would expect some description of the physical domain (e.g. vertical grid), and reassurance that important vertical processes are represented (e.g. mixing). Similarly, key results appear to depend on model parameterisations of calcification, but details of this are entirely omitted. These omissions make it difficult to determine the validity of some of the conclusions reached. (Even the precise nature of the scenario simulations, e.g. emissions- vs. concentration-driven is not made clear.)

& 4. While deficiencies with models are always inevitable, it is important that these are addressed in manuscripts. The noted lack of model description and evaluation are joined by a lack of appropriate discussion and caveats. Combined with the authors strong (over-confident?) conclusions, these absences make it difficult for readers to accurately assess the significance of this manuscript.

**We recognise that a fuller description of the model was required, which R2 also picked up on. Although the model is described in detail elsewhere (and for simpler studies citing those may be sufficient), given our intended audience is a wider selection of Earth system model users and those more generally interested in climate feedbacks we agree that it is useful to provide more model details. In our revisions we included more model details on physical aspects, hard pump processes, the ecological interactions between the different plankton size classes represented, and the limitations these factors introduce (pg. 7 ln.212 – pg.9 ln.284). We also emphasise more the limitations that the low-res physical representation imposes in the Discussion and Conclusions. (pg.17 ln.543)**

3. On a related point, the manuscript's evaluation of model performance (physical and biogeochemical) is insufficient. There is nothing on how physically realistic the preindustrial and future scenarios are; for instance, how aspects such as the magnitude and pattern of temperature and mixing change compare to other more physically-realistic models. These are of key importance to the biogeochemistry models used here, and specifically to the temperature and nutrient responses investigated.

**You are right to say that the physical climate change response of c/ecoGEnIE is important context to interpreting the response of the biological pump and ocean carbon sink in comparison to other models. We have added details of this at the start of the Results section (pg.11 ln.340 – pg.12 ln.350), and also make clear in the Methods that our experiments are emissions-driven rather than concentration-driven (so as to allow carbon cycle feedbacks to fully emerge) (pg.11 ln.330). We also confirm that in this study we quantify the biological pump as the POC export from cGEnIE's surface box (with its base at ~81m depth), which we indicate in ln.334.**

5. When presenting its key findings of change in export production between the various combinations of model and parameterisations, the manuscript does not any provide contextualising information. How, for instance, do the modelled changes compare to those of the CMIP models mentioned elsewhere? Are the changes found here within or outside the ranges found by these other (less sophisticated) models? This omission, in particular, makes it difficult to understand this work in context. Table 1, for example, could be expanded to include such metrics from CMIP5.

**We have provided additional context of past biological pump (pg.12 ln.363-366) and carbon sink projections (pg.17 ln.533-536), which show our results as within a similar range. However, despite the many papers available on CMIP5, relatively few provide directly equivalent numbers of POC export decline by 2100 under various scenarios (rather than a baseline strength across the late 20[th] century for intercomparison, for example), and estimates of the specific effect of the biological pump on ocean carbon sink projections are relatively lacking.**

All that said, the manuscript is otherwise relatively straightforward to understand and follow. And I am fairly confident that their findings are accurate in identifying certain processes as important, and therefore relevant to both observational and modelling scientists. However, it remains the case that the manuscript's omissions (both in background and analysis) undermine its credibility, and I recommend major revisions before it can be accepted for publication.

Minor comments

Ln. 28: saying "gradually" in the same sentence as "centuries to millennia" might convey the wrong impression; I might be inclined to just delete "gradually"

**Clarified (pg.1 ln.28)**

Ln. 32: what do you mean by "sink feedbacks"?; this could be enhanced stratification decreasing mixing and / or nutrient supply; be clearer please

**Briefly clarified, but as this is an Introduction there isn't enough space to fully describe the contents of every cited paper (and in this particular case the method they used doesn't pinpoint the specific processes responsible). (pg.2 ln.32-33)**

Ln. 40: what about AR6?; quite a lot of model output is available, together with a lot of supported information about model formulation; you might also wish to consult doi:10.1007/s40641-020-00160-0 or doi:10.5194/bg-17-3439-2020 on AR6

**Much but not all CMIP6 output is indeed already available. Acquiring and analysing all these results would be a whole study in itself (and one that is likely being led by CMIP6 participants themselves, not yet published). Our focus is on the representations within the models, rather than the intercomparison of outputs; however, we have added more mention of CMIP6 and its associated model improvements to make this point clearer. (pg.2 l.44, pg.7 l.201-207)**

Ln. 52: a "this manuscript is structured as follows" (or similar) might help here

**Added (pg.2 ln.59-63)**

Ln. 55: erm, what about a straight decline fed by a decreased amount of production?

**We get to this in more detail later as well, but we have clarified this here. (pg.3 ln.67)**

Ln. 60-61: I seriously doubt this; there is a large body of work focused exclusively on the soft pump whereas there is far less focus on the hard and silicic acid pumps; in part because, unlike the soft pump, the activity of these latter pumps is less ecologically driven and more physiochemically-driven (cf. the importance of the CCD)

**We originally intended to mean specifically within high-res ESMs (rather than broadly including better developed subsystem models and EMICs as well), but this was poorly phrased. Given its low importance to our case, we have simply removed this statement. (pg.3 ln.80)**

Ln. 72: note that Kwiatkowski et al. (2014) describes 6 models, of which only 1 has a single size class; the rest have at least 2 size classes

**Kwiatkowski et al. by their own description compare both NPZD and PFT biogeochem models, with the single size class model representing NPZD models. However, we agree that an additional reference describing NPZD models is useful here. (pg.4 ln.101)**

Ln. 74-76: as opposed to some other "class of plankton model" which is not limited ...

**We are of course aware all models are limited, and here we refer to the specific limitations of this particular class of model (pg.4 ln.104). Elsewhere in the text we have endeavoured to make the limitations of this study's models more explicit to make it clear that we don't believe our models to be not limited (e.g. pg.9 ln.273-288).**

Ln. 84: can you give an idea why such evidently superior (as described here) models still fail to capture BGC and "large-scale" dynamics?; do you mean physical resolution problems?

**Clarification added (on how these ecosystem models have not yet been incorporated in to ESMs, hence they don't allow ES feedbacks to emerge). (pg.4 ln.122)**

Ln. 94: "shoaling" rather than "raising"?

**Clarified. (pg.5 ln.144)**

Ln. 94: "the point at which most POC is remineralised" - is this a formal definition you're using in this study?; it might be good to give it a name (e.g. R_50, or something) if so

**We don't directly measure this depth in this study and definitions in the literature somewhat vary, but we have clarified this using the most robust definition. (pg.5 ln.144)**

Ln. 95: at some point it might be nice to demonstrate that faster remin. leads to greater export despite the remin.; maybe that's yet to come

& Ln. 98-99: cf. my previous remark, you'll need to demonstrate that faster remin. doesn't act as a *positive* feedback mechanism (i.e. warming -> faster remin. -> less C export -> more atmospheric CO2 -> more warming -> …)

**This is part of the aim of the study to explore, and is described in more detail later (pg.13 ln.380-401). In the background we are first outlining how this is possible, before demonstrating it later in the results. We now explicitly mention the potential for this to be a positive feedback as well (pg. 5 ln.150)**

Ln. 107: "will not substantially affect productivity in existing oligotrophic regions" - is this actually important?; the important aspect of oligotrophication is that formerly productive regions of the ocean become less productive; that unproductive regions remain equally unproductive is not obviously important

& Ln. 107: "the depth rather than the intensity" - my understanding is that most research already focuses on the depth and not the intensity of stratification; usually studies focus on change in mixed layer depth - key for resetting surface nutrient conditions during periods of deeper seasonal mixing (e.g. winter)

**We have clarified this sentence to more simply state oligotrophication affects productive rather than already oligotrophic regions, and have refocused the second part on a more relevant aspect. (pg.5 ln.161 – pg.6 ln.165)**

Ln. 111: "result in ocean interior deoxygenation" - as does warmer surface temperatures that control the oxygen concentration of water ventilated into the interior

& Ln. 111: "reduce nitrogen availability" - are you referring to denitrification here?; you should be specific if you are; otherwise one might assume you mean reduced nitrogen (and other nutrients) from reduced mixing

**Clarified. (pg.6 ln.168-169)**

Ln. 114-117: it might be helpful to note the significant uncertainties in OA feedbacks on marine BGC; something like doi:10.1146/annurev-environ-012320-083019 might help here

**Clarified. (pg.6 ln.178)**

Ln. 116: "by which POC sticks to denser falling PIC" - this kind-of reads as POC is somehow "attracted" to CaCO3; instead, the ballast hypothesis is more about POC *already associated* with CaCO3 somehow being "protected" by it to reach further into the ocean interior

**Clarified. (pg.6 ln.176)**

Ln. 124: "IPCC AR5" - as already mentioned, the archive of simulations used in AR6 is already quite well-stocked with models

**As stated in response to the comment for old l.40, much CMIP6 output is indeed already available. We have added more mention of CMIP6 and its associated model improvements both earlier in the text (pg.2 l.44) and after this paragraph (pg.7 l.201-207) to make this point clearer.**

Ln. 130: "(Bendtsen et al., 2015; Dunne et al., 2007; Martin et al., 1987)" - this list of citations is ambiguous; are these submodels of remineralisation used in specific models in Table 1 or what?

& Ln. 130: Given that Dunne et al. invokes a ballast model that dynamically alters the remineralisation profile, this characterisation seems inaccurate

**Here we aimed to cite the general concept of observed remineralisation profiles being approximated by the Martin curve, which Dunne 2007 also describes and Bendtsen 2015 presents updated empirical measurements of. However, we can see that this is not clear from the wording, so we have expanded this statement and attach these citations to the sub clause on observations. (pg.6 ln.192)**

Ln. 131: this statement is characteristic of many in this manuscript; i.e. bombastic on how deficient existing models are; given this manuscript is using a reduced-physics ocean model with extremely poor vertical resolution (<= 16 levels; it's unclear, see below), a more measured toned might seem appropriate

**We do not agree that this statement is overly bombastic – given that allometric effects are shown to be important for climate impacts on plankton community structure and function, it seems relatively uncontroversial to state that models that don't represent allometric effects will be limited in resolving climate impacts on ecosystem structure and function. It was definitely not our intention though to imply that the model we use here is not limited in any way, or that other models are fatally limited in comparison, and have clarified this throughout the manuscript. (pg.6 ln.195)**

Ln. 141-154: this description must be augmented by a description of the underlying physical model so that readers understand that it imposes its own limitations on this study; while pointing to other uses of the model is fine, a minimal description that notes model resolution (horizontal, vertical, temporal) and the reduced physics nature of this model is key; noting what the model does around processes relevant to export production (e.g. mixing, convection) would help readers unfamiliar with this particular EMIC

& Ln. 151: "have lower spatiotemporal resolution" - cf. my previous remark, expand on what's meant by "lower"

& Ln. 151-154: ": : : well-suited for investigating more complex biogeochemical dynamics : : : " - only if the physics they represent is up to the job, of course; and that may depend on the specific application; a reader should be interested in how reduced vertical resolution impacts a model of vertical POC transfer

**We recognise a fuller description of the model was required, which R2 also picked up on. Although the model is described in detail elsewhere (and for simpler studies citing those may be sufficient), given our intended audience is a wider selection of Earth system model users and those more generally interested in climate feedbacks we agree that it is useful to provide more model details. More detailed physical model description has now been included, including on physical aspects, hard pump processes, ecological interactions between the different plankton size classes, and the limitations these factors introduce. We also emphasise more the limitations that the low-res physical representation imposes in the Discussion and Conclusions. (pg. 7 ln.212 – pg.9 ln.284, pg.17 ln.543)**

Ln. 155: the history of the models can probably be skipped to focus on what's being used here; unless it's important for this particular study (which it might be)

**We feel that it's useful to have some brief model history here for readers who might want to replicate or follow-up to give specific version, and is also useful for background of what the model has been used for before. (pg.7 ln.217, pg.8 ln.243-246)**

Ln. 161: what's the relationship between the size classes?; does each predator graze everything smaller than it is, or is there another scheme?

& Ln. 161: it would be helpful to have some idea (e.g. a sentence or two) on how the different size classes differ from one another; e.g. photosynthesis parameters, nutrient uptake, growth rates, grazing rates, mortality rates

**Sentences added on physiological and ecological processes represented and their temperature/size-dependence. (pg.8 ln.249 – pg.9 ln.265)**

Ln. 168: "16 layers" - ah, finally!

**We have introduced a more detailed description of cGEnIE's physical setup earlier in the Methods. (pg.8 ln.229-244)**

Ln. 169: does the model resolve seasonal mixing of different ecological regimes?

**No, but insolation and light attenuation is seasonal, and ecoGEnIE does reproduce seasonal variation in primary production. (pg.8 ln.249 – pg.9 ln.263)**

Ln. 170: is there information about what "not quite as well" means?; e.g. which properties were examined to determine this?; ones directly relevant to this study, or more indirect?

**Assessed properties added, along with more justification. (pg.9 ln.285)**

Ln. 176: ah-ha; we do need to know about BIOGEM; good!

**We have added more explicit mention of BIOGEM earlier in the Methods. (pg.8 ln.235)**

Ln. 186: "POC export" - how is this defined?; it is the flux at a particular depth, or just out of the model's uppermost level?; in cGENIE this can be quite deep ( 80m?), but it may not conform to field norms; e.g. Martin et al. (1987) used 100m

**This was defined at end of paragraph, but have moved it up for clarity. (pg.10 ln.304)**

Ln. 186: "Gt C" - The SI unit is Pg C rather than Gt C

**GtC is common in the literature and is easier to grasp for many readers, but we have updated to PgC anyway. (pg.10 ln.304& throughout)**

Ln. 191: "shorter sub-overturning timescales" - this is opaque; I presume you mean that, as this model doesn't include sediments, DIC is conserved within the ocean regardless of export production (which might not be true if it were "lost" to sediments); but I don't understand the reference to timescales; not least because a 10,000 y simulation should be enough for several complete overturnings of the ocean

**Clarified, with reference to transient nature of pump perturbations being critical relative to the overturning timescale. (pg.11 ln.324)**

Ln. 193: "total CO2 emission scenarios" - is the model being run in emissions mode (i.e. a time-varying amount of CO2 is being added to the model atmosphere and then redistributed, including into the ocean) or in concentrations mode (i.e. a time-varying atmospheric concentration is specified but cannot be affected by different ocean uptake responses); make this clear

**Clarified. (pg.11 ln.330)**

Ln. 193: this list of scenarios seems to omit the low scenario RCP 2.6; any reason?; this unlikely, low emissions scenario (as SSP126) is still used in CMIP6

**3PD is what was available as long-term extended RCPs from http://www.pik-potsdam.de/~mmalte/rcps/ and represents a realistic low emission scenario. While 2p6 is more common for 21st century projection comparisons, we wanted to also assess the longer-term multi-centennial impact, and so used the available extended datasets – we have clarified this in the text. (pg.11 ln.331)**

Ln. 193: "3PD, 4p5, 6p0, and 8p5" - these scenarios are more typically referred to as e.g. RCP 8.5, where the period is a decimal point

**Typically yes, but using a 'p' instead is not uncommon (for example in discussions of the IPCC's SR1p5 report) and was useful for file naming systems and for aesthetic reasons (avoiding excessive periods) in some sentences. We have updated this for clarity though. (pg.11 ln.330)**

Ln. 199: do you evaluate the performance of your modelled warming relative to other models (e.g. CMIP5, CMIP6)?; as the marine BGC models you're using are sensitive to warming, it would be useful to know how

realistic this is in the model; it should be straightforward to compare model output to, say, corresponding CMIP5 / CMIP6 output; e.g. change in magnitude / pattern of ocean temperature, mixing, etc.

& Ln. 200: cf. my last remark, maybe say something here about what happens physically in your scenario simulations

& Ln. 200: you should also perhaps begin by discussing how the different models represent the pre-industrial situation; some of this is covered (I think) in the supplementary material, but I have comments there too

**Global and sea surface warming by 2100 as well as preindustrial baselines is now explicitly stated and compared with CMIP5 (pg.11 ln.341 – pg.12 ln.350). However, although some preliminary CMIP6 numbers are available and compare favourably (~2.1C, ~3.3C, ~3.6C, and ~5.5C in CarbonBrief's-summary of the CONSTRAIN project's output so far), they are not fully published and so cannot be easily referenced here. There is also no direct mixing output available in cGEnIE, so this cannot be plotted here.**

Ln. 217: "0.3%" - I've mentioned over at the table itself, but you might like to add such relative stats there

**This (and other %s by 2100) was from Figure 2, and only represent the biological pump strength at 2100 exactly relative to preindustrial. In contrast, Table 2 summarised the cumulative POC export over the entire 21st century, and so the former %s did not map on to the latter numbers. However, we have added the % deficit difference to preindustrial and default BIO+FPR scenarios to Table 2 as extra useful context.**

Ln. 221: how is the mixed layer handled in cGENIE?; older versions of the model don't really have a mixed layer

**I was using mixed as a synonym for surface – have clarified (pg.13 ln.391) and discussion of mixing has been included in the model description (pg.8 ln.234).**

Ln. 223: do you really mean "new production" here?; the distinction may be confused by changes in mixed layer depth itself across the scenario period

**We do mean new (rather than regenerated) production here as PO4 from layer 2 is allochthonous to layer 1, but should have specified surface rather than mixed layer (the former of which is fixed in cGEnIE, but the latter can vary). We have edited these sentences to make the role of new vs. regenerated production clearer. (pg.13 ln.392)**

Ln. 268: this part of the manuscript is confusing without some clarity on how the hard pump works in this model; both in terms of CaCO3 production (and controls on this), and how it dissolves down the water column

**We have clarified the hard pump in the Methods section and added some clarification here as well. (pg.16 ln.495-509)**

Ln. 277: it's difficult to tell, but this just sounds like the two models differ in the strength of their hard pumps (probably relative to the soft pump), with the result that they have different hard pump changes relative to soft pump ones into the future; and because the models are not well-described on this point, it's hard to decide what's going on

**In current versions of cGEnIE/ecoGEnIE PIC production is simply set as a ratio to POC production (subject to saturation state-dependence), so a decrease in POC production automatically results in a decrease in PIC production. Increasing acidification reduces PIC production further, but this works synergistically with the effects described here (as OA from increased pCO2 helps reduce PIC prod further). We have clarified the hard pump in the Methods section (pg.8 ln.238) and added some clarification here as well. (pg.16 ln.495-509)**

Ln. 282: why "initial"?

**Because of the need for follow-up to deal with limitations – admittedly potentially confusing though, so we have removed this. (pg.17 ln.529)**

Ln. 282: "the importance of incorporating multiple dimensions of ecological complexity" - the paper doesn't really present anything concerning the modelled ecological complexity; passing comment is made on shifts in size structure, but nothing is shown, not even as supplementary information

**Addressed above with plankton size figure (Supp. Fig. S50). Resolving multiple size classes within phyto- and zooplankton represents a clear increase in ecological complexity though, and so we have kept this statement.**

Ln. 284-287: the modelled change in export production is presented without any contextualised reference to other work on this; there is a large body of published work on how export may change, ranging from studies using individual models through to meta-analyses using, for instance, CMIP output; to make clear the significance of the distinction the authors are highlighting, the existing span of estimates needs to be clear

**We have added more context for past projections of POC export decline in section 4.2. However, here we are talking about the overall impact on the ocean carbon sink capacity, which as explained in section 4.3 is separate to POC export changes. There are fewer quantifications available for impact of warming on the carbon sink specifically due to changes in metabolism and ecology, with the main existing estimate from Segschneider & Bendsten given here and the current annual sink given for context too. (pg.17 ln.534-537)**

Ln. 286: "a much simpler NPZD-based ecosystem" - much simpler, perhaps, but possibly different in an important way for the hard pump

**This is discussed in more detail later in the Discussion, but some clarification added here. (pg.17 ln.536)**

Ln. 307-309: I wouldn't necessarily expect TDR to increase the production of diatoms because opal is dissolved and not remineralised; if anything, one might expect diatoms to do less well as time passed because - although N and P might be becoming more available (because they're getting remineralised faster) Si would not be; there's probably some subtlety I'm missing here, however; expand to make clearer

**This issue of surface Si becoming depleted leading to diatom decline is tackled in the next sentence. We have already explained in detail here the results of another study, and feel that it is already clear enough that if a reader wants to know they can consult the citation itself. We have clarified that this is from another model study though. (pg.18 ln.569-573)**

Ln. 311: "a subsequent increase in calcifying plankton and PIC export" - this result may be quite dependent on how calcifiers are modelled; this already isn't clear in this study, so I'd suggest drawing the parallels out more fully

**This is referring to the modelling of Segschneider & Bendtsen rather than our own modelling, and for full details of how exactly they represent calcifiers the reader is referred to that citation. As ecoGEnIE currently has no independent calcifier or silicifer classes yet it is not yet possible to draw parallels out fully – the point being made is that other studies have shown that this element is important, and so in future ecoGEnIE should include this too. However, we have added a subsequent sentence to make this point clearer. (pg.18 ln.575)**

Ln. 314: "which we have shown is critical" - I don't think this has been clearly shown here; the model is too incompletely described, and the physical model probably leaves something to be desired

**Although ecoGEnIE is indeed physically limited – which we have no clarified in the Methods section along with a fuller model description – our result of up to -10% difference from including size traits versus TDR is sufficient to say that allometry is important for understanding the biological pump response. However, we have adjusted the terminology (from "critical" to "important") to reflect that it's not the only important factor. (pg.18 ln.578)**

Ln. 320: there are no caveats in this paper about the quality of the model, physical or biogeochemical; the only caveats seem to relate to making the biogeochemical model even more complex without any consideration of whether the physical framework is adequate; work has emphasised the potential importance of physical frameworks for BGC models, e.g. doi:10.1016/j.pocean.2009.10.003

& Ln. 320: things left undiscussed include: 1. how realistic is this model's response under climate change (e.g. pattern and magnitude of temperature change; compare with CMIP5 / CMIP6); 2. can this model realistically represent mixing; 3. can this model realistically represent vertical gradients of properties given grid cell thicknesses (even close to the surface); 4. how dependent are results on (undescribed) hard pump submodel

& Ln. 320: some of the above points cannot easily be addressed here; but they should be properly acknowledged and discussed, and they should temper the conclusions drawn here; it may well be that these are accurate, but the physical and BGC models used here should give some pause for thought

**While we agree that clearer statements on the caveats and limitations of our model approach are required (and which we have now addressed and included elsewhere), we disagree that there were "no caveats" at all. However, we have now more fully discussed: 1) the models physical climate response (S4), 2) the model's representation of mixing (S3), 3) how well ecoGEnIE represents vertical gradients (S3), 4) the hard pump representation and how it affects our results (S3).**

Ln. 320: per my comment on Table 2, the different scenarios get pretty short shrift here; they're just stand-ins for different amounts of warming / emissions; it's not clear that they couldn't be thinned to small, medium and large warming

We use the four RCPs as they it is fairly standard practice to use them to enable direct inter-comparison with the results of other models, and they are also included within the model already making repeat runs by others with cGEniE easier. We have added extra context of the warming level associated with each scenario to the Results (pg.11 ln.342) and Table 2 though for extra clarity though.

Ln. 320: equally, it's not made very clear what the differences between the atmospheric CO2 concentrations across the scenarios mean for the ocean uptake numbers here; we should expect larger numbers for higher RCPs, but does efficiency of uptake of CO2 change (or is that too far for this paper?)

**The focus of this paper is how adding either ECO or TDR (or both) changes the biological pump and ocean carbon uptake within each scenario, rather than the differences between scenarios. Furthermore, although we do simulate differences in ocean carbon uptake on each scenario, as pointed out elsewhere they are relatively small compared to total uptake (pg.17 ln.531) and so the uptake efficiency would not change significantly. However, in order to satisfy another query on Table 2 (regarding ocean uptake %) we now report the partitioning of emitted carbon between ocean and atmosphere up to 2100, which also addresses the question of how CO2 uptake efficiency varies across configuration and scenarios.**

Ln. 322: "critical"? - while there may well be feedbacks such as those described here, I think the authors are arguably exaggerating their importance, especially given the magnitude of the numbers they find; I'd suggest "may be important" is more suitable wording

**We didn't mean 'dominant' by 'critical', just 'important', which we have clarified. (pg.19 ln.593)**

Ln. 328: "as expected" -> "than expected"?

**No, but clarified anyway. (pg.19 ln.599)**

Ln. 326-329: this sentence is too long to be parsed well; it's important so make it clear

**This sentence has now been split in two. (pg.19 ln.599)**

Ln. 333: "post-CMIP5 projections"? - you might need to explain what you mean by this

**This was referring to Segschneider & Bendtsen (2015) & Steffen et al (2018) – changed to broader "some model projections" and references added. (pg.19 ln.607)**

Fig. 1: "surface layer" = cGENIE's top box?

**Clarified. (pg.30 ln.946)**

Fig. 2: perhaps use different colours rather than linestyles to separate the four different models?; then save solid and dashed styles for the two scenarios

**We originally avoided too many colours for colourblind-friendliness reasons, but have re-plotted this figure using a new palette to avoid this issue. (pg.31)**

Ln. 651: it might be helpful to note the distinction you're drawing between "N/A", "No", and "No mention" here; also, assuming "no mention" means you couldn't find any reference to this in the model descriptions, have you considered contacting the model authors to ask?

**Distinctions clarified in footnote. (pg.38 ln.986)**

Table 2: while this sort of summary is of key importance, it might also be useful to see how these numbers change in time (beyond the single supplementary figure)

**Results through time are now fully provided via biological pump and ocean carbon sink results (Figures 2 & 5, and Supplementary Figures S47, S61, & S64).**

Table 2: it occurs that the manuscript does not clearly address the different scenarios; it might be better to reorganise so that the results are organised by model first and then by scenario; that way the span of results between scenarios (i.e. the effect on the properties for different degrees of warming) are clearer to see

**We believe it is more interesting to see differences between the different configurations, as that and not the overall impact of each degree of warming (which is covered elsewhere) is the paper's focus. There is also not much change in broad pattern across the different scenario within each configuration anyway, with each configuration yielding a similar % difference relative to BIO+FPR in each scenario. (pg.39)**

Table 2: this kind-of omits what happens for the period 1850-2000; it might not be important, mind

**We originally chose to focus on the 21st century as we're most interested in policy relevant timescale and the signal is small before 2000. However, we have updated the numbers to represent all cumulative change prior to 2100 to allow for easier intercomparison between biological pump and ocean carbon sink changes. (pg.39)**

Table 2: might it be useful to note what the changes in this table represent in relative terms as well (e.g. what's this delta as a percent of the total flux over this period?)

**Percentage differences have been added for each column relative to a relevant comparator (against no emission scenarios for cumulative POC export, total ocean-atmosphere carbon for ocean carbon sink, and BIO+FPR for relative POC export and ocean carbon sink changes for each configuration). (pg.39)**

Table 2: columns 4 and 6 - "default cGENIE" = "BIO+FPR", so perhaps just say that?

**Done. (pg.39)**

Table 2: thinking about the ocean uptake column, what about the efficiency of ocean uptake and how this varies with scenario and time?; this may require information about emissions (see my previous remark about emissions vs. concentration simulations)

**We have now included the corresponding atmospheric carbon buildup and the % of the total the ocean represents (NB: there is no explicit terrestrial biosphere in cGEnIE, so the ocean accounts for all natural carbon sinks). (pg.39)**

Supplementary material

Ln. 17: perhaps show the observational field as well so that the relative size of these errors is clear

**We have plotted surface maps and depth plots of observational data for DIC, ALK, & PO4 (Supp. Figs. S1-2, S7-8, S13-14) and in the captions compare the differences in subsequent plots with the observed mean in order to put these plots in observational context.**

Fig. S1: which time point is being compared here?; presumably near-present day given the choice of observational product

**These figures were all pre-industrial baseline – this has been clarified in the captions. (Fig. S3 and thereafter)**

Fig. S1, caption: when you say "surface" are you comparing the concentration in the uppermost layer of GLODAP with the uppermost layer of your model, or are you depthaveraging so that the intercomparison is fairer?; if not, you will need to explain why the intercomparison you're doing is the right one; this applies to alkalinity too (and nutrients if you plot them)

**The GLODAP data was re-gridded on the ecoGEnIE grid by Ward et al (2018) to allow direct comparison – this has now been clarified in the captions. (Fig. S3 and thereafter)**

Fig. S1, caption: why not write this as 2 mmol / kg?; ditto for the graphs; "E-0X" notation is a little annoying when we've got scientific prefixes available to us

**The E-0X notation is the default in the Panopoly software used to create these plots – we have now updated all supplementary figures to use the SI prefixes. (Fig. S1 and thereafter)**

Figure S3: this pattern looks interesting; is it salinity-related?; i.e. does it reflect a bias in model salinity?

**We have added a description of how this pattern may relate to salinity in the caption (cGEnIE has a similar salinity pattern to observations but with a curtailed range, so low-salinity regions are a bit more saline and high-salinity regions are a bit less saline). The overall impact is very marginal though. (Fig. S9)**

Fig. S3: alkalinity is usually given in equivalents rather than mols

**Alkalinity is expressed in mols in both the cGEnIE output and the re-gridded GLODAP data, and so we have maintained this for consistency. (Fig. S9 and thereafter)**

Fig. S3, colour scale: "1.00E-04" - see previous remark about units

**Done. (Fig. S9)**

Fig. S5: I'm assuming annual mean chlorophyll here; although I note that the model's Arctic is negatively biased to almost 1 mg chl / m3 - that implies quite a high annual mean observational chlorophyll; has it been time-averaged correctly?

**Clarified that this is annual mean chlorophyll. Chlorophyll is very high in the Eastern Arctic in the SeaWiFS data, so the difference here is correct as far as this dataset stands. (Fig. S13)**

Fig. S5: chlorophyll is not usually a brilliant metric to compare models to; I'd suggest using nutrients

& Fig. S5: also, you could compare to productivity; that possibly is even more relevant to the problem at hand

& Fig. S5, colour scale: I see the "milli" prefix is getting used here! ;-)

**Phosphate plots added (Supp. Figs S14-19) and SI prefixes used.**

Fig. S6: I presume these colour scales are being used for parity with the previous delta plots?; I understand that, but it might be more informative to use a more relevant colour scale to help readers delineate where models differ geographically

**Colour scales adapted throughout Supp. Figs to prioritise spatial pattern definition.**

Figure S19, colour scale: "5.8E-07" - as well as "milli", there is also "micro"

**Changed. (Fig. S49)**

**Response to Reviewer 2 Jamie Wilson (R2):**

In general, the concept of the manuscript is interesting given that few studies have approached the interactions between ecological and biogeochemical complexity due to computational limitations. The use of EcoGEnIE here facilitates this novel idea in a straightforward and logical way. The key results seem generally sound but there are significant parts of the results that are not backed up with figures and many explanations about the role of different processes that are not quantified. As such, the manuscript makes a good case for resolving ecological complexity but not necessarily which components of this complexity are important and why. The manuscript would benefit from major revisions including new figures and additional experiments to quantitatively show how the various components of the ecological complexity lead to the main results.

**We have now provided additional supplementary figures providing support for our process explanations, discussed in greater detail the role of particular processes such as grazing or stoichiometry, and have also presented simulations from alternative calibrations to control for potential calibration effects. (details below)**

General Comments

1) Calibration of global biological pump strength

The spin-ups are all calibrated to have the same global POC export ( 7.5 Gt C year- 1) but there are no details about how this has been achieved in the model, i.e., what parameters were modified. However the calibration has been achieved it needs to be described as the BIO+TDR and ECO+FPR set-ups now differ from their published versions (John et al., 2014. P3; Ward et al., 2018. GMD).

The supplementary plots comparing each set-up after calibration also need to be more comprehensive. It is not totally surprising that surface fields are similar across set-ups as POC export has been calibrated to the same global value, particularly for those fields that are strongly influenced by export such as PO4. These fields may then differ more in the ocean interior. The authors should add difference plots showing depth slices for the various fields and/or a Taylor diagram to show how the calibration affects global fit statistics like correlation and standard deviation.

The main concern I have is that the authors may have achieved the same POC export by altering parameters associated with POC remineralisation - because the BIO+FPR and ECO+FPR output in Table S1 should have the same POC sedimentation:export ratio if the fixed remineralisation profile is the same. Apologies if this is not the case, but if it is it may have implications for the results in the manuscript. Firstly, the differences in POC export in Figure 4 would be a combination of adding the various TDR and ECO components but also the calibration adjustments that presumably vary across set-ups to achieve the same POC export. (This is significant for any calibration). Secondly, if the ECO experiments have deeper remineralisation to offset the higher POC export in the Ward et al., (2018) set-up, this could potentially bias the results if the deep ocean takes longer to experience changes in temperature, i.e., the transient ECO response may be slower due to the calibration. The relative change in carbon/nutrient feedbacks may also differ because the residence time of carbon/nutrients in the ocean interior is different and because carbon/nutrients may be redistributed spatially via different

circulation pathways (e.g., Pasquier and Holzer 2016, JGR Oceans). While I don't think this changes the general findings of the manuscript, it does make me question the relative magnitude of changes between each set-up.

I do appreciate that the baseline states will aways differ in some way because of the use of different parameterisations! The authors need to acknowledge the reason for choosing to constrain POC export across runs and what issues this may introduce, e.g., are there differences in spatial export patterns; are you compensating for any errors in the circulation and biogeochemical model? Alternatively, the original BIO+FPR, BIO+TDR and ECO+FPR set-ups have all been (somewhat) calibrated to achieve similar global distributions of dissolved tracers compared to observations. The authors could repeat their experiments with these published set-ups and recalibrate just the ECO+TDR set-up to achieve similar global tracer distributions. This can be defined using various fit statistics like root mean square error. This alternative set of results would help demonstrate that the POC export calibration is not biasing the results.

**You are correct that in order to achieve equivalent POC export in the four configurations some POC remineralisation parameters were altered, although we did not change the remineralisation depths (and so the potential depth timescale bias described doesn't directly apply). Instead we changed the proportion of recalcitrant POC (increasing it in the ECO configurations) and to a lesser extent the PIC:POC ratio, with the aim of equivalent baseline POC & PIC export across all four configurations and as similar carbonate chemistry as possible. Recalibrating the setups to have as similar a carbon cycle as possible was pursued in order to make the results easily comparable across the configurations, while POC export was chosen as the primary calibration constraint as the main variable being analysed. However, we recognise that we weren't clear as to how and why the model was recalibrated or how the calibration choices may limit the results, and so this along with revised supplementary plots of the model-data fit for each calibration has been provided in the revised manuscript. (pg.10 ln.311 – pg.11 ln.324)**

**SC1 also brought up a similar issue on whether difference in [CO3] across the setups are affecting our ocean carbon sink results, which we also investigated in our revisions by presenting the results of existing uncalibrated/published configurations for BIO+FPR, BIO+TDR, ECO+FPR, and ECO+TDR in order to illustrate the impact of the POC export calibration relative to the changed ecological dynamics. (pg.11 ln.324-328, pg.16 ;.511 – pg.12 ln.526)**

2) Background and Model Description

The description of processes in the Background section and the model description is too brief to support the main results. The biological pump is described mainly in terms of export production but has little description of the role of POC remineralisation and circulation. A few statements describing that the POC flux rapidly decreases with depth to a small asymptotic flux by 1000m and that the ventilation age of the ocean increases with depth would really help clarify a lot of later statements in the results. Similarly, there is no basic description of the allometric relationships for plankton and how they relate to metrics like primary production.

The model description is also very sparse in specific details that would aid the reader in understanding the results in more detail. For example, important details such as the saturation-state dependent PIC:POC rainratio (Ridgwell et al., 2007: Biogeosciences) and the nature of allometric trends like size-dependent DOC:POC export

production (Ward et al., 2018: GMD) are not described. Whilst these are described fully elsewhere, it would help to describe these briefly as they are directly relevant to the results and discussion.

**We recognise that a fuller description of the model was required, which R1 also picked up on. Although the model is described in detail elsewhere (and for simpler studies citing those may be sufficient), given our intended audience is a wider selection of Earth system model users and those more generally interested in climate feedbacks we agree that it would be useful to provide more model details. In our revisions we include a more detailed model description, including information on state-dependent PIC:POC ratio and size-dependent DOC:POC production. (pg.8 ln.229 – pg.9 ln.271)**

**We also provide additional background discussion of the role of POC remineralisation depth and ventilation processes in the operation of the biological pump (pg.3 – pg.6), and clarify that BIOGEM is a parametrised rather than an explicit NPZD-type biogeochemistry module – this was an inadvertent mistake (pg.2 ln.56 & pg.10 ln.292).**

3) Results from the plankton ecosystem modelling

The description of how plankton ecosystem structure impacts the biological pump is difficult to follow (mainly lines 231 - 246 and other related sentences throughout). There are a lot of discussion of changes in plankton size but this is never visualised despite being a standard output of the model.

**A supplementary figure showing the shift in mean plankton size with warming in ECO+FPR has now been provided, along with changes in individual size classes for the more in-depth discussion of ecological responses. (Fig. S50 &S55-60)**

I am not totally convinced by the explanation of why the ecosystem model leads to a greater decrease in export production. Size structure, variable stoichiometry and DOC:POC export ratio are all alluded to throughout the manuscript but there are additional components that haven't been considered. In steady-state warm-climate experiments using the same model there is a net decrease in plankton biomass due to increased grazing pressure because grazing rates are temperature dependent in Eco- GEnIE (Wilson et al., 2018, Paleoceanography and Paleoclimatology). This grazing effect also co-varies with nutrient availability leading to distinct latitudinal trends in size, biomass and export. This needs to be factored into the explanation here. This is a novel application application of a model of this type so it would be really helpful and informative to know what aspects of the ecological complexity are crucial to this result!

& 4) Results from the Ocean Carbon Sink capacity

I found it hard to follow the logic in this section because the factors involved are not quantified and/or illustrated in figures. A figure illustrating the changes described would really help to clarify the text in this section.

The increase in export production but decrease in carbon sequestration has been noted before (Kwon et al. 2009, Nature Geoscience; Gnanadesikan & Marinov 2008, Marine Ecology Progress Series). The impact on carbon sequestration is in part due to a change in organic carbon cycling and in inorganic carbon cycling but It is

not clear in the manuscript what the relative impact of these processes are. This could be separated by running additional experiments with a uniform PIC:POC rain-ratio to remove the impact of any spatial differences in POC export between se-ups and a prescribed spatially variable ratio from the associated spinup to isolate the impact of changing saturation state.

**We recognise that as critical elements to the paper our explanations of the mechanisms proposed to drive both the biological pump and ocean carbon sink responses could have been clearer. We have expanded and clarified these explanations utilising extra supplementary figures (S50-60 & S62) and comparisons with previous applications of ecoGEnIE, including the role of additional mechanisms such as zooplankton grazing. (pg.13 ln.386 – pg.15 ln.450). We have also provided a supplementary figure (Fig. S63) illustrating spatial changes in the PIC:POC ratio, which as well as demonstrating ocean acidification also reveals that its decline is relatively uniform across the global ocean and so unlikely to have a significant spatial impact. Fixing PIC:POC at preindustrial pattern in an additional run could indeed help isolate and quantify the ocean acidification component, but given the representation of calcifiers and PIC is currently rather limited in ecoGEnIE we did not want to overanalyse this aspect, and instead kept the focus on overall synergistic impacts. Further work repeating these simulations with a stronger focus on OA when calcifiers and silicifiers are available in ecoGEnIE as well would be worthwhile.**

Specific Comments

Lines 18 - 20: the manuscript does not actually show plankton size or deal significantly with ocean acidification

**We have now more explicitly shown the impact of size classes in our results (pg.13 ln.407 – pg.15 ln.450; Supp. Fig. S50), de-emphasised OA in the abstract (pg.1 ln.19), and clarified the interactions with OA in our discussion of how carbon sink changes relate to biological pump changes (pg.16 ln.495-509).**

Introduction/Background: generally I found the structure of these sections difficult to follow. Particularly there are a number of concepts and abbreviations in the Introduction, such as Fixed Profile Remineralisation, that are not described sufficiently until the Background section.

**We've removed some of the specific terminology and made the Intro more generic, plus some restructuring in the Background. (pg.1 ln.25 – pg.2 ln.63)**

Line 51: cGEnIE does not have a NPZD model. It parameterises the export of production by plankton as a function of nutrient availability using Michalies-Menten kinectics and other limiting factors. This needs to be made clearer throughout the manuscript.

**Clarified. (pg.2 ln.56 & pg.10 ln.292)**

Line 54: "a weakening carbon sink" - w.r.t. anthropogenic climate change?

**Clarified. (pg.3 ln.65)**

Line 56: the biological pump is described too briefly here and focused very much on the export of organic matter from the surface. It would help readers to expand here on the additional role of depth variation in

remineralisation rates and ocean ventilation ages, particularly as this is a key concept needed to understand the model results.

**We intended the whole Background section to explore the biological pump (pg.3 – pg.6), and the statement here was intended more as a headline opener before exploring in more depth in later paragraphs (for example in the next paragraph on remineralisation and 4 paragraphs later on the effect of remin. depth shifts with climate change). However, we have added extra details to our description of the biological pump to make sure this is clear. (pg.3 ln.86 – pg.4 ln.96)**

Lines 60 - 61: this statement surprised me! There have been significant model developments that try to resolve the ecological drivers of the soft-tissue pump such as cell-size and aggregation (e.g., Jokulsdottir & Archer 2016. GMD; Omand et al., 2020. Scientific Reports). I am not sure we are at a stage where we fully understand the interactions yet or are able to couple these models into global biogeochemical models though.

**We originally intended to mean specifically within high-res ESMs (rather than broadly including better developed subsystem models and EMICs as well), but this was poorly phrased. Given its low importance to our case, we have simply removed this statement. (pg.3 ln.80)**

Lines 90 - 91: Strictly speaking it is the metabolic rates that increase between 100% and 200% whereas gross primary production and community remineralisation are additionally limited by other factors.

**Clarified. (pg.5 ln.140)**

Line 94: "raising the remineralisation depth : : : higher up in the water column" - this is repetition. Either the remineralisation depth moves higher up or it is raised.

**Clarified. (pg.5 ln.144)**

Line 94: "(the point at which most POC is remineralised)" - an e-folding depth is often used to define this as the depth at which 63% of the exported flux has been remineralised.

**Clarified. (pg.5 ln.144)**

Lines 106 - 108: please briefly outline why this happens

**This sentence has been simplified in response to R1 to clarify that oligotrophication has most impact in currently productive rather than already oligotrophic regions. (pg.5 ln.161 – pg.6 ln.165)**

Line 161: "better representation of biodiversity" - relative to what? If relative to cGE nIE then this is really just resolving diversity (and biomass!) compared to the export production parameterisation.

**We meant relative to models without any/many size classes – this has been clarified. (pg.9 ln.263)**

Line 175 & 179: NPZD here is misleading as the export production scheme in cGENIE does not resolve plankton biomass, phytoplankton or zooplankton.

**Clarified. (pg.10 ln.292)**

Line 204: though the biological pump strength does increase for the BIO+TDR experiments by 2100

**Only for 3PD (4p5 ends up net neutral by 2100, but this wasn't clear without a zero line) – we have now clarified this with an "almost" though. (pg.12 ln.357)**

Line 207: How does the 6.1% decrease (and generally across all experiments) compare with CMIP model simulations?

**Comparison added – the default BIO+FPR bio pump decline is in line with past CMIP5 and key EMIC simulations. (pg.12 ln.364-367)**

Lines 210 - 212: this is not an explanation of what is happening in cGEnIE as it does not resolve plankton and productivity is restricted to a single surface layer.

**Clarified, as the previous explanation was too general. (pg.12 ln.371-376)**

Line 218: "more POC is being remineralised with warming" - I struggled to follow the logic of this. Does this mean more POC is remineralised in the surface ocean so lowering export production? If so, this should be checked that POC remineralisation occurs in the surface grid-boxes in cGEnIE and is not exported from the base of the surface layer.

**We did indeed mean more POC is remineralised in the surface layer leading to lower export, but we understand that this paragraph could have been clearer as to the exact mechanisms (including new vs. regen production) involved and have rephrased accordingly. (pg.13 ln.383-402)**

Lines 220 - 221: "warming-induced shoaling of the remineralisation depth has been modelled to reduce POC export (Kwon et al., 2009)" - this may be a typo or the wrong reference? The Kwon paper perturbs the remineralisation depth directly for a fixed climate, and it shows POC export increasing, not decreasing, with increasingly shallower remineralisation depths (e.g., Fig. 2a in Kwon for values of b>0.9).

**This was indeed a typo, which we have now corrected and rephrased these sentences to improve clarity. (pg.13 ln.395)**

Line 237: "rapidity of carbon cycling within the surface ocean" - what does this refer to? A shift from POC to DOC export production? If so, I would expect the increase in the rate of nutrient cycling associated with more semi-labile DOC production to rather increase production and biomass because it will be remineralised near the surface due its short lifetime.

**This terminology is from Finkel et al (2010) who used it (in the reverse case) to describe the difference between DOM-producing small plankton classes and POM-producing large plankton classes. However, this was not clear in our original text, and in this case the biomass decline also has an additional driver in oligotrophication, and so we have clarified this sentence and added extra explanatory paragraphs. (pg.13 ln.410 – pg.14 ln.437)**

Lines 231 - 246: plankton size outputs are available in EcoGENIE but are not plotted to support any of these statements. This would be an interesting thing to see!

**Added as Supp Figure S50 and discussed in greater detail in the text. (pg.14 ln.439 – pg.15 ln.450)**

Line 277: "adding ECOGEM reduces total ecosystem POC/PIC production" - i cannot see this in a figure and it is not described or demonstrated why this happens as a result of having ECOGEM

**This was described in section 4.1 (on how the shift to smaller sizes reduces overall productivity and biomass). However, we have added a Supp. Figure S51/54 to demonstrate this and have clarified the text here. (pg.16 ln.506)**

Lines 294 - 304: I wonder if resolving plankton biomass also plays an important role as part of this? Galbraith et al., (2015) in JAMES showed nicely that seasonal/transient behaviour varies between a model with parameterised export of POC and one that explicitly resolves plankton biomass. A parameterised model, like cGEnIE, responds much more rapidly to environmental changes because growth rates are not buffered by a biomass pool. There are a few entries in Tables 1 that this parameterised export model. Following on from this, it would be interesting to speculate what the representation of ecological complexity needs to be to reliably simulate the biological pumps response to environmental change.

**We've now mentioned the potential impact of incorporating explicit biomass in the Methods section (pg.8 ln.249-258). However, it's difficult to disentangle the impact of explicit vs implicit biomass in this study from the wider issues around parametrisation and the other changes introduced with ECOGEM (and we also focus very much on broad, long-term trends rather than seasonal/spatial issues), and so we've not delved much into this issue in the Discussion. We agree that it's an interesting question though.**

Lines 313 - 314: "does not feature trait-based size classes or flexible stoichiometry, which we have shown is critical for determining the soft-tissue biological pump response" - the role of flexible stoichiometry has not been explored here.

& Lines 325: "flexible nutrient usage" - the influence/impact of this has not really been quantified or discussed in the manuscript.

**We have now more explicitly discussed the importance of flexible stoichiometry both in modelling and in our results. (pg.4 ln.125 – pg.5 ln.136, pg.6 ln.174, pg.14 ln.146-423)**

Figure 5: it would help to combine these panels with Figure 3 so they can be compared side by side.

**Done. (pg.32)**

Figure 4: I spent most of the time thinking these differences were 2100 vs. baseline because that is the format of the other figures. Expanding the labelling might help clarify this.

**The labelling and legend has been updated in both figures to make the difference clear. (pg.33)**

Table 1: this is very valuable, thank you!  **Thanks!**

**Response to Short Comment 1 Karin Kvale et al. (SC1):**

Conceptual issues:

Page 3, Line 69: We suggest rephrasing as this is not the marine carbon sink of anthropogenic CO2 mentioned earlier in the text. In fact, the biological pump is, in steady state, neither a carbon sink nor source as it fluxes as much (organic) carbon to depth as is transported back in inorganic form to the surface ocean. The timescale describes how long it take one carbon atom to take a full loop, but it does not say anything about a timescale of a possible sink or source of carbon.

**We have clarified our description of the biological pump and its indirect relation to the ocean carbon sink. However, transient mismatches in exported vs. returned carbon can make it effectively a temporary sink on centennial timescales, much like in steady-state geological carbon sinks are balanced by volcanic emissions, and an increased geological outflux is still only temporary sink when considered on a long enough timescale. (pg.5 ln.152-155)**

Page 9, Lines 259-266: We believe this interpretation needs to be revisited. The overwhelming contribution to the ocean's uptake of anthropogenic CO2 is from the solubility pump. Though your models share the same physics, surface ocean DIC and ALK are only 'calibrated' to be similar (lines 188-190). Table S1 indicates that at least BIO+TDR has a considerably smaller surface carbonate ion concentration compared to BIO+FPR, indicating that the carbonate buffer is reduced compared to BIO, suggesting that in that model run the solubility pump should be weaker. It is actually this pair of model experiments, which shows by far the strongest difference in marine CO2 uptake (Table 2). BIO+TDR has lower CO2 uptake, consistent with the lower initial buffer, but inconsistent with the higher POC-export response.

In the follow-up paragraph (lines 268ff) this is being discussed somehow, but still analyzed as an effect of differences of biological pump representations. However, the different marine CO2 uptake seems in fact to result from different surface DIC and ALK concentrations at the end of the respective model spin-ups. If the models were calibrated to have identical surface DIC and ALK values, buffer factors and the responses of the solubility pump to increasing atmospheric CO2 would be similar, allowing a more straightforward identification of biological effects on the oceanic CO2 uptake.

**We now include the original calibrations of each configuration, which we present in the supplementary material (Supp. Figs. S61 & S64) and discuss in the text (pg.15 ln.462-468, pg.16 ln.511 – pg.17 ln.526). While there are difference between the original and new calibrations, these mostly reflect the substantial differences in baseline biological pump strength, while the general pattern of the results remain unchanged. In particular, we show that [CO3] differences are unlikely to have a substantial confounding impact on our results.**

Page 10, Line 289: Recent work has also demonstrated the importance of representing interacting Ncycle processes (such as N2 fixation and water column and benthic denitrification) to capture important feedback processes that affect biological export production (Somes et al., 2016; Landolfi et al., 2017) and potentially air-sea CO2 exchange (Buchanan et al., 2019) and ecosystem restructuring (Dutkiewicz et al., 2013). Also, redoxdependent feedbacks in nutrient cycles are not included in most current models, but may be relevant even on centennial timescales and will require an adequate representation of marine oxygen distributions (Watson, 2016, Niemeyer et al., 2017).

**These processes have been added as additional limitations to our results. (pg.17 ln.541-544)**

Unjustified statements:

Page 1, Line 22 (Abstract) and, similarly, p 10, Line 282: The last column of Table 2 shows that the addition of ecological complexity results in a maximum weakening of the ocean carbon sink capacity of 2.39 GtC over a 100 year period (a 0.4% weakening). If ecological complexity is removed and only temperature dependent remineralization is considered, the maximum weakening is still only 0.9%. These differences in marine carbon uptake, even if buffer factors were similar and effects were caused predominantly by differences in the representation of biological processes, do not strongly demonstrate the need for additional model complexity. Instead, if the same metric is applied, one might even argue that uncertainties can probably be reduced by a far larger amount by investigating better representations of, for example, physical processes.

**We have clarified that the biological pump results are more substantial than for the carbon sink, and have refocused the implications in the discussion and abstract to reflect this. (e.g. pg.1 ln.22, pg.17 ln.531, pg.19 ln.606)**

Page 10, Line 297: While we agree that such a debate is important and should always be encouraged, this statement makes the misleading impression that this paper is the first to propose potential gains from shifting some resources from even higher ocean resolution to more complex bgc models. However, what has been hindering the application of more complex biogeochemical models is not necessarily computation-power issues related to performing simulations, but the uncertainty in biogeochemical model parameters and the associated computational costs in properly calibrating these parameters. Fortunately, since a few years, biogeochemical and ecological parameter optimization has emerged as a very active field of research that exploits recent gains in computational power. Please see Frants et al. (2016), Chien et al. (2020), Schartau et al. (2017), Kriest et al. (2020), Kriest 2017, Niemeyer et al. (2019), Sauerland et al. (2019), Yao et al. (2019), to name just a few. So far most of this work has not been carried out with 'proper' ESMs, but with EMICS. To pose all discussion of ocean carbon cycle dynamics research in the framework of CMIP is misleading. Actually, including poorly calibrated more complex bgc models in high resolution ESMs is likely not in favor of more reliable marine CO2 uptake predictions. Since such calibration can be done on the EMIC level, it is rather more important to propose studies with well calibrated models to better understand the relationship between bgc complexity and marine CO2 uptake.

**We have adapted this section to directly focus on the potential of trait-based models in representing biogeochemical complexity, and removed any unintentional implication that this study was unique. (pg.18 ln.553-563, pg.19 ln.596)**

Page 11, Line 318: Please see Kvale et al. (2015, 2019) for 2 explorations of how ballasting can affect export in a temperature-dependent remineralization model (EMIC) over long-term simulations. Depending on how you

choose to represent calcification, the ballasting effect can alter the pathway to the long-term response of your model via modification of suboxic volumes, which regulate denitrification, nitrate availability, and hence primary and export production.

**Extra detail on how ballasting could affect the biological pump added, but we believe the caveat on mixed observational support still stands. (pg.19 ln.584-591)**

In summary, this manuscript could be improved on two fronts. The first is stylistic, in which the introduction and discussion of the state-of-the art should include references to the recent ecology and biological pump work happening with both EMICs and ESMs. This is important for giving proper context to the present study. The second is technical, in which the major conclusions must be shown to be independent of different states of the carbon chemistry at the end of the spin up of the respective models. This can be demonstrated, for example, by carbon separation techniques (e.g., Koeve et al., 2014) or a better model calibration that adequately controls for buffer chemistry. On technical aspects we are always happy to offer advice, and invite the manuscript authors to contact us for further discussion.

**As described above, we have both bolstered our background and discussion with greater referencing of wider EMIC work on the biological pump, and we have also presented additional results from default calibrations in order to constrain the effect of differing carbonate chemistry in our results (and show that it does not change the overall pattern of our results).**

**Additional Changes**

**In addition to the changes described above, for our main results we also reran the model with a newer version of cGEnIE with an updated TDR calibration (cGEnIE.muffin v.0.9.13; Crichton et al, 2021, GMD), which has had a marginal impact on our main results (Table 2 – N.B. cumulative values here are now 1765-2100 rather than 2000-2100).**

---

## Referee Report (RR1)

**Review of "Resolving ecological feedbacks on the ocean carbon sink in Earth system models" by Armstrong McKay et al.**

Armstrong McKay et al., have put a lot of of effort into revising the manuscript and responding to the reviewer comments, especially given the varied pressures of working during a pandemic. Overall, I think the manuscript is improved with the addition of adequate details about the experimental set up, new analysis of results and improved general presentation. The overall concept and findings are interesting to the Earth system community and are currently very relevant to a wider discussion of resolving plankton ecosystems in Earth system models.

Unfortunately I think there are still some outstanding issues with the experimental set-up and analysis that need to be resolved before I can recommend the manuscript for publication. Overall, I think the problems described in the comment by Kvale et al., combined with my own comments highlight a broader issue: can a common baseline be defined for different models and does this limit the level of quantitative analysis? This seems like a tricky issue to resolve but I have tried to outline two ways in which I think it could be achieved below as I think this is worth pursuing. I have also included a more detailed criticism of the recalibration process now that the process has been fully described, but this forms a part of the broader issue.

**Experimental Set-up**

Based on the previous reviewer comments and the clarifications in the revised manuscript, I think there is a fundamental issue in trying to define a common baseline for all the experiments. It seems essentially impossible to get a common baseline that is *exactly* the same using different biogeochemical and ecosystem models despite them sharing the same physical model. In matching one variable, such as POC export, there will always be a subsequent trade-off in another (POC remineralisation, surface carbonate chemistry) that will have an impact on the results (nutrient delivery timescales, carbon sink). A key question here is whether the findings are robust to this issue. Figures S61 and S64 show some insight into this question. Figure S61 shows some level of agreement in the trends of POC export across different baselines but disagreement in transient behaviour and magnitudes. Figure S64 shows disagreement in both magnitude and sign of the trends in the ocean carbon sink. However, trends in S64 are relative to the BIO+FPR run not the individual baseline as in S61 so it's hard to tell whether the presentation choice is a factor here.

Unfortunately I think this is a difficult issue to deal with. I have two suggestions for resolving this:

1) The issue could partially be resolved by presenting changes relative to the corresponding baselines as is now done for POC but not for the ocean carbon sink. Then, at least, there is a clear distinction between experiments. There needs to be additional discussion that details that the response of models with varying complexity has two components: a dynamical response to environmental change driven by the model itself, and a dependence on the initial state that is inherent in using different models. The downside of this approach is that it really limits the findings to more semi-quantitative comparative descriptions because it's very difficult to separate out the impacts of the dynamical-responses from the initial state. It's not obvious even that this would be consistent across the experiments. In my opinion, the recalibration process used adds

additional biases (see comment below) and is arguably not necessary if the experiments are compared to their own baseline anyway. I would therefore strongly suggest presenting the results using the default versions of the model. This actually facilitates a broader discussion that has more relevance to the wider modelling community given that models are replaced by newer versions and assessed against a broad range of metrics, e.g., Seferian et al., (2020).

2) To resolve the baseline issue completely the experiments need to be run using the same model set up for each experiment. The ECO+TDR model should be used to create a single preindustrial spin-up. The impact of temperature-dependent remineralisation, size-dependent partitioning of DOC:POC and non-Redfield stoichiometry can then be quantified by controlling each element. For example, the FPR experiment can be replicated by forcing the remineralisation to "see" the preindustrial temperature field, thereby causing it to behave as the FPR experiment but not deviate as a baseline. Similarly, one could control for elements of the ecosystem model such as the size-dependent POC:DOC export or stoichiometry. This would allow the authors to quantify the influence of each component to a much greater extent and reliability. However, this approach requires some adjustments to the model to enable this and extensive revision of the text.

**Biological Pump - Recalibration and Interpretation Issues**

The authors have now fully detailed the recalibration process involving POC remineralisation. I understand the justification for recalibration but I think this adds additional biases to the findings that are not quantified or even acknowledged in places. The crucial issue here is that the authors achieve the same global POC export production across the different model set-ups by altering the fraction of export that is remineralised as refractory POC. In GENIE, export is divided into "labile" POC (~95%) that attenuates strongly across the upper 1000m and "refractory" POC (~5%) that attenuates minimally with most POC remineralising in the grid-boxes overlying the seafloor boundary. The authors defend the calibration by stating that "biological pump perturbations on sub-overturning timescales (<500-1000 years) will not significantly affect surface DIC..." (lines 295 - 300).

I strongly disagree with the author's defence. There is an average characteristic lifetime of regenerated DIC (and correspondingly, nutrients) that is a function of ocean ventilation times (First Passage Time: Primeau 2005) and remineralisation rates. By lowering the labile:refractory export partitioning the authors are increasing the average lifetime of regenerated DIC and nutrients in the ocean. I agree that the ventilation time of DIC from the deepest ocean to the surface is predominantly longer than the timescales analysed, but, this is compensated by reducing the amount of regenerated DIC entering the intermediate ocean where the ventilation timescales are relevant to the timescales analysed.

To demonstrate this as an issue, I have run some idealised experiments in an offline transport-matrix based version of GENIE (in-preparation for publication based on earlier work described in Wilson et al., 2015). The circulation is diagnosed from the equilibrium annual-mean circulation in GENIE at the native resolution. The biogeochemistry model is the same as reported in Ridgwell et al., 2007 and Cao et al., (2009) which is the same as the BIO+FPR set-up used by the authors. I created two spin-ups with a simple phosphorus cycle: one using the same set-up as the BIO+FPR experiment and one where

I increase the refractory export of POC to 35% as per the author's re-calibration. Each run is then continued for 500 years with an immediate cessation of biological uptake and DOP remineralisation. The surface ocean is also subject to a zero boundary condition, i.e., supplied PO4 is removed at each timestep to isolate the ventilation of PO4 from the interior ocean. Because the circulation is static any differences in the transient response of interior-to-surface PO4 supply results from the difference in initial distribution of PO4 associated with the re-calibration of refractory export. The spin-up global mean concentrations of PO4 in the ocean interior are 2.19 µmol kg-1 and 2.21 µmol kg-1 for the default and recalibration set-ups respectively.

Figure 1 shows that the re-calibrated model does have a different transient behaviour well within the timescales explored in the manuscript. Both the supply rate (Fig. 1A) and cumulative supply (Fig. 1B) of PO4 to the surface ocean are correspondingly lower for calibrated run with deeper remineralisation. Whilst this is a simplified scenario, it demonstrates that the transient adjustment of nutrients (and carbon) in the ocean interior in response to a decrease in export production is impacted by this re-calibration. As such, the recalibration has some impact on the transient features of export production and the air-sea gas exchange of carbon that may in-part explain the differences between set-ups seen in Figures S61 and S64.

It is notable that changes in remineralisation are generally not considered in the analysis and discussion of the results. It is more complex with the TDR model as the transient behaviour is driven by the rate of warming across the ocean water column but it is likely that it will have some impact.

**Figure 1.** Transient changes in global PO4 supply to the surface ocean over 500 years following a complete cessation of the biological activity for the default (solid line) and recalibrated (dashed line) BIO+FPR spin-ups. Panel A shows the supply rate of PO4 (Pmol year-1). Time is shown on a log scale to show the initial rapid change. Panel B shows the cumulative supply of PO4 to the surface (Pmol). All experiments are run from a spun-up initial state using an offline version of GEnIE.

**Specific Comments:**

Line 65: "weakening of the biological pump" - this phrasing is used throughout the manuscript. Weakening and strengthening are used in various ways by the wider community from referring to export production and the total sequestered carbon (Csoft). Because you are not quantifying Csoft, these terms need to clearly defined.

Line 84: "follows a power law distribution" - this is somewhat pedantic but Cael & Bisson (2018) showed that a power law is no better a description (statistically) of the Martin Curve sediment trap data than other functions.

Line 140: "global deepening of 24m" - I feel like "of the e-folding depth" is missing in this sentence.

Lines 142 - 144 - I appreciate this was a point from another reviewer and I agree that at steady state the pump is neither a source or sink. But all things being equal (and assuming a closed system w.r.t. CaCO3 sediments) a "stronger" pump, either through higher export or deeper remin), will be associated with lower atm. CO2, e.g., the relationship between CO2 and Cbio in Goodwin et al., (2008). I think there is a conflation between source/sink of carbon in a transient sense and equilibrium states of atm CO2 here.

Lines 348 - 350: "more POC is remineralised within the surface layer" - this is a misunderstanding of what is happening in GENIE. For the FPR runs the exponential remineralisation curve is normalised to the base of the surface grid-boxes, i.e., no POC remineralisation occurs within the surface boxes. I believe this is the same for the temperature-dependent remineralisation scheme - particles sink explicitly from the base of the surface layer. As such, it's tricky and potentially misleading to define new and regenerated production in this way. This section needs to be reanalysed and presented using the correct understanding of what is happening in GENIE.

Line 374 - I am struggling to follow the logic of mean cell size becoming smaller and extending the number of trophic levels. In this model trophic levels are primarily initiated by the presence of size-dependent grazing.

Lines 383 - 384: "...the amount of carbon exported for every unit of phosphorus increases with warming in response to stratification, reducing surface phosphorus loss..." - PO4 is the model currency here not DIC so C is changing relative to P.

Line 387: It's worth noting that Wilson et al., (2018) is showing equilibrium results which, though related, are not directly comparable to the transient results here.

Line 393: "allowed" instead of "made"?

Line 397: there is a problem with the sentence structure.

Lines 431 - 436: see the comment for lines 142 - 144. There is maybe a conflation between transient source/sinks and equilibrium CO2.

Lines 450 - 463: There is no discussion of remineralisation changes here!

Lines 450 - 463: It would be useful to state that the circulation response (temperature and stratification) are the same across the experiments here.

Line 461: "...and so adding TDR results in a synergistic interaction with ocean acidification" - to me this does not follow logically and the details of how TDR interacts is not well described.

**References**

Cao et al., (2009) The role of ocean transport in the uptake of anthropogenic CO2. *Biogeosciences.*

Goodwin et al., (2008) Analytical relationships between atmospheric carbon dioxide, carbon emissions, and ocean processes. *Global Biogeochemical Cycles.*

Primeau (2005) Characterizing Transport between the Surface Mixed Layer and the Ocean Interior with a Forward and Adjoint Global Ocean Transport Model. *Journal of Physical Oceanography.*

Ridgwell et al., (2007) Marine geochemical data assimilation in an efficient Earth System Model of global biogeochemical cycling. *Biogeosciences.*

Seferian et al., (2020) Tracking Improvement in Simulated Marine Biogeochemistry Between CMIP5 and CMIP6. *Current climate change reports*

Wilson et al., (2015) Can organic matter flux profiles be diagnosed using remineralisation rates derived from observed tracers and modelled ocean transport rates? *Biogeosciences.*

**Reviewed by Jamie Wilson**

---

## Author Response (AR2)

**Authors' response to 2nd round reviewers' comments on "*Resolving ecological feedbacks on the ocean carbon sink in Earth system models*" (Manuscript reference: esd-2020-41)**

Dear Prof. Crucifix and Reviewers,

Thank you for your second round of reviews of our manuscript "*Resolving ecological feedbacks on the ocean carbon sink in Earth system models*". Please find below our detailed responses to the reviews. We include the original comments and our response under each point in **bold** text and with line numbers referencing the revised manuscript with changes marked.

We hope we have sufficiently answered your queries in our response.

Yours sincerely,

David I. Armstrong McKay & co-authors

**Response to Reviewer 1, Anonymous (R1):**

I'd like to thank the authors for addressing the majority of the points identified. I have a few follow-up points on some of the responses, and a couple of suggestions the authors might like to consider that may help improve the manuscript.

Follow-up:

- A significant outstanding omission remains the contextualising with CMIP5 (I accept the authors' stated issues with CMIP6). The authors assert that there is little information on this, but omit Fu et al. (2016)'s quite comprehensive study on this specific subject. This should provide helpful relevant context that the manuscript is currently weak on.

**The headline result of Fu et al 2016 for export production closely reflects those of Bopp et al (2013) already cited (both give 7-18% EP decline in 2090s vs. 1990s, to which we add ~2% for inter-comparison with a preindustrial baseline), but we have included Fu et al here as well for completeness (lines 344-346).**

- While the manuscript is now clearer on how CaCO3 production is parameterised in the model, I was a little confused about how it dissolves down the water column. Line 218-219 states …

"BIOGEM by default uses a fixed remineralisation profile similar to the Martin curve for the sinking labile fractions of both POC and PIC"

… which tends to suggest that PIC dissolution follows the Martin curve. More typically it is assumed that it has a relationship with the CCD, beneath which carbonate is undersaturated and dissolution occurs.

**PIC remineralisation/dissolution follows a similar procedure to POC in this version of cGEnIE, in that it's parameterised to fit a curve according to a characteristic length scale. This is indeed not entirely typical, although the original wording wrongly implied that both POC and PIC both followed the same Martin-like Curve rather than different ones, which has now been clarified (line 221)**

- I previously asked about mixing in the study's model …

"Ln. 169: does the model resolve seasonal mixing of different ecological regimes?

No, but insolation and light attenuation is seasonal, and ecoGEnIE does reproduce seasonal variation in primary production. (pg.8 ln.249 – pg.9 ln.263)"

This suggests to me that there is no seasonal mixing. However, I'm a little confused because the new manuscript revision explicitly mentions (ln. 229-230) …

"photosynthesis … subject to light limitation, photoacclimation, and seasonal light attenuation within a variable mixed layer depth"

Figure S48 also tends to suggest that there is a representation of the mixed layer, so there's perhaps just a mix-up in the description that needs clearing up.

**Apologies, the original response was mistaken – cGEnIE has a mixed layer scheme based on Kraus & Turner 1967, which of course is a seasonal thermocline model. This has been made clear in the preceding section to avoid confusion (lines 215-216).**

Additional specific points:

- As an aside, it would probably increase reader interest in this manuscript if the headline changes in carbon cycle processes were quantified in the abstract. This block of text could really benefit from having numbers (e.g. from Table 2) added where suggested:

**We have added headline figures to the abstract as suggested (lines 18-22).**

- Supplementary figures that show OBS and then comparative model deltas could be better arranged to allow readers to directly compare the plots. At present, readers must flip back and forth to compare plots making something that should be easy needlessly difficult. For example, Figures S1, S3 and S5 (and S2, S4 and S6) should be rearranged onto the same page so that readers can easily compare patterns of bias. This should be repeated throughout the supplementary material. Also, Figure S13 shows the model-obs difference, but without showing the observations, so the reader cannot tell how large the biases really are.

**We have reorganised the relevant supplementary figures to group maps and depth plot model-observation comparisons together. However, fitting 3 plots per page would significantly reduce the figure resolution and make some features difficult to see, and the reorganisation significantly reduces the need to flip back and forth, so we have kept the plots at full size. We have also added the SeaWiFS observational data in a new figure prior to Figure S13.**

**Response to Reviewer 2, Jamie Wilson (R2):**

Armstrong McKay et al., have put a lot of effort into revising the manuscript and responding to the reviewer comments, especially given the varied pressures of working during a pandemic. Overall, I think the manuscript is improved with the addition of adequate details about the experimental set up, new analysis of results and improved general presentation. The overall concept and findings are interesting to the Earth system community and are currently very relevant to a wider discussion of resolving plankton ecosystems in Earth system models.

Unfortunately I think there are still some outstanding issues with the experimental set-up and analysis that need to be resolved before I can recommend the manuscript for publication. Overall, I think the problems described in the comment by Kvale et al., combined with my own comments highlight a broader issue: can a common baseline be defined for different models and does this limit the level of quantitative analysis? This seems like a tricky issue to resolve but I have tried to outline two ways in which I think it could be achieved below as I think this is worth pursuing. I have also included a more detailed criticism of the recalibration process now that the process has been fully described, but this forms a part of the broader issue.

**It is indeed a tricky issue, and as discussed in this and the previous revision round neither recalibrating nor directly comparing different configurations fully gets round the issue. We agree that it needs ameliorating and discussing, and outline how we do so now below.**

Experimental Set-up

Based on the previous reviewer comments and the clarifications in the revised manuscript, I think there is a fundamental issue in trying to define a common baseline for all the experiments. It seems essentially impossible to get a common baseline that is *exactly* the same using different biogeochemical and ecosystem models despite them sharing the same physical model. In matching one variable, such as POC export, there will always be a subsequent trade-off in another (POC remineralisation, surface carbonate chemistry) that will have an impact on the results (nutrient delivery timescales, carbon sink). A key question here is whether the findings are robust to this issue. Figures S61 and S64 show some insight into this question. Figure S61 shows some level of agreement in the trends of POC export across different baselines but disagreement in transient behaviour and magnitudes. Figure S64 shows disagreement in both magnitude and sign of the trends in the ocean carbon sink. However, trends in S64 are relative to the BIO+FPR run not the individual baseline as in S61 so it's hard to tell whether the presentation choice is a factor here.

Unfortunately I think this is a difficult issue to deal with. I have two suggestions for resolving this:

1) The issue could partially be resolved by presenting changes relative to the corresponding baselines as is now done for POC but not for the ocean carbon sink. Then, at least, there is a clear distinction between experiments. There needs to be additional discussion that details that the response of models with varying complexity has two components: a dynamical response to environmental change driven by the model itself, and a dependence on the initial state that is inherent in using different models. The downside of this approach is that it really limits the findings to more semi-quantitative comparative descriptions because it's very difficult to separate out the impacts of the dynamical-responses from the initial state. It's not obvious even that this would be consistent

across the experiments. In my opinion, the recalibration process used adds additional biases (see comment below) and is arguably not necessary if the experiments are compared to their own baseline anyway. I would therefore strongly suggest presenting the results using the default versions of the model. This actually facilitates a broader discussion that has more relevance to the wider modelling community given that models are replaced by newer versions and assessed against a broad range of metrics, e.g., Seferian et al., (2020).

2) To resolve the baseline issue completely the experiments need to be run using the same model set up for each experiment. The ECO+TDR model should be used to create a single preindustrial spin-up. The impact of temperature-dependent remineralisation, size-dependent partitioning of DOC:POC and non-Redfield stoichiometry can then be quantified by controlling each element. For example, the FPR experiment can be replicated by forcing the remineralisation to "see" the preindustrial temperature field, thereby causing it to behave as the FPR experiment but not deviate as a baseline. Similarly, one could control for elements of the ecosystem model such as the size-dependent POC:DOC export or stoichiometry. This would allow the authors to quantify the influence of each component to a much greater extent and reliability. However, this approach requires some adjustments to the model to enable this and extensive revision of the text.

**Approach 2) would indeed give a thorough experimental basis, but there is the issue that ecoGEnIE itself (i.e. ECO+FPR) is yet to be fully recalibrated (and is still missing some processes). Because of this, using the ECO+TDR spin-up as the basis for all runs as suggested would still have limitations affecting the robustness of specific numbers.**

**We have chosen to follow approach 1), and have updated figures and the text throughout the manuscript to reflect this (e.g. lines 289-310, 359-404, 435-459, 472-483, 510-538, 545-550, 607; Table 2; Figures 2-5, S48, S62, & S67). As the reviewer notes, this means our approach is indeed more semi-quantitative – a point also relevant to the missing calcifier/silicifier classes and other important processes mentioned in the Discussion. Nevertheless, the trends that emerge from across the different calibrations despite these limitations (relative biological pump strengthening and weakening for activating TDR and ECO respectively, translating to relative ocean carbon sink weakening and strengthening in turn) is a useful outcome that demonstrates the relevance of including ecological and metabolic dynamics in Earth system models.**

**We have also replotted the absolute ocean carbon sink capacity as requested, however we have presented it alongside the original relative plot rather than instead of. This is because unlike for POC export we cannot plot Air-to-Sea CO2 flux as a % relative to preindustrial baseline (as by definition the ASG flux at preindustrial was close to zero), and in the absolute cumulative ocean carbon plot it is hard to tell apart some of the curves. As a result, we now show the absolute cumulative ocean carbon plot as context as well as the relative cumulative ocean carbon plot to show the detailed differences between the scenarios.**

Biological Pump - Recalibration and Interpretation Issues

The authors have now fully detailed the recalibration process involving POC remineralisation. I understand the justification for recalibration but I think this adds additional biases to the findings that are not quantified or even acknowledged in places. The crucial issue here is that the authors achieve the same global POC export production across the different model set-ups by altering the fraction of export that is remineralised as

refractory POC. In GENIE, export is divided into "labile" POC (~95%) that attenuates strongly across the upper 1000m and "refractory" POC (~5%) that attenuates minimally with most POC remineralising in the grid-boxes overlying the seafloor boundary. The authors defend the calibration by stating that "biological pump perturbations on sub-overturning timescales (<500-1000 years) will not significantly affect surface DIC…" (lines 295 - 300).

I strongly disagree with the author's defence. There is an average characteristic lifetime of regenerated DIC (and correspondingly, nutrients) that is a function of ocean ventilation times (First Passage Time: Primeau 2005) and remineralisation rates. By lowering the labile:refractory export partitioning the authors are increasing the average lifetime of regenerated DIC and nutrients in the ocean. I agree that the ventilation time of DIC from the deepest ocean to the surface is predominantly longer than the timescales analysed, but, this is compensated by reducing the amount of regenerated DIC entering the intermediate ocean where the ventilation timescales are relevant to the timescales analysed.

To demonstrate this as an issue, I have run some idealised experiments in an offline transport-matrix based version of GENIE (in-preparation for publication based on earlier work described in Wilson et al., 2015). The circulation is diagnosed from the equilibrium annual-mean circulation in GENIE at the native resolution. The biogeochemistry model is the same as reported in Ridgwell et al., 2007 and Cao et al., (2009) which is the same as the BIO+FPR set-up used by the authors. I created two spin-ups with a simple phosphorus cycle: one using the same set-up as the BIO+FPR experiment and one where I increase the refractory export of POC to 35% as per the author's re-calibration. Each run is then continued for 500 years with an immediate cessation of biological uptake and DOP remineralisation. The surface ocean is also subject to a zero boundary condition, i.e., supplied PO4 is removed at each timestep to isolate the ventilation of PO4 from the interior ocean. Because the circulation is static any differences in the transient response of interior-to-surface PO4 supply results from the difference in initial distribution of PO4 associated with the re-calibration of refractory export. The spin-up global mean concentrations of PO4 in the ocean interior are 2.19 µmol kg-1 and 2.21 µmol kg-1 for the default and recalibration set-ups respectively.

Figure 1 shows that the re-calibrated model does have a different transient behaviour well within the timescales explored in the manuscript. Both the supply rate (Fig. 1A) and cumulative supply (Fig. 1B) of PO4 to the surface ocean are correspondingly lower for calibrated run with deeper remineralisation. Whilst this is a simplified scenario, it demonstrates that the transient adjustment of nutrients (and carbon) in the ocean interior in response to a decrease in export production is impacted by this re-calibration. As such, the recalibration has some impact on the transient features of export production and the air-sea gas exchange of carbon that may in-part explain the differences between set-ups seen in Figures S61 and S64.

**This is a good point, and we thank the reviewer for providing a model demonstration clearly illustrating the issue. Although recalcitrant POC reaching the seafloor wouldn't impact surface waters on millennial timescales, of course a higher recalcitrant fraction would also have the effect of depriving intermediate waters of remineralised nutrients as well with consequences emerging on nearer-term timescales. We have added discussion of this issue as a confounding factor for the recalibrated configurations (which have also now been adjusted to be supplementary rather than main results) in both the Methods and Results section.**

**As discussed previously, either using the default or recalibrated configurations leads to different problematic confounding factors (that could be partly but not wholly dealt with by using one configuration), but the broad trends that emerge regardless and the dynamics they reveal remain as useful results that demonstrate the role of ecological complexity in Earth system models.**

It is notable that changes in remineralisation are generally not considered in the analysis and discussion of the results. It is more complex with the TDR model as the transient behaviour is driven by the rate of warming across the ocean water column but it is likely that it will have some impact.

**We have added more specific discussion of the impact of remineralisation changes in our discussion of the carbon sink capacity results (lines 485-508), which supplements existing discussion of remineralisation in the biological pump results (lines 362-383 & 452-456).**

Specific Comments:

Line 65: "weakening of the biological pump" - this phrasing is used throughout the manuscript. Weakening and strengthening are used in various ways by the wider community from referring to export production and the total sequestered carbon (Csoft). Because you are not quantifying Csoft, these terms need to clearly defined.

**We have clarified here at the first mention of biological pump weakening that we define it as reduced POC export (line 68), and have also clarified this POC export focus elsewhere (e.g. in the abstract).**

Line 84: "follows a power law distribution" - this is somewhat pedantic but Cael & Bisson (2018) showed that a power law is no better a description (statistically) of the Martin Curve sediment trap data than other functions.

**Edited to "power law-like" for accuracy, but keeping power law as a commonly understood function shape (line 86).**

Line 140: "global deepening of 24m" - I feel like "of the e-folding depth" is missing in this sentence.

**Clarified (line 142).**

Lines 142 - 144 - I appreciate this was a point from another reviewer and I agree that at steady state the pump is neither a source or sink. But all things being equal (and assuming a closed system w.r.t. CaCO3 sediments) a "stronger" pump, either through higher export or deeper remin), will be associated with lower atm. CO2, e.g., the relationship between CO2 and Cbio in Goodwin et al., (2008). I think there is a conflation between source/sink of carbon in a transient sense and equilibrium states of atm CO2 here.

**We agree that a change in the biological pump will lead to a transient carbon sink/source even if this is not the case for the equilibrium state, and that in the long run a system with a stronger biological pump would store more carbon in the ocean, but in responding to the short comment the initial point was not so clear in the manuscript. We have further adjusted this and at the start of section 4.3 to make this clearer (lines 143-146).**

Lines 348 - 350: "more POC is remineralised within the surface layer" - this is a misunderstanding of what is happening in GENIE. For the FPR runs the exponential remineralisation curve is normalised to the base of the surface grid-boxes, i.e., no POC remineralisation occurs within the surface boxes. I believe this is the same for the temperature-dependent remineralisation scheme - particles sink explicitly from the base of the surface layer. There is remineralisation of *dissolved* organic carbon in the surface layer. As such, it's tricky and potentially misleading to define new and regenerated production in this way. This section needs to be reanalysed and presented using the correct understanding of what is happening in GENIE.

**You are of course correct here, and this section was poorly phrased and constructed. We have edited and rearranged the text here to focus on the effect of the remin. depth shoaling towards cGEnIE's subsurface layers, which is what actually drives the relative increase in export in cGEnIE (lines 364-366).**

Line 374 - I am struggling to follow the logic of mean cell size becoming smaller and extending the number of trophic levels. In this model trophic levels are primarily initiated by the presence of size-dependent grazing.

**This explanation was unclear, in that it referred to a hypothesis by Riebesell et al 2009 that has a parallel to but in fact doesn't directly apply to cGEnIE. We have edited this sentence to make it clearer that the primary driver of reduced biomass is stratification-induced nutrient decline, and removed the trophic level discussion as being tangential (lines 393-396).**

Lines 383 - 384: "…the amount of carbon exported for every unit of phosphorus increases with warming in response to stratification, reducing surface phosphorus loss…" - PO4 is the model currency here not DIC so C is changing relative to P.

**"reducing surface phosphorus loss" has been removed as you are correct in that P is not being changed here, but the overall effect ameliorating carbon export decline is still the case (as more C is lost per unit export) (line 404).**

Line 387: It's worth noting that Wilson et al., (2018) is showing equilibrium results which, though related, are not directly comparable to the transient results here.

**Paragraph edited to make this clearer (lines 406-420).**

Line 393: "allowed" instead of "made"?

**Changed (line 413).**

Line 397: there is a problem with the sentence structure.

**Adjusted and split into two sentences for clarity (lines 418-420).**

Lines 431 - 436: see the comment for lines 142 - 144. There is maybe a conflation between transient source/sinks and equilibrium CO2.

**Extra detail added to clarify that this is the case for comparing equilibrium states such as with warm palaeoclimates, and that our critique here is targeted at misapplication to the modern transient case (lines 461-46).**

Lines 450 - 463: There is no discussion of remineralisation changes here!

Lines 450 - 463: It would be useful to state that the circulation response (temperature and stratification) are the same across the experiments here.

**Remineralisation changes were implicit in the description of the initial production changes impacting DIC and ALK, but could have been spelled out better as a key factor in surface DIC/pCO2 increase – this has now been clarified, along with the supplementary schematic updated to include more mechanisms and the parallels to Kwon et al (2009)'s mechanisms made explicit. We have also clarified in the previous paragraph and in the 4.1 physical climate response section at the start of the Results that the climate response is almost identical across the experiments, and so the observed differences must be biogeochemically driven (lines 485-508).**

Line 461: "…and so adding TDR results in a synergistic interaction with ocean acidification" - to me this does not follow logically and the details of how TDR interacts is not well described.

**We simply mean here that both warming+TDR and OA result in increasing surface pCO2 and decreasing saturation state, and so adding TDR slightly worsens OA. We have edited this sentence to make this clearer, and remove any implication that more complex interactions are being described (lines 498-501).**